# A highly energy-efficient multi-core neuromorphic architecture for training deep spiking neural networks

Mingjing Li[1,8], Huihui Zhou [1,2,3,8] ✉, Xiaofeng Xu[1,8], Zhiwei Zhong[1,8], Puli Quan[1], Xueke Zhu[1], Yanyu Lin [1], Wenjie Lin[1], Xiaosha Li[1], Dong Wang[1], Junchao Zhang[1], Yunhao Ma [1,4], Xiaole Cui[1,5], Wei Wang [1], Qingyan Meng[1], Zhengyu Ma [1], Guoqi Li[1,6] ✉, Xiaoxin Cui [1,7] ✉ & Yonghong Tian [1,2,3] ✉

There is a growing necessity for edge training to adapt to dynamically changing environments. Neuromorphic computing represents a significant pathway for highly efficient intelligent computation in energy-constrained edges, but existing neuromorphic architectures lack the ability of directly training spiking neural networks based on backpropagation. We developed a multi-core neuromorphic architecture with Feedforward-Propagation, Back-Propagation, and Weight-Gradient engines in each core, supporting highly efficient parallel computing at both the engine and core levels, achieving 190%~330% performance of Jetson Orin. It combines various data flows and sparse computation optimization by fully leveraging the sparsity in spiking neural network training, obtaining a high energy efficiency of 1.05TFLOPS/W@ FP16 @ 28 nm, 55~85% reduction of memory access compared to A100 GPU in the training. Additionally, we deployed the architecture on Field Programmable Gate Arrays, successfully demonstrating 20-core deep spiking network training and 5-worker federated learning. Our study develops the first multi-core neuromorphic architecture supporting direct training of spiking neural network, facilitating neuromorphic computing in edge-learnable applications.

In the wake of escalating demands for intelligent computing, conventional computing architectures, such as GPUs, increasingly grapple with challenges of energy efficiency[1], especially in the energy-constrained edge applications[2]. Due to the demands for privacy protection and low-latency high-accuracy processing, and limitations of cloud-edge transmission bandwidth, there is a growing necessity for model training at the edge to adapt to dynamic environments[3–5]. Neuromorphic computing, a paradigm aimed at emulating the structure and functionality of the human brain, offers a promising pathway towards high-energy-efficiency intelligent computing systems[6–8] especially for edge applications[2].

Spiking Neural Networks (SNNs), a cornerstone of neuromorphic computing[9], have attracted considerable attention owing to their potential for high energy efficiency and fidelity to biological neural processes[6,10,11]. Through the direct training methods based on Back-Propagation (BP) with surrogate gradient learning[12–15] there have been rapid advancements in the performance of deep SNN models in recent years, bringing them close to the performance level of conventional

[1]Pengcheng Laboratory, Shenzhen, China. [2]School of AI for Science, Peking University Shenzhen Graduate School, Shenzhen, China. [3]Beijing Key Laboratory of Brain-inspired Spiking Large Models, School of Computer Science, Peking University, Beijing, China. [4]Southern University of Science and Technology, Shenzhen, China. [5]School of Electronics and Computer Engineering, Peking University Shenzhen Graduate School, Shenzhen, China. [6]Institute of Automation, Chinese Academy of Sciences, Beijing, China. [7]School of Integrated Circuits, Peking University, Beijing, China. [8]These authors contributed equally: Mingjing Li, Huihui Zhou, Xiaofeng Xu, Zhiwei Zhong. ✉e-mail: zhouhui_h@qq.com; guoqi.li@ia.ac.cn; cuixx@pku.edu.cn; yhtian@pku.edu.cn

Artificial Neural Networks (ANNs)[15–18] indicating substantial potentials in intelligent edge applications such as sensory processing, robotics, UAV, embodied intelligence, and federated learning[19–29].

Neuromorphic processors supporting SNN computation have been developed quickly, such as TrueNorth[30], Loihi[31], Tianjic[24], BrainScale[32,33], Neurogrid[34], DYNAPs[35], Darwin[36]. These neuromorphic architectures commonly adopt a 2D Mesh multi-core structure, whose computing powers are generally below 1000 GOPS with computational precision from 1 bit to 9 bits[2]. Early architectures are designed mainly for brain simulation using sparsely activated neural networks with certain connection randomness (e.g., SpiNNaker, 2013[37]; TrueNorth, 2014). Recent architectures (Loihi 2018; Loihi2 2021[38]; Darwin3 2024; etc.) also support more structured networks such as spiking convolutional networks with higher computation intensity, while Tianjic (2019) supports DNN/SNN hybrid neural network models. For on-chip learning capabilities, TrueNorth (2014) only supports forward inference, DYNAPs (2017) supports Hebbian learning, Loihi (2018) enables two-factor local learning (e.g., STDP), and Loihi2 (2022) and Darwin3 (2024) introduce more complex three-factor local learning (e.g., RSTDP, SDSP). However, these local learning methods only support shallow network structure optimization[39]. BP-based direct training of deep SNNs has not been achieved on existing multi-core neuromorphic architectures.

High-efficiency computation for SNN training still faces fundamental challenges in parallel computation, inter- and intra-core communication, and storage usage. First, diverse computational forms, including both floating-point and binary (0/1) spike-based computations, and complex data dependencies across computing cores during SNN training, encompassing both spatial and temporal dimensions, make it difficult to achieve high-efficiency parallel computing. Second, bandwidth demand for data transmission is far more than that required in inference, and complex data transmission along both model layers and time steps in both sequent and reverse orders, increases the complexity for inter- and intra-core communication design. Finally, deep SNN training generates a large volume of intermediate data, requiring extensive access to off-chip memory (e.g., DRAM), leading to high power consumption. Reducing DRAM access is particularly significant for SNN training. To overcome the limited learning ability of existing neuromorphic architectures, this study aims to develop a highly efficient multi-core neuromorphic architecture supporting BP-based direct training of deep SNNs.

To the best of our knowledge, we propose the first multi-core, sparsity-aware neuromorphic architecture enabling distributed deployment of the entire deep SNN model across cores to perform BP-based training. We develop the highly parallelable computing core, efficient intra- and inter-core communication mechanisms, multi-level parallelism strategy, and comprehensive sparse computation optimization specifically tailored for SNN training. In the neuromorphic architecture aspect, we propose the three-engine computing core inter-connected with a 2D mesh network. The Feedforward-Propagation (FP), Back-Propagation (BP), and Weight-Gradient (WG) engines based on PE-array with FP16 Adder or MAC execute diverse computational forms in SNN training. Communication mechanisms featuring the six-way router, three primary inter-core/engine (FP-FP, BP-BP, FP-BP) pathways, and unified package format for bit-based spiking and floating-point data, enable efficient data transmission along both model layers and time steps across computing engines. The 2D mesh topology and Globally Asynchronous Locally Synchronous (GALS) approach deployed in our multi-core architecture support its high scalability. To accelerate SNN training, we develop a multi-level parallel computing strategy with inter-core pipeline parallelism and intra-core engine parallelism. At the inter-core level, our multi-core architecture performs parallel feedforward and backward operations on different model layers simultaneously. Within the same core, the three engines perform FP, BP, and WG computation in parallel. For

energy efficiency optimization, we introduce sparsity-aware circuits in the three engines, leveraging all types of sparse signals in SNN training to skip redundant computations and memory access, reducing energy consumption by 45%–60%. Through multiple dataflows in different engines to enhance data reuse and single-bit storage of spiking data, SNN training in our architecture achieves 55 ~ 85% reductions of the high-energy-consuming DRAM access compared to the A100 GPU. Overall, our architecture achieves 190% ~ 330% performance of Jetson AGX Orin, with a high energy efficiency of 1.05TFLOPS/W@ FP16 @ 28 nm. Along with our software toolchain, our architecture is implemented on FPGAs and achieves successful supervised learning and federated learning. Together, our work significantly enhances the training efficiency of neuromorphic models and broadens the applications of neuromorphic architectures in the resource-constrained scenario, especially in edge-learnable environments.

## Results
Fig. 1a highlights the advantages, challenges, and our proposed solutions for neuromorphic computing. The training process of SNN (Figs. 1b and 1c) with the Leaky Integrate-and-Fire (LIF) neuron model includes the FP, BP, WG stages[40,41]; The SNN parameters $N, T, H, W, C, R, S, M, E, F$ are described in Supplementary Table 2, and Fig. 1d shows examples of membrane potential updating and spike generation in LIF neurons during FP.

### Instruction set
We designed an instruction set to support SNN training (Supplementary. Table 1). Convolution instructions include FP, BP, WG convolutions, supporting different kernel sizes, strides, interleaves, and padding configurations (Eqs. (2), (7), and (8) in Methods). Soma/Grad instructions support the LIF model, including Soma operations during FP (Eqs. (1) and (3) in Methods) and Grad operations during BP (Eqs. (4) and (6) in Methods). BN instructions support batch normalization during the FP and BP. Vector instructions support cross-core accumulation.

### Multi-core near-memory neuromorphic architecture for SNN training
We proposed a multi-core near-memory neuromorphic architecture that supports FP, BP, and WG computations to achieve efficient deep SNN training. The register-transistor level (RTL) design was completed and verified in both software and FPGA hardware. Fig. 2a shows the design of a single computing core within the multi-core architecture (Fig. 2b), including an FP Sub-Core, a BP Sub-Core, and a Router. DRAM is shared by all computing cores in the architecture.

The FP computing engine in the FP Sub-Core implements the forward inference of SNN, including FP Array, Soma, Fire, Pooling, and related SRAM (Fig. 2a). The FP Array consists of a 16×16 selector and adder array with FP16 precision. Its input includes spike signals **s** and the weights $w$. Since the spike signals are either 0 or 1, the FP engine performs the conventional multiply-add convolution by selector-gated addition operations, that is, only adders in the column of FP array with input spike value of 1 perform addition on weight values. The FP Array performs accumulation of $w$ across the $C, R, S$ and $T$ dimensions to generate 16$M$ dimension convolution results **Conv_FP**. The spike signal $s$ is represented by a single bit to save storage space. The Soma module (Fig. 2c) updates the neuron's membrane potential based on its previous membrane potential, leakage factor $\alpha$, spike firing state, and **Conv_FP** according to Eq. (2) in the Methods. The membrane potential is compared with a threshold to generate the output spike signal in its Fire sub-module. An FP engine includes 16 Soma modules, corresponding to 16 $M$ dimension channels. The Pooling module supports max pooling operations on the output spike signals. The BP Sub-Core includes a BP Engine and a WG Engine (Fig. 2a), which computes membrane potential gradients **∇u** and weight gradients **∇w**

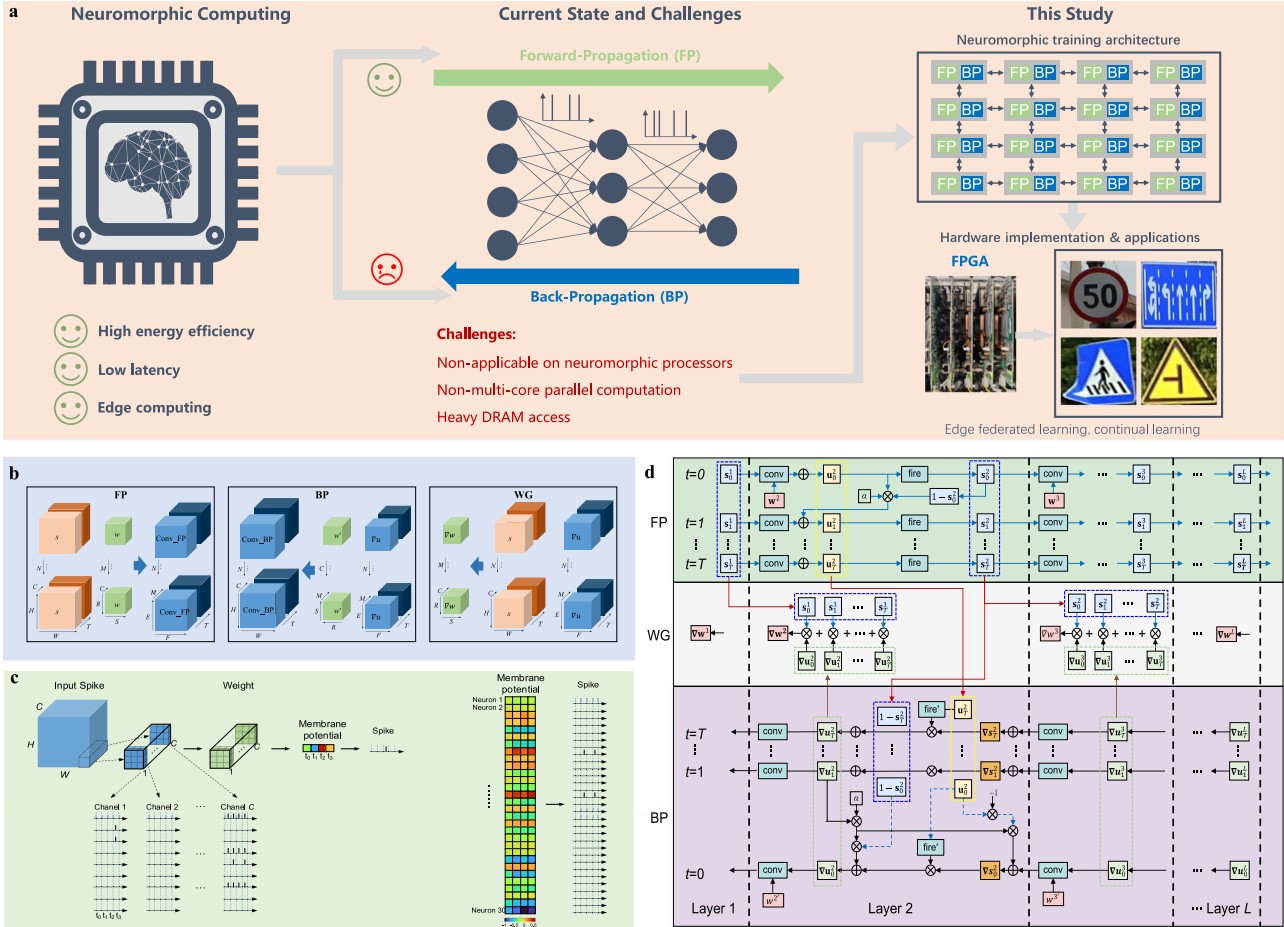

**Fig. 1 | Neuromorphic computing and SNN. a** An overview of neuromorphic computing, highlighting its advantages, challenges, and our proposed solutions. **b** Convolution computations in Feedforward-Propagation (FP), Back-Propagation (BP) and Weight Gradient (WG) processes in SNN training. The SNN parameters $N$, $T$, $H$, $W$, $C$, $R$, $S$, $M$, $E$, $F$ are described in Supplementary Table 2. **c** Membrane potential updating and spike generation in LIF neurons: an example LIF neuron in a convolution layer of SNN (left) and 30 LIF neurons in a convolution layer (right). **d** Computation and data transmission in the FP, BP, and WG processes. Full illustration is shown for layer 2, and detailed computations of Soma and Grad are shown for time step 0.

during backpropagation in parallel. The two engines share the input of $\nabla\mathbf{u}$ backpropagated from the next layer. The BP Engine contains BP Array, Grad, Pooling, and related SRAM. The BP Array is a 16×16 MAC array with FP16 precision, calculates products of $\nabla\mathbf{u}$ and kernel-rotated weight $\mathbf{w}'$ and accumulates these products across the $M$, $S$, $R$, and $T$ dimensions to generate 16 $C$ dimension convolution results **Conv_BP**. The Grad module (Fig. 2d) calculates the spike gradients $\nabla\mathbf{s}$, the surrogate gradient **fire**$'$, and $\nabla\mathbf{u}$ according to Eqs. (6), (5), (4) in the Methods, respectively. A BP engine includes 16 Grad modules, corresponding to 16 $C$ dimension membrane potential gradients $\nabla\mathbf{u}$. The WG engine contains a WG Array and related SRAM. The WG Array consists of a 16×16 adder array and selectors with FP16 precision, performing accumulation of $\nabla\mathbf{u}$ across the $T$ and $N$ dimensions to generate weight gradients $\nabla\mathbf{w}$. Its input includes output spike signals $\mathbf{s}$ obtained during forward computation and $\nabla\mathbf{u}$. Only the adders in the column of WG array with input spike value of 1 perform addition on $\nabla\mathbf{u}$. The WG engine supports the weight decay mechanism[42] to alleviate overfitting during training.

For intra- and inter-core communication, our multi-core architecture employs a 4×8 2D-Mesh topology (Fig. 2b) using the XY routing strategy, with each node comprising a router and a computing core (Fig. 2b). Each router features six channels (Fig. 2e): four for east, west, north, and south inter-node communication, and two dedicated channels for the FP sub-core and BP sub-core within the computing core. We designed a unified packet format for both bit-based spiking

and floating-point data (Fig. 2h) and modules such as Dispatch Unit (DU) (Fig. 2f) to support multiple types of data transmission. To support temporal dependency across cores, data of a single time step is organized as a message (Fig. 2h) to avoid mixing data from different time steps. The message is split into packets. The DU module serves as a data distribution hub for data transmission with SRAMs in sub-cores (see details in Methods 4.2B). The Controller unit in each sub-core monitors its working status and coordinates the orderly operation of modules in the sub-core. We implemented clock domain isolation (Fig. 2g) to decouple the working frequency of Routers and computing cores, with the computing core operating at 500 MHz and the Routers operating at a higher frequency of 667 MHz in the RTL simulation, achieving a higher data transmission bandwidth of 170 Gbps. This Globally Asynchronous Locally Synchronous (GALS) communication mechanism also avoids clock skew issues in global clock tree designs and improves scalability of our multi-core architecture. The 32-core neuromorphic architecture achieves 16 TFLOPS @ FP16 @ 500 MHz and contains 64.78MB SRAM (Supplementary Table 3). To support SNN training, the multi-core architecture includes three primary on-chip data transmission channels: 1) FP-FP transmission transferring the spike output $\mathbf{s}$ of a layer to the FP sub-core processing the next layer; 2) BP-BP transmission transferring the output $\nabla\mathbf{u}$ of a layer to the BP sub-core processing the previous layer; 3) FP-BP transmission transferring the spike $\mathbf{s}$ and membrane potential $\mathbf{u}$ obtained in forward computation to BP sub-cores through the path within the same

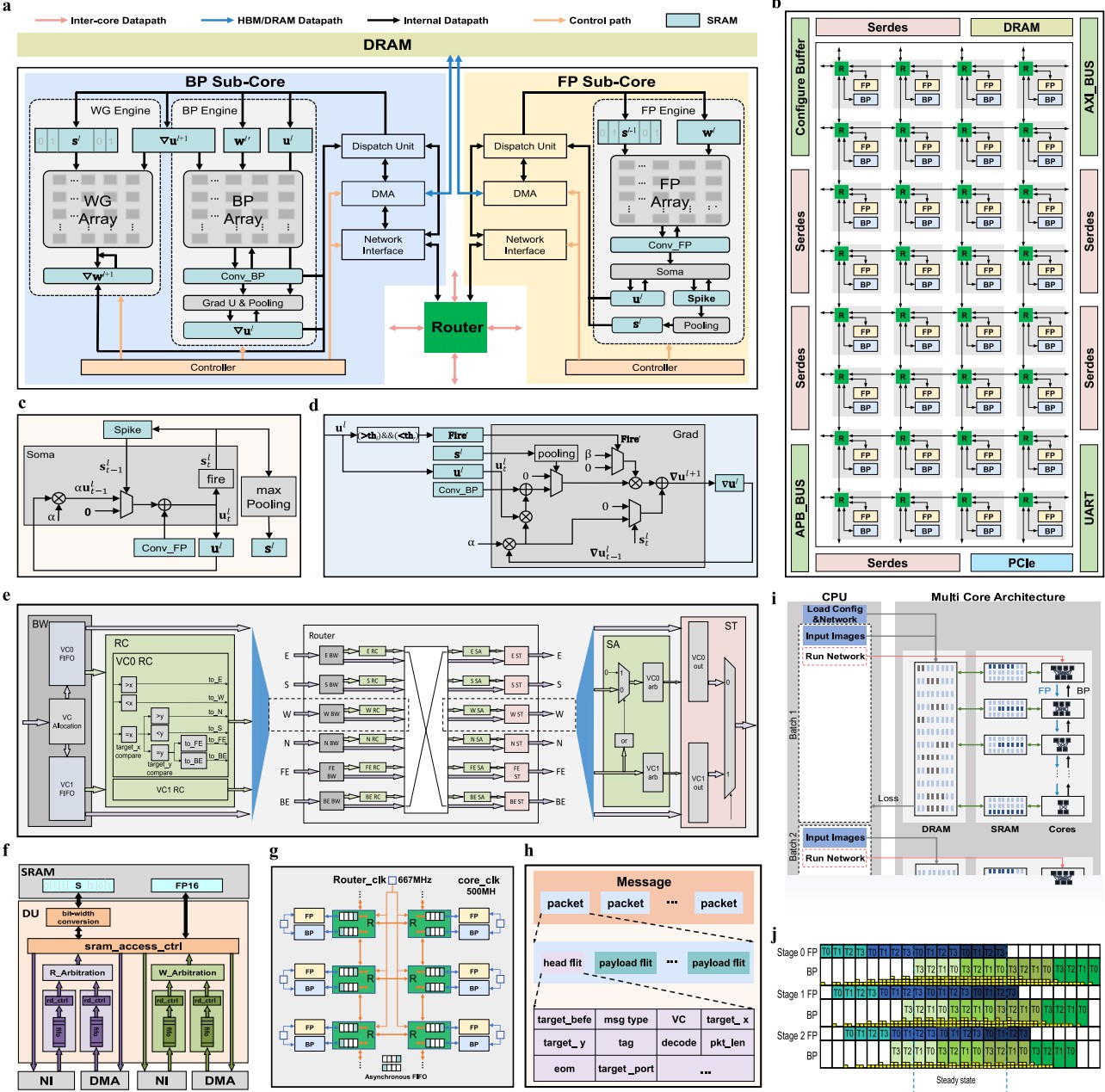

**Fig. 2 | The multi-core neuromorphic architecture for SNN training.**
**a** Architecture of a single computing core with an FP sub-core and a BP sub-core. "DMA" refers to Direct Memory Access. **b** The architecture with a 4×8 2D-Mesh topology. **c** Soma module in FP engine. **d** Grad module in BP engine. **e** Router (middle) with its associated BW (Buffer Write) and RC (Route Computation) modules (left), SA (Switch Allocation) and ST (Switch Traversal) modules (right). "E", "S", "W", "N", "FE", and "BE" refer to the east, south, west, north, forward-engine, and backward-engine channels, respectively. "arb" refers to arbitration. "VC" refers to Virtual Channel. **f** The Dispatch Unit (DU) module. "NI" refers to Network Interface. **g** The Globally Asynchronous Locally Synchronous (GALS) design. Clock frequency for orange lines: 667 MHz; for blue lines: 500 MHz. "R" refers to router. **h** The structure of messages, packets, and head flits. **i** Procedure of SNN deployment and parallel forward and backward computation in training. **j** Distributed training procedure in the multi-core neuromorphic architecture.

computing core: source FP sub-core SRAM→DU→NI → Router→ NI→DU → target BP sub-core SRAM (Fig. 2a). The inter-core communications support the FP-FP (or BP-BP) transmission through the paths: source FP (or BP) sub-core SRAM → DU → NI → Router → target core Router→ NI→DU → target FP (or BP) sub-core SRAM in our architecture (Fig. 2a and Supplementary Fig. 1a). Coordination of modules in these data transmission pathways ensures accurate intra- and inter-core communication spatially and temporally across model layers and time steps in both sequent and reverse orders in the distributed SNN training (Fig. 2j).

In contrast to the training procedure on GPUs, where only a portion of a model is deployed at a time, requiring multiple times of instruction transmission from CPU to perform the training on one batch of samples (Supplementary Fig. 3), the entire SNN is deployed on our multi-core architecture with different layers allocated on different computing cores, requiring only once of instruction transmission from CPU to perform training on one batch of samples (Fig. 2i). Training loss is used for the CPU to monitor the training process. Our training procedure facilitates efficient data transfer and reuse across computing cores, reducing DRAM access during training.

**Algorithm 1.** Parallelized SNN Training Operations in the Multi-core Neuromor-phic Architecture

```
1: N, L;                    % N: batch size,  L: SNN layer number
2: l∈{0,•••, L-1};          % Layer index
3: T ;                      % Total number of time steps
4: Core_Num = 32;           % Total number of Compute cores in
                              the architecture
5: Engine_Num=3;            % Total number of computing engines
                              in a core
6: n=zero(1, Core_Num);     %Indices of samples processed in FP
                              engines
7: n_bp= zero(1, Core Num); %Indices of samples processed in BP
                              engines
8: n_wg = zero(1,Core Num); %Indices of samples processed in WG
                              engines
9: Parallel for k= 0: Core_Num-1  do
10:    Parallel for e_i= 0: Engine_Num-1  do
11:      if e_i==0   %    computation in the FP engine
12:         Read w_k^l from SRAM;
13:         for t=0: T-1  do
14:            Read Spike inputs s_{n[k],t}^{l-1} from SRAM;
15:            Compute FP convolution Conv_FP_{n[k],t^{[i]}}^l;
16:            Compute potential u_{n[k],t^{[i]}}^l;
17:            Compute spike outputs s_{n[k],t^{[i]}}^l;
18:            Send the spike output to engines for following FP
                 computations;
19:            Send the potential and spike to engines for following
                 BP/WG computations;
20:         endfor
21:         if l == L − 1& t == T − 1 calculate loss;
22:         Update sample index n[k]; Goto line 13 to process the
                 next sample.
23:      if e_i==1:  % computation in the BP engine
24:         Read w_k^{l+1} from SRAM;
25:         for t=T-1: 0  do
26:            Read potential gradients ∇u_{n_bp[k],t}^l from SRAM;
27:            Read inputs S_{n_bp[k],t}^l, u_{n_bp[k],t}^{l+1} from SRAM;
28:            Compute potential gradient   ∇u_{n_bp[k],t^{[i]}}^l;
29:            Send the potential gradient to engines for following
                 BP/WG computation;
30:         endfor
31:         Update sample index n_bp[k]; Goto line 25 to process the
                 next sample.
32:      if e_i==2:  %Computation in the WG engine
33:         for t=T-1: 0  do
34:            Read potential gradients ∇u_{n_wg[k],t}^{l+1} from SRAM;
35:            Read spike inputs S_{n_wg[k],t}^l from SRAM;
36:            Compute gradients ∇w^{l+1}[i,j];
37:         endfor
38:         Compute weight gradients ∇w^{l+1};
39:         Update sample index n_wg[k]; Goto line 33 to process
                 the next sample;
40:            Update weight when finishing a batch.
41:    endfor
42: endfor
```

## Data flow and parallel FP-BP-WG computations

During SNN training, an SNN model is deployed in a distributed manner across all computing cores of our multi-core architecture, with one layer or a segment of a layer allocated to an engine within a computing core. Algorithm 1 shows our strategy of parallelized operations in our multi-core architecture during SNN training with inter-core pipeline parallelism and intra-core FP, BP, WG engine parallelism. At the core level, our multi-core architecture performs parallel feedforward and backward operations on different model layers, processing different samples simultaneously. Within the same core, FP and BP engines compute spike outputs and membrane potential gradients of the same model layer on different samples with opposite time-step orders simultaneously, and the WG engine computes weight gradients of the next layer in a reverse order in parallel with the computations in the FP and BP engines. Thus, our architecture supports highly parallel computing at both the core and engine levels to accelerate SNN training (Supplementary Algorithm 1 for more details).

We designed multiple data flows for FP, BP, and WG computations to enhance data reuse and reduce energy consumption of data transmission during training. The FP and BP engines adopt Weight Stationary (WS) data flow and partial sum SRAM reuse (Fig. 3a, b). During computation, weights **w** are retained in the PE arrays, while input **s** during FP and **s** during BP of different spatial positions and time steps are fed into the PE arrays. All weights are read only once from SRAM during the whole convolution computation. The WG engine adopts an Output Stationary (OS) data flow with products of ∇**u** and **w**′ retained and accumulated in WG array (Fig. 3c). We also designed storage reuse of **Conv_FP** and Soma in FP engine, **Conv_BP** and Grad in BP engine, and ∇**w** in WG engine (Fig. 3a–c).

Figure 3d–f show the high utilization of FP, BP, and WG engines during Spiking ResNet-18, −50, and spiking VGG-16 training in the software simulations (see Methods), supporting the highly parallel computations of FP, BP, and WG engines within a computing core. The three SNNs were selected based on their representativeness (Supplementary Table 12) and adoption in previous neuromorphic architecture studies[36]. In the software simulations, our 32-core architecture achieved the average performance (frames per second, fps) of 233.8, 115.8, and 44.8 during training of Spiking ResNet-18, −50, and spiking VGG-16, respectively. It achieved 18.20%, 15.88%, 17.47%, 17.14% of A100 GPU performance during Spiking ResNet-18 training (Fig. 3m) with batchsize 16, 32, 64, 128, respectively, and 40.11%, 39.48%, 38.46%, 38.08% of A100 during Spiking ResNet−50 training (Fig. 3n), and 18.69%, 19.08%, 19.52%, 23.64% of A100 during Spiking VGG-16 training (Fig. 3o), although our architecture consisted only 5.13% of A100 computing capability. The performance of our architecture was 190%, 330%, and 210% of that of the NVIDIA Jetson AGX Orin, respectively (Fig. 3m–o), indicating the high performance of our architecture over popular edge AI systems in SNN training. For the convergence speed of SNN training in our architecture, fine-tuning Spiking ResNet-18 on newly emerged CIFAR-10 samples required minutes, while training from scratch on CIFAR-10 required 2.5 hours (Supplementary Table 11). Fine-tuning a shallow SNN on the traffic sign classification task required less than 10 seconds.

The dataflow designs to enhance data reuse, together with single-bit storage of spike data, reduced the DRAM occupancy and access. We analyzed DRAM occupancy and accesses of our 32-core architecture and A100 GPU during Spiking ResNet-18, −50, and spiking VGG-16 training. Compared with A100, the DRAM occupancy of our architecture was reduced by 41.57%, 50.31%, 55.85%, and 59.00% during Spiking ResNet-18 training of batchsize 16, 32, 64, and 128 (Fig. 3g), respectively, by 44.62%, 52.69%, 57.65%, and 60.42% during Spiking ResNet−50 training (Fig. 3h), and by 59.83%, 67.83%, 72.53%, and 75.10% during Spiking VGG-16 training (Fig. 3i). The above reduction was mainly due to the decrease in the occupancy of intermediate data (Fig. 3r–t) accounting for 72.14% ~ 80.08% of the total DRAM occupancy of the A100 GPU training with the batch size of 16, and training in our architecture reduced this part of occupancy by 62.43% ~ 77.89%. Across batch sizes 16, 32, 64, and 128, the intermediate data accounted for 72.04% ~ 97.03% of the A100 DRAM occupancy, and training in our architecture reduced this part of occupancy by 62.43% ~ 77.89% (Supplementary Table 4). DRAM access in our architecture was reduced by 56.65%, 57.54%, 57.71%, and 57.53% during Spiking ResNet-18 training

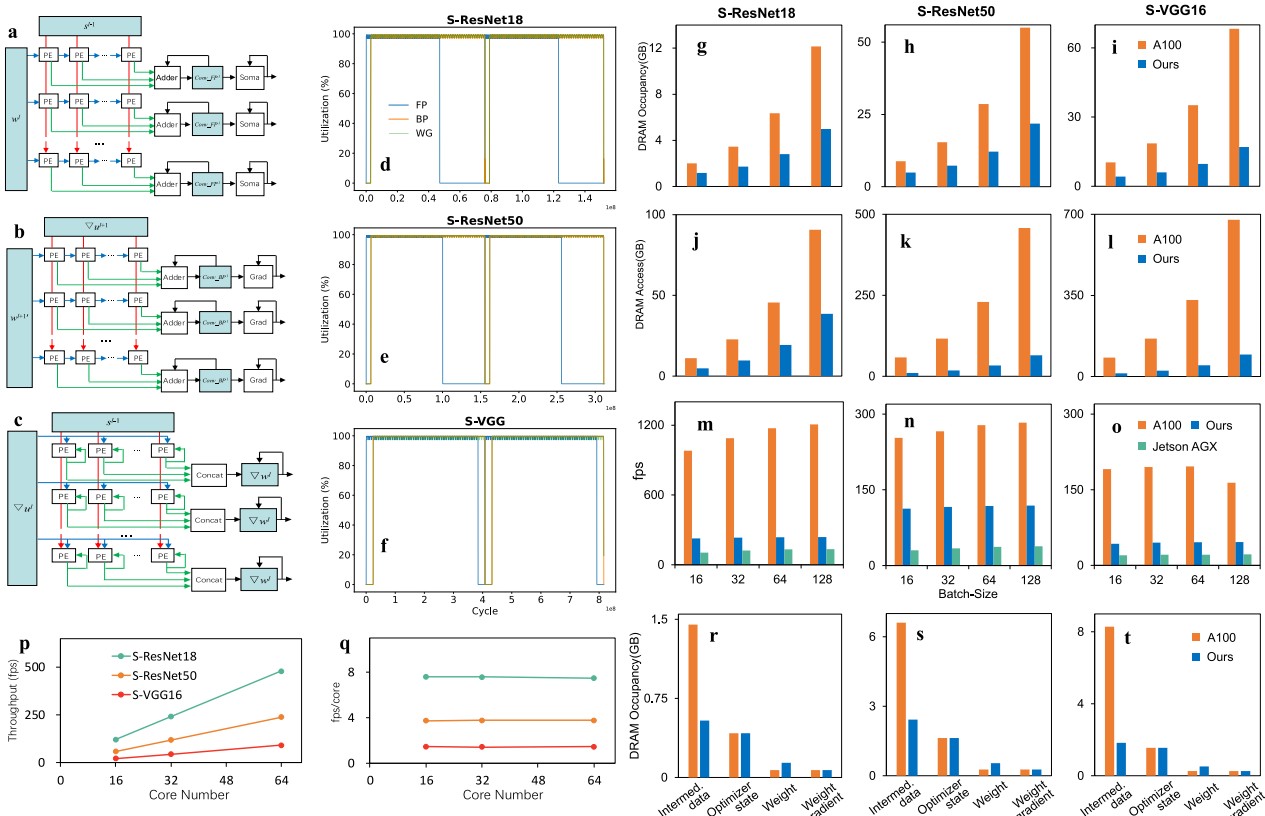

**Fig. 3 | Dataflow and parallel computations in the multi-core neuromorphic architecture. a–c** Dataflows of FP, BP, and WG engines, respectively. **d–f** The utilization of FP, BP, WG engines and computing core during Spiking ResNet-18, -50, and spiking VGG-16 training, respectively. Utilization curves during two batches of training period are plotted. **g–i** The comparison of DRAM/HBM occupancy between our multi-core architecture and A100 GPU during Spiking ResNet-18, −50, and spiking VGG-16, respectively. **j–l** The comparison of DRAM access during the three training, respectively. **m–o** The comparison of training performance (fps) during the three training, respectively. **r–t** The comparison of DRAM occupancy consisted of intermediate data, optimizer state, weight, and weight gradient. Batch size: 16.

(Fig. 3j), by 82.53%, 84.57%, 85.41%, and 85.89% during Spiking ResNet −50 training (Fig. 3k), and by 82.93%, 84.41%, 85.28%, and 85.97% during Spiking VGG-16 training (Fig. 3l). As the model parameters and batchsize increased, our architecture was more effective in reducing DRAM occupancy and access. The on-chip SRAM size of our 32-core architecture (64.78MB) was smaller than the SRAM size of A100 (88.75MB, NVIDIA A100 White book). Our multi-core architecture showed high scalability. Its throughputs (fps) increased almost linearly with the number of cores (Fig. 3p), with the slope (fps/core) metric holding steady with the number of cores during these trainings (Fig. 3q). It also supported inter-chip extension in 4 directions for chip-level scalability (Supplementary Fig. 1b).

## Sparse computation optimization

We designed sparse control circuits for the three engines. Figure 4a shows the design for a single output channel of the FP engine, which leverages the spike sparsity to control the first-stage adders (gating1) to perform addition only on the weights of the channels exhibiting spike activity. When all 16 channels of **s** are zero (spike16 sparsity), the entire adder tree computation and partial-sum update are skipped through the gating1 and gating2 signals. Figure 4b shows the design of a single output channel of the BP engine. When **fire**′ is zero, the entire multiplier array and adder tree computation, **∇u** reading, and partial-sum update are skipped through the gating1 signal. The gating2 signal and gating1 signal go through an AND operator to generate the gating signals for 16 multipliers and control signals for selectors. When the **∇u** input of a multiplier is zero, its computation is skipped and its output is

selected as zero. In the WG engine (Fig. 4c), when a channel of **s** is zero, the corresponding column of 16 accumulators skip accumulation through gating1 signal. When all 16 channels of **s** are zero, **∇u** reading is also skipped through the gating2 signal. Together, we leveraged the sparsity of **s**, **fire**′, and **∇u** signals in our design to reduce redundant computation and SRAM access. The RTL power analysis showed that our sparse design could significantly reduce the energy consumption of the computing core during Spiking ResNet-18, −50, and spiking VGG-9 training by61.09%, 46.81%, and 51.69%, respectively (Fig. 4d–f), including 42.53%, 27.01%, 37.11% of reductions in the FP engine leveraged on the spike and spike16 sparsity of these models (Supplementary Table 5), 67.74%, 53.92%, 55.75% of reductions in the BP engine leveraged on the **fire**′, and **∇u** sparsity (Supplementary Table 5), and 60.16%, 46.24%, 56.52% of reductions in the WG engine leveraged on the spike, spike16, and **∇u** sparsity (Supplementary Table 5). Supplementary Table 6 provides detailed energy consumption information during SNN training with and without sparse computation optimization.

## Area and power consumption analysis

All reported data is based on TSMC N28HPC process under the tt_0.9V_25°C condition. During the performance evaluation, the engine and router clocks reach 500 MHz and 667 MHz, respectively (Supplementary Fig. 8). PVT (process, voltage, temperature) variation (Supplementary Table 9) and timing closure analysis (Supplementary Fig. 9) support the stability and reliability of our multi-core architecture design under different conditions. The area and power consumption of a single computing core were estimated in RTL

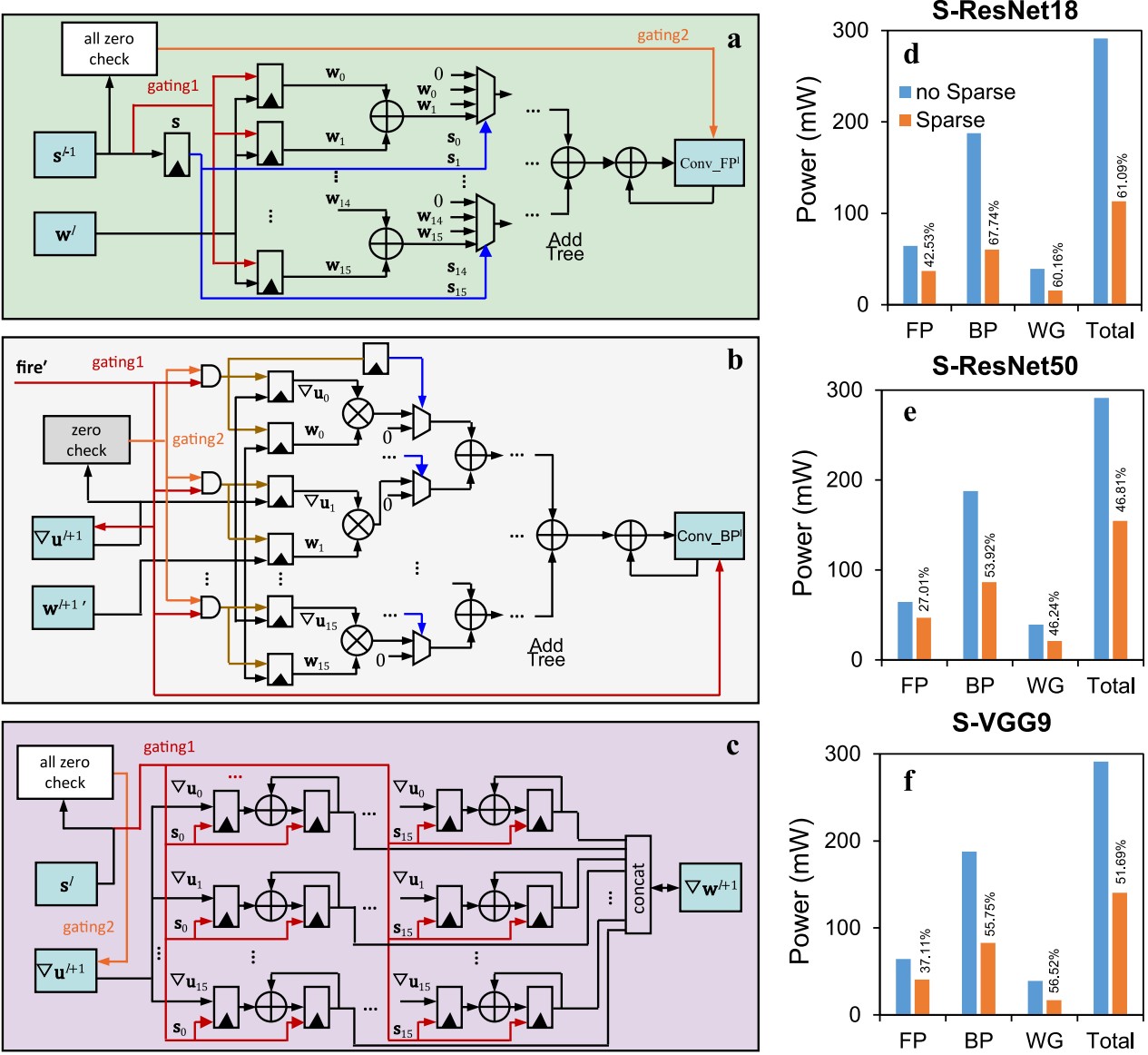

**Fig. 4 | Sparse computation optimization. a–c** Sparse computation optimization circuit designs for the FP engine, BP engine, and WG engine, respectively. The red and orange lines serve as gating signals of computation units and SRAM accesses, and blue lines as control signals for Multiplexers. **d–f** Comparisons of the energy consumption before and after sparse optimization for Spiking ResNet-18, −50, and spiking VGG-9 training, respectively. The percentage values near bins indicate the energy consumption reduction by the sparse computation optimization.

simulation. The BP, FP, and WG Engine together accounted for 78% of the total area and 63% of the total power consumption, while the rest modules including Router, NI, DMA (Direct Memory Access), DU, and Controller, together accounted for 22% of the total area and 37% of the total power consumption (Supplementary Fig. 4a). Grouped into computation, storage, and transmission, the on-chip storage occupied the largest area of the computing core (67%), while the power consumption of the three groups was comparable (30~38%; Supplementary Fig. 4b). The code coverage scores in the RTL functional verification were higher than 95% (Supplementary Fig. 5).

Table 1 shows the performance comparison between our 32-core near-memory neuromorphic architecture and representative architectures such as neuromorphic chips and GPUs. The computation capacity of our 32-core architecture (16 TFLOPS @ FP16) was clearly higher than that of prior neuromorphic architectures (typically ~≤1TOPS with computational precisions 1-9 bit). Notably, only our

multi-core architecture supported BP-based SNN training, while prior multi-core neuromorphic architectures only supported inference and local learning. A recent study proposed SATA[43], which was only a single-core SNN training architecture and lacked hardware implementation. Compared with general AI architectures such as GPUs, a commonality was that both can support SNN training. Our architecture achieved an energy efficiency of 1.05 TFLOPS/W @ FP16 @ 28 nm, which was comparable to the A100, DOJO D1, and Graphcore with 7 nm technology, demonstrating the high energy efficiency of our architecture.

## FPGA implementation and applications
The multi-core near-memory neuromorphic architecture was deployed on an FPGA platform (Fig. 5a) to perform SNN training through our software toolchain (Fig. 5c-d). Bidirectional conversion between our custom IR and general-purpose open-source Neuromorphic IR (NIR, Supplementary Fig. 10) is developed to improve the

**Table 1 | Comparison between our work and representative architectures**

| Items | Neuromorphic Architecture | | | | | | | General AI Architecture | | | | This Work |
|---|---|---|---|---|---|---|---|---|---|---|---|---|
| | Loihi | Loihi2 | TianjiC | True North | Brain Scale2 | Darwin3 | SATA | Jetson AGX Orin 64GB | Graph core | DOJO | GPU A100 | |
| Process | 14 nm | 7 nm | 28 nm | 28 nm | 65 nm | 22 nm | 65 nm | 8 nm | 7 nm | 7 nm | 7 nm | 28 nm |
| Network | SNN | SNN | ANN/ SNN | SNN | SNN | SNN | SNN | ANN/ SNN | ANN/ SNN | ANN | ANN/ SNN | SNN |
| Frequency (MHz) | Async | Async | 300 | Async | – | 333 | 500 | 1300 | 1850 | 2000 | 1410 | 500 |
| On-chip Memory (MB) | 32 | 24 | 14.44 | 260 | 0.08 | – | 4 | – | 368 | 440 | 88.38 | 64.78 |
| Topology | 8×16 | 8×16 | 12×13 | 64×64 | 2×2 | 24×24 | – | – | – | 18×20 | – | 4×8 |
| Area (mm2) | 60 | 31 | 14.44 | 430 | 65 | 358.53 | – | – | 832 | 645 | 826 | 218.63 |
| Power (W) | 0.11 | – | 0.95 | 0.065 | 0.2 | 1.8 | – | 15–60 | 300 | 400 | 400 | 14.49 |
| Peak Performance | 0.03 TOPS | – | 1.214 TOPS | 0.058 TSOPS | – | – | – | 85 (Sparse) TFLOPS | 250 TFLOPS | 362 TFLOPS | 312 TFLOPS | 16 TFLOPS |
| Precision | 1–9 bit | 1–9 bit | 8 bit | 1 bit | 6 bit | 1/2/4/8/ 16 bit | 8 bit | INT8/ FP16 | FP/BF16 | FP16 | FP/BF16 | FP16 |
| Energy Efficiency | 0.27 TOPS/ W | – | 1.278 TOPS/ W | 0.89 TSOPS/ W | – | – | – | – | 0.83 TFLOPS/ W | 0.91 TFLOPS/ W | 0.78 TFLOPS/ W | 1.05 TFLOPS/ W |
| On-chip Learning | Two-factor Local Learning | Three-factor Local Learning | No | No | Two-factor Local Learning | Three-factor Local Learning | BP | BP | BP | BP | BP | BP |

cross-platform compatibility of the toolchain. The detailed hardware utilization is listed in Supplementary Table 7. Figure 5e shows that a 20-core neuromorphic architecture was deployed on five FPGA cards interconnected with 2×10 2D Mesh topology. We developed an Ethernet Adapter module (Fig. 5b; see details in Methods) to support inter-FPGA data transmission among any cores deployed on different FPGAs through the XY routing strategy. Training from scratch of spiking ResNet-18 deployed on the 20-core architecture converged normally in image classification tasks on MNIST, CIFAR-10, and CIFAR-100 datasets (Fig. 5f, g), with accuracies 99.23%, 85.5%, and 63.89%, respectively. The accuracy loss compared to GPU was 0.05%, 0.49%, and 2.41%. It required increased training epochs to obtain stable classification accuracy with the increased complexity of the datasets, from nearly 1 epoch for MNIST, 30 epochs for CIFAR-10, to 70 epochs for CIFAR-100 (Fig. 5f, g). For CIFAR-100, our implemented weight decay in the WG engine improves the accuracy on unseen data by 1.30% (from 62.59% to 63.89%). We then implemented continual learning (Fig. 5i) and federated learning (Fig. 5m) applications in the FPGAs for a smart traffic scenario (Fig. 5h). In continual learning, pre-trained SNN models were fine-tuned based on BP with samples of new classes. The continual learning improved inference accuracy on mixed new and old data by 17.97% (from 75.01% to 92.98%) for the Traffic Sign Classification and Recognition dataset (Fig. 5j, orange line vs blue line), 33.21% (from 58.09% to 91.30%) for the Vehicle and Pedestrian Detection dataset (Figs. 5k), and 42.41% (from 43.06% to 85.47%) for the DVS-Gesture dataset (Fig. 5l). In the 5-worker federated learning, the federated model aggregated gradient information generated separately by each worker based on its local dataset. Figure 5o–r shows that the federated model (red line) significantly outperformed worker models (gray lines), achieving accuracies of 94.75% and 84.03% on the Traffic Sign Classification and Recognition dataset and the DVS-Gesture dataset, respectively.

## Discussion

To the best of our knowledge, we developed the first multi-core neuromorphic architecture enabling BP-based training of deep SNN

models in a distributed manner across computing cores. The proposed computing cores with three PE-array based engines execute diverse computational forms in SNN training. Communication mechanisms featuring the six-way router, three primary SNN training-oriented inter-core/engine pathways, and a unified package format ensure efficient transmission of multi-type data between intra- and inter-core engines. Previous neuromorphic computing approaches rely exclusively on GPUs[39,43,44]; or implement a hybrid training loop, combining inference on the neuromorphic platforms with BP on GPUs[32,45]; Recently, Renner[46] combined a gating mechanism with Hebbian learning to enable BP for a single hidden-layer fully connected network on Loihi, while its applicability to popular SNNs, such as convolutional SNNs, remains a question. Additionally, the proposed training paradigm consumes over 200 times the energy of inference, raising concerns about its training efficiency. Our architecture also shows high scalability with its 2D-mesh topology and the GALS design, exhibiting a near-linear increase of throughput with core number. Various scales of the architecture have been implemented successfully in FPGAs for supervised learning and federated learning, indicating its high flexibility in scale. The computation capacity of our 32-core architecture is clearly higher than that of prior neuromorphic architectures. It achieves a high energy efficiency of 1.05 TFLOPS/W @ 28 nm, which is comparable to that of A100, DOJO, and Graphcore with 7 nm technology.

Our architecture supports highly efficient parallel computing at both the engine and core levels in deep SNN training, a capability absent in previous neuromorphic architectures. Leveraging on our multi-level parallel computing strategy with inter-core pipeline parallelism and intra-core FP, BP, WG engine parallelism, the architecture achieves 15 ~ 40% performance (fps) of an NVIDIA A100 GPU with only ~5% of A100 computing capability during SNN training, and 190% ~ 330% performance of an NVIDIA Jetson AGX Orin. The convergence speed of SNN training in our architecture indicates its real-time learning potential in fine-tuning conditions. To tackle the challenge of real-time adaptation, it relies on both the hardware architecture and the algorithmic optimization.

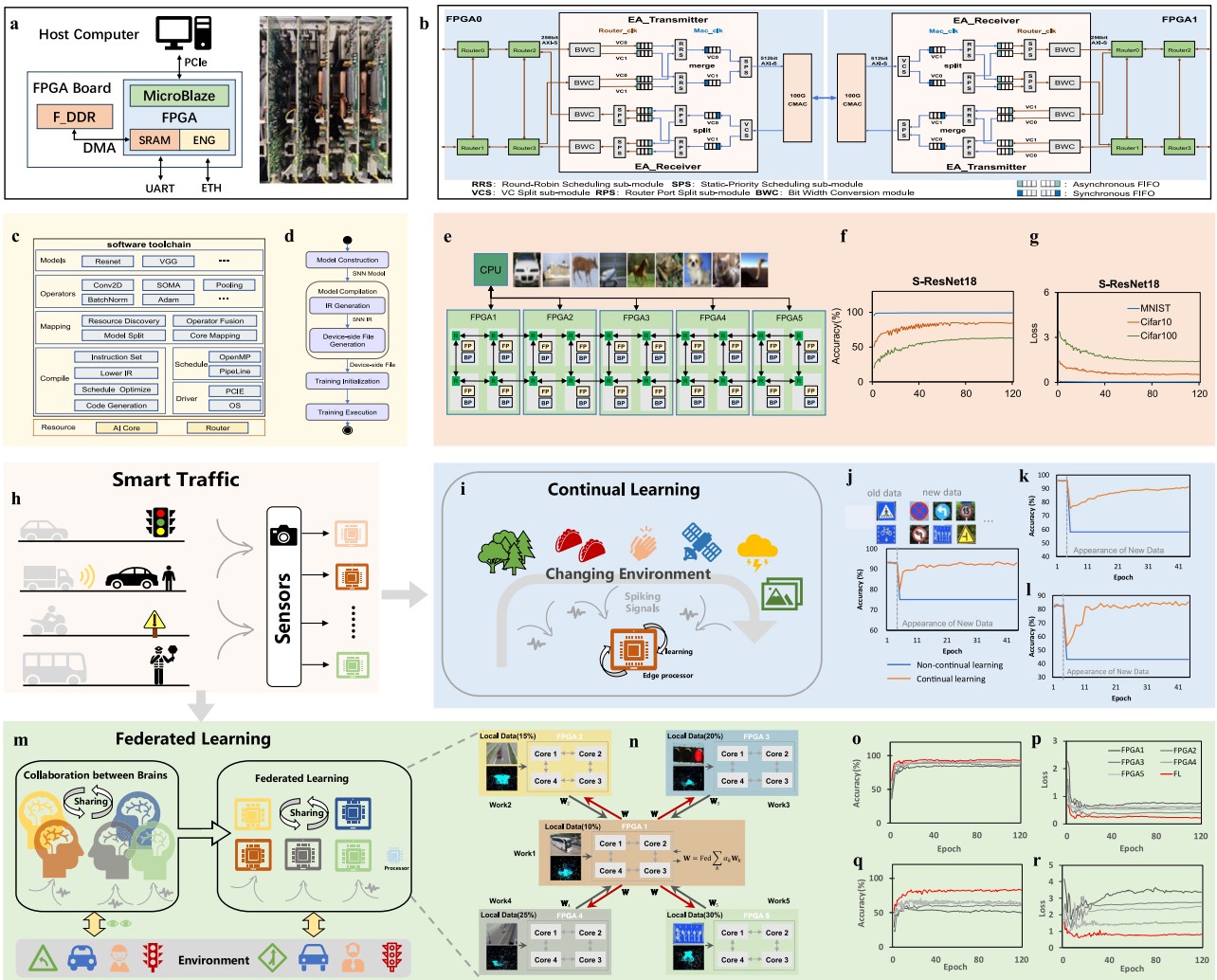

**Fig. 5 | FPGA implementation of the multi-core neuromorphic architecture.**
**a** FPGA platform. **b** Inter-FPGA communication. **c** Software toolchain for our multi-core architecture. **d** Steps of SNN construction, deployment, and training execution. **e** Implementation of the 20-core neuromorphic architecture for SNN training based on BP on FPGAs for image classification. **f**, **g** Accuracy and loss curves of SNN training on FPGAs in the image classification tasks. **h** Smart city scenario. **i** Illustration of the continual learning procedure on an edge processor. **j**–**l** Accuracy curves of the continual learning and non-continual learning models on the Traffic Sign Classification and Recognition dataset, Vehicle and Pedestrian Detection dataset, and DVS-Gesture dataset, respectively. **m** Illustration of federated learning procedure on an edge processor. **n** Implementation of the 5-worker federated learning based on FPGAs, with each worker implemented on one FPGA, and worker1 also used as the server. **o**, **p** Accuracy and loss curves of the five workers and the federated model on the Traffic Sign Classification and Recognition dataset. **q**, **r** Accuracy and loss curves of the five workers and the federated model on the DVS128 Gesture dataset.

Compared to prior studies (e.g., Loihi, TrueNorth) that focused exclusively on spike sparsity, our architecture fully leveraged the sparsity of spikes, **fire**′, and **∇u** in SNN training, achieving 45% - 60% reduction of energy consumption. Through single-bit spike storage, multiple dataflows within computing engines, SNN training in our architecture achieved 40 - 75% reduction of DRAM occupancy and 55% - 85% of DRAM access compared to GPU baselines, which not only significantly alleviated the DRAM storage requirements, facilitating the training in storage constrained conditions, but also improved the energy efficiency as energy consumption of DRAM access is an order higher than that of on-chip SRAM[47]. DRAM usage optimization has not been the focus in previous neuromorphic architectures designed for inference and local learning.

We also developed a software toolchain for our architecture, allowing construction of various deep SNN models and automatic construction of hardware-friendly computing graphs depending on various SNN tasks. While GPU-based training typically requires layer-by-layer control from the host, the toolchain enables hierarchical parallelism for SNN training with one-time deployment of the entire

model in our multi-core architecture, reducing control overhead significantly. The bidirectional conversion between our custom IR in our software toolchain and NIR, a recent general-purpose IR for neuromorphic computing[48], renders our software toolchain compatible with multiple hardware platforms.

Our multi-core architecture has been assessed in FPGA implementation, RTL functional verification including code coverage and time closure analysis, and PVT variation analysis, demonstrating the stability and reliability of our multi-core architecture design under different conditions. It still requires further assessment on taped-out chips, and the taped-out chip should achieve significantly higher performance in clock frequencies and data transmission bandwidth (Supplementary Table 10).

Together, our study extends multi-core neuromorphic computing architectures in global learning capability, multi-level parallel computing, efficient DRAM usage, and sparse computation optimization, facilitating their applications, such as those in edge-learnable environments.

## Methods

### SNN training algorithm

The SNN adopts the LIF neuron model, and the training process is divided into the following stages[40,41]:

**Forward propagation (FP) stage.** The membrane potential and output spike signal during the SNN's feedforward inference process are calculated. $\mathbf{u}_t^l[i]$ and $\mathbf{s}_t^l[i]$ represent the membrane potential and spike signal of neuron $i$ in layer $l$ and time step $t$, respectively. $\mathbf{u}_t^l[i]$ is determined by the state of the neuron at time step $t-1$ and the membrane potential change caused by its synaptic input, calculated as:

$$\mathbf{u}_t^l[i] = \alpha \mathbf{u}_{t-1}^l[i]\left(1 - \mathbf{s}_{t-1}^l[i]\right) + \mathbf{Conv\_FP}_t^l[i], \tag{1}$$

where $\mathbf{Conv\_FP}_t^l[i]$ is the convolution of the spike output signal of layer $l-1$ and the weight value of the current layer $l$:

$$\mathbf{Conv\_FP}_t^l[i] = \sum_j \mathbf{s}_t^{l-1}[i]\mathbf{w}^l[j,i], \tag{2}$$

where $\alpha$ is the leakage factor of the membrane potential. The membrane potential of the neuron is then compared with the threshold voltage to obtain its spike output signal $\mathbf{s}_t^l[i]$ through the spike activation function $\mathrm{fire}\left(\mathbf{u}_t^l[i]\right)$:

$$\mathbf{s}_t^l[i] = \mathrm{fire}\left(\mathbf{u}_t^l[i]\right) = \begin{cases} 1, & \mathbf{u}_t^l[i] \geq \mathrm{th}_f, \\ 0, & \text{else}. \end{cases} \tag{3}$$

When the membrane potential exceeds the threshold voltage, the neuron emits a spike and resets the membrane potential to zero.

**Back propagation (BP) stage.** The membrane potential gradient $\nabla\mathbf{u}_t^l[i]$ and spike gradient $\nabla\mathbf{s}_t^l[i]$ during the Back-Propagation process are calculated as follows:

$$\nabla\mathbf{u}_t^l[i] = \alpha\nabla\mathbf{u}_{t+1}^l\left(1 - \mathbf{s}_t^l[i]\right) + \nabla\mathbf{s}_t^l[i]\mathrm{fire}'\left(\mathbf{u}_t^l[i]\right), \tag{4}$$

where the $\mathrm{fire}'(\mathbf{u}_t^l[i])$ is

$$\mathrm{fire}'\left(\mathbf{u}_t^l[i]\right) = \begin{cases} 1, & th_l \leq \mathbf{u}_t^l[i] \leq \mathrm{th}_r, \\ 0, & \text{else}. \end{cases} \tag{5}$$

The spike gradient $\nabla\mathbf{s}_t^l[i]$ is calculated as:

$$\nabla\mathbf{s}_t^l[i] = -\alpha\nabla\mathbf{u}_{t+1}^l[i]\mathbf{u}_t^l[i] + \mathbf{Conv_{BP}}_t^l[i], \tag{6}$$

where $\mathbf{Conv\_BP}_t^l[i]$ is the convolution of the membrane potential gradient and the weight value of layer $l+1$:

$$\mathbf{Conv\_BP}_t^l[i] = \sum_j \nabla\mathbf{u}_t^{l+1}[j]\mathbf{w}^{l+1}[i,j]. \tag{7}$$

**WG Stage.** The weight gradient $\nabla\mathbf{w}^l[i,j]$ is calculated based on the membrane potential gradient of current layer and the spike output signal of layer $l-1$ obtained during the forward computation:

$$\nabla\mathbf{w}^l[i,j] = \sum_t \nabla\mathbf{u}_t^l[j]\mathbf{s}_t^{l-1}[i]. \tag{8}$$

Finally, the weights are updated using $\nabla\mathbf{w}^l[i,j]$.

### Multi-core neuromorphic architecture design for SNN training

**A. Multi-core neuromorphic architecture design.** We designed a multi-core near-memory neuromorphic architecture that supports FP, BP, and WG computations for deep SNN training with a computational precision of FP16 due to its common usage in SNN training

(Supplementary Table 12). The single computing core within the multi-core architecture includes FP Sub-Core, BP Sub-Core, and Router.

The FP Array in the FP Sub-Core consists of a 16×16 selector and adder array with FP16 precision. The 16 $C$ dimension input spike signals **s** are broadcasted to 16 columns of the FP Array, respectively. The FP Array performs parallel computations of 16 output channels along $M$ dimension vertically and accumulation across 16 different input channels along the $C$ dimension horizontally, and once outputs 16 channel partial-sums of weights **w** accumulated across the 16 $C$ dimensions, and further accumulates these weight partial-sums across the $R$, $S$, and $T$ dimensions to generate 16 $M$ dimension convolution results **Conv_FP** by refreshing the input spike signals **s** and weights **w**. Through the spike signals' control of selectors, only adders in the column of FP array with input spike value of 1 perform addition on weight values. For the Soma module, if there was no spike firing in the previous time step, the membrane potential at the previous time step is first multiplied by its leakage factor $\alpha$ and added to **Conv_FP** to get the membrane potential at current time step. Otherwise, the membrane potential at current time step is set to the **Conv_FP**.

The BP Array in the BP Sub-Core is a 16×16 MAC array with FP16 precision. Its input includes 16 membrane potential gradients $\nabla\mathbf{u}$ backpropagated from next layer and 256 kernel-rotated weights **w'**. The membrane potential gradients of 16 output channels along the $M$ dimension are broadcasted to the 16 columns of the BP Array, respectively. The BP Array performs parallel computations on 16 the input channels along the $C$ dimension vertically and accumulation across 16 $M$ dimension output channels horizontally, and once outputs 16 $C$ dimension channel partial-sums of products between $\nabla\mathbf{u}$ and **w'** accumulated across the 16 $M$ dimensions. By refreshing input membrane potential gradients $\nabla\mathbf{u}$ and weights **w'** multiple times, the BP Array accumulates the partial-sums across the $M$, $S$, $R$, and $T$ dimensions to obtain the convolution result **Conv_BP**.

The WG Array in the WG engine consists of a 16×16 adder array and selectors with FP16 precision. 16 $C$ dimension channels of input spikes $s$ are broadcasted to 16 columns of the WG Array, respectively. 16 $M$ dimension channels of membrane potential gradients $\nabla\mathbf{u}$ are broadcasted to 16 rows of the WG Array, respectively. Each column of the WG Array shares a selector controlled by the input spike signal. Thus, only the adders in the column of WG array with input spike value of 1 perform addition on membrane potential gradients. The WG Array first computes the weight gradients for one timestep, and accumulates the gradients of different time steps to obtain $\nabla\mathbf{w}$ according to Eq. (8).

**B. Intra- and inter-core communication in the multi-core architecture.** During SNN training, an SNN model is distributed across all computing cores in our multi-core architecture, with one layer or a segment of a layer allocated to an engine within a computing core. The communication mechanisms ensure accurate transmission of multiple types of data, such as bit-based spiking data, floating-point membrane potential and its gradients, along both model layers and time steps in both sequent and reverse orders during SNN training. Our intra- and inter-core communication design focuses on the following aspects:

**Unified packet design.** To support temporal dependency across cores, data of a single time step was organized as a message (Fig. 2h) to avoid mixing data from different time steps. The message was split into 2KB packets. Both floating-point and bit-based spiking data in SNN training are transferred with a unified packet format. Packets are transmitted in 32B flits, with each flit carrying 16 FP16 data or 256 single-bit spiking data. The detailed information in the packet header is shown in Supplementary Fig. 2d.

**2D-mesh topology and router design.** Our multi-core architecture employs a 4×8 2D-Mesh topology (Fig. 2b), with each node comprising a router (Fig. 2e) and a computing core (Fig. 2a). Each router features

six channels (Fig. 2e): four for east, west, north, and south inter-node communication, and two dedicated channels for the FP sub-core and BP sub-core within the computing core to increase the bandwidth of data transmission during FP and BP computations. Each router direction supports two virtual channels (VCs), with incoming data stored in corresponding FIFOs (First-In First-Out memories). The Route Computation (RC) unit reads the head flit from the FIFO, parses the head flit, and determines the next transmission direction using the XY routing strategy by comparing the target X and Y coordinate information (target_x, target_y) in the head flit with the router's coordinates (x, y). If the router is the target router during transmission, it uses information in the target_befe field of the head flit to select the FE or BE output port. When multiple packets from different directions request the same output port, the Switch Allocation (SA) module employs a round-robin arbitration policy to prioritize transmission. The Switch Traversal (ST) module transfers the packet flit-by-flit based on back-pressure from the downstream unit. The 2D-Mesh topology has been commonly adopted in previous multi-core neuromorphic architectures.

**Globally asynchronous locally synchronous (GALS) communication mechanism (Fig. 2g).** The architecture adopts a globally asynchronous design with separate clock frequencies for computing cores and routers: all routers share a common clock (Router-clk, 667 MHz), while each core has an independent clock (core-clk, 500 MHz). Asynchronous FIFOs are implemented at the router-core interface to support cross-clock domain data transfer. The GALS design avoids clock skew issues in global clock tree designs and improves the scalability of our architecture. The higher router clock frequency increases data transmission bandwidth to 170 Gbps/channel.

**Data transmission modules and pathways.** We also designed Dispatch Unit (DU, Fig. 2f; see Supplementary Fig. 2a-b for detailed DU in FP and BP sub-cores) and Network Interface (NI) module (Supplementary Fig. 2c) to support intra- and inter-core transmission for multiple types of data. The DU serves as a data distribution hub between SRAMs in sub-cores and NI or DMA. The NI Transmitter fetches data payloads from SRAMs via DU and encapsulates them into packets, and performs flow control of data transmission via the router. The NI Receiver performs decapsulation of packets, and manages their address mapping in SRAMs/DRAMs. Based on these modules, our multi-core architecture forms three primary data transmission pathways (Supplementary Fig. 1a) – inter-core FP-FP transmission, inter-core BP-BP transmission, and intra-core FP-BP transmission – to support SNN training in our multi-core architecture.

**C. Architecture evaluation based on software simulation.** To quantitatively evaluate the performance of the multi-core near-memory computing architecture during SNN training, we developed a software tool S-ZigZag based on ZigZag and SimST, with ZigZag a performance evaluation tool for computing architecture during ANN inference[49] and SimST an energy simulation framework for SNN-based models[50]. S-ZigZag supports the evaluation of energy consumption, throughput, and latency for different data flows during SNN training tasks under given computing architecture. All software tools are implemented under Python (version 3.8.17).

SNN BP convolution and ANN convolution both use floating-point convolution. Therefore, in S-ZigZag, we adopted ZigZag's convolution computation simulation module to simulate the BP convolution of one timestep in SNN, with multiple simulations for multiple time steps of BP convolution. For FP and WG convolution computations of SNN, we developed custom simulation modules to simulate convolution computation between bit-based spiking and floating-point weight data using selector and adder arrays. We developed modules for Soma and Grad computation simulation. Soma simulation includes selection, addition, multiplication, and comparison operations, while Grad simulation includes selection, addition, and multiplication operations. We also developed a NoC module to simulate the data transmission volume and energy consumption between computing cores.

In S-ZigZag, we adopted Json format for the description files. The computing architecture description file includes the type, count, and energy consumption information of computation parts in the three engines; capacity, bandwidth, datawidth, bank number, and read/write energy consumption information for DRAM, SRAM, and REG three-levels of storage. The model description file includes dimension information for all layers of SNN models. The data flow description file described the temporal and spatial mapping on the computation array through the for-loop form.

$P$ represents the number of cycles to train one batch of data, as follows:

$$P = P_{FP}^{1st} + \max(P_{BP}, P_{WG}) \tag{9}$$

where $P_{FP}^{1st}$ represents total cycle number of FP processing the first data of each batch, and $P_{BP}$ and $P_{WG}$ represent total cycle number of BP and WG processing a batch data.

The utilization rate of the computing core is:

$$\text{Util} = \frac{\sum_{i=1}^{P}\left(\text{Op}_{FP\_used}^i + \text{Op}_{BP\_used}^i + \text{Op}_{WG\_used}^i\right)}{P \times \text{Op}_{total}} \tag{10}$$

where $\text{Op}_{FP\_used}^i$, $\text{Op}_{BP\_used}^i$, and $\text{Op}_{WG\_used}^i$ represent the number of active multiplication and addition components in FP, BP, and WG during cycle $i$, respectively. $\text{Op}_{total}$ represents the total number of computing components of the three engines.

We compared our 32-core near-memory computing architecture with the NVIDIA A100 80GB GPU in DRAM/HBM memory usage and access during SNN training. Our 32-core computing architecture has a total of 64.78MB on-chip SRAM storage, while the A100 has 88.38MB. We compared the peak DRAM usage and total accesses of one batch training with the ImageNet2012 dataset. Batch sizes include 16, 32, 64, and 128, and trained models are spiking_ResNet18, _ResNet50, and _VGG16. On the A100 GPU, we implemented the BPTT-based SNN model using PyTorch[51] and obtained the peak HBM usage during SNN training through the API (https://pytorch.org/docs/2.0/generated/torch. cuda.max_memory_allocated.html). We used NVIDIA's Nsight Compute tool (https://developer.nvidia.com/nsight-compute) to collect volumes of HBM access through the dram_bytes_read.sum and dram_bytes_write.sum, and summed the statistics of all CUDA kernel functions to get the total access volume for training one batch. We also implemented the BPTT-based SNN training on the Jetson AGX Orin (64GB) using PyTorch[51] and obtained the processing speed (fps) during the training. SNN models were split into 32 parts and deployed them on 32 cores in a balanced manner to balance the total delay of each core during model training. We simulated training one batch, considering the on-chip SRAM size and the data volume required for intra-core computation. Using the S-ZigZag tool, we analyzed the storage locations and access counts of all variables during FP, BP, and WG processes to obtain the total data access volume and peak DRAM usage for training one batch.

**D. RTL coding and functional verification.** We developed Verilog code in the gvim editor to implement the multi-core near-memory computing architecture. The computing core was coded separately according to its modules, then these modules were packed together. Data flow and sparse optimization were also coded in the relevant modules. The total codes for a single computing core were about 30,000 lines. The codes were compiled and synthesized using Design Compiler (DC, Synopsys Inc), and formed an independent IP. To

implement the 32-core architecture, 32 copies of the IP were interconnected via an 8×4 2D-mesh NoC network.

We used the Verilog Compiled Simulator (VCS, Synopsys Inc) to perform multi-platform verification at single module, single computing core, and multi-core architecture levels, verifying the correctness of computation, data transmission, control functions, and software configuration. At the single module level, we designed test cases specific for each module including convolution operations with different kernel sizes and strides, and Soma calculations, partial sum accumulation for the FP/BP/WG engine modules, data transmission for DU, Router, NI, and DMA, and status detection and state jump for the control module. At the single computing core level, we designed test cases including multi-timestep forward and backward computations of a single SNN layer for the FP sub-core and BP sub-core, $s/u$ transmission across sub-cores, and multi-path data transmission for NoC and DMA. At the multi-core architecture level, SNN models were deployed layer by layer across different computing cores on the RTL simulation platform. The forward and backward RTL simulation outputs for all layers were compared with the SNN training simulation results (Golden values) based on the C language.

The effectiveness of the Weight Stationary (WS) data flow in FP and BP engines was verified by analyzing the SRAM accesses and PE retaining of weights during forward or backward computations. The weight value in the FP and BP engines was read from SRAM only once during convolution computation and retained in the PE. The effectiveness of the output stationary data flow of the WG engine was verified by analyzing the iterative accumulation values of products of $\nabla u$ and $s$ in each PE. The output results of all PEs remained in the PE, verifying the output stationary data flow of the WG engine. By counting the SRAM accesses of **Conv_FP** in FP engine, **Conv_BP** and $\nabla u$ in BP engine, and $\nabla u$ in WG engine, we verified the effectiveness of the sparse design of the three engines.

**E. Area and power consumption analysis.** We used DC and VCS (Synopsys Inc.) to evaluate the area and power consumption of our multi-core neuromorphic architecture. The computing core was divided into modules for analysis. We generated gate-level netlists and area reports based on the RTL code with TSMC 28 nm process.

Power analysis was based on the gate-level netlists using VCS. Test cases were constructed based on data sparsity during the actual training process of Spiking ResNet-18, -50, and spiking VGG-9 models. The spike sparsity for Spiking ResNet-18, -50, and spiking VGG-9 was 90%, 80%, and 88%, respectively. The $fire'$ sparsity was 68%, 54%, and 56%, and $\nabla u$ sparsity was 4%, 1%, and 62%, respectively. VCS simulation outputs waveform files in FSDB format, which were converted to toggle rate files of SAIF format. Power reports were generated based on gate-level netlists and SAIF files using DC.

We constructed test cases with 0% sparsity to obtain power consumption without any sparse optimization ($P_0$). To eliminate the influence of power consumption unrelated to sparsity, we also constructed test cases with 100% sparsity and obtained its power consumption ($P_{100}$). If the power consumption of SNN test case is $P_{SNN}$, the percent of power reduction by sparse optimization for the SNN case was defined as: $(P_0 - P_{SNN}) / (P_0 - P_{100}) \times 100\%$.

## SNN compilation and deployment on the multi-core neuromorphic architecture

We developed a lightweight deep learning software framework using Eclipse by the C language, in which SNN models are represented through computation graphs expressed by a basic operator library. The Frontend supported construction of various deep convolutional SNN models through flexible composition of basic operators from our custom SNN operator library, covering a large number of recent SNN models (as listed in Supplementary Table 12). Additionally, it provided a C-language interface to support the rapid development of new operators at the kernel level to follow the latest SNN progress. The compiler automatically constructed various hardware-friendly computing graphs compatible with our architecture, depending on various SNN tasks. The computation graphs of an SNN model were split into feedforward and backpropagation sub-graphs according to convolutional layers. One sub-graph was deployed on one sub-core, and adjacent layers were deployed on adjacent cores in the multi-core architecture. The sub-graphs, after the basic operator fusion, were first converted into the custom-designed intermediate representation (IR) described in the C language. Then, the IR was compiled to generate device-side files for each sub-graph, including task description of the sub-core to compute a sub-graph. Task-aware instruction flows for intra- and inter-core data transmission and computation execution associated with each core were generated automatically based on the device-side files. Driver running on the Controller unit parsed device-side files and generated detailed implementation information for each sub-core, such as instruction's opcodes and operands, instruction sequence, input and output data orchestration in SRAM and HBM, and wrote them into Registers of the sub-core. In our multi-core architecture, detailed implementation information for all layers in sub-cores was generated before the training started; therefore, it required instruction information transmission only once from the host to perform training on one batch of samples.

The exact steps of SNN deployment and training setup in our multi-core architecture (Fig. 5d) include the following steps: 1) Model Construction. Generate the SNN model represented in C Language through composing basic operators in our custom operator library in Frontend. 2) Model Compilation, including IR Generation and Hardware-side File Generation. 3) Training Initialization, including storage (SRAM, DRAM) allocation, loading model parameters and samples, and generation of instruction flow according to the Hardware-side file. 4) Training Execution. All computing cores execute their assigned computation tasks in parallel.

We adopted the custom intermediate representation (IR) to facilitate the SNN representation and its mapping to our multi-core neuromorphic architecture. Open-source neuromorphic software frameworks such as Norse, SpikingJelly, snnTorch, Sinabs, and Spyx are built upon PyTorch or JAX IRs. These IRs are primarily targeted at artificial neural networks (ANNs), lacking the explicit representation of the temporal dimension and spiking behavior of neurons in SNN[48]. The SNN representation based on our custom IR can be mapped to our multi-core architectures more straightforwardly, compared with the representation based on those open-source IRs (for details see Supplementary Content 1). To improve the cross-platform compatibility of our software toolchain, we developed two Python-based IR conversion modules, convert_ir_to_nir.py and convert_nir_to_ir.py, to implement bidirectional conversion between our custom IR and NIR. The results of conversion are determined by comparisons between the converted and the directly exported IRs using the Notepad + +'s Compare plugin (Supplementary Fig. 10).

## FPGA implementation

The multi-core near-memory neuromorphic architecture was implemented on a platform with five FPGA boards (VCU128, Xilinx) and a CPU server (ThinkSystem SR670, Lenovo). FPGAs were connected with the CPU via the PCIe 3.0 interface.

Inter-FPGA communication is via 100 G CMAC (Xilinx CMAC). We developed an Ethernet Adapter (EA, Fig. 5b) module to address bus width mismatch, channel merging and splitting, clock frequency mismatch between the routers and CMAC for the inter-FPGA connection. The transferred 32B flits supported by the 256-bit AXI-S bus from two routers on one-side FPGA are converted in a two-to-one manner in the BWC of EA (Fig. 5b), yielding the 64B flits supported by the 256-bit AXI-S bus connected with 100 G CMAC. The merge circuit in EA_Transmitter (Fig. 5b) employs a Round-Robin scheduling policy across

routers and a priority-based policy across VCs to select one of four VC channels from two routers on one side for current transmission. Data is transmitted via 100 G CMAC to the EA on the other side. The split circuit in EA_Receiver decodes the VC channel and targets Router information of Packets based on the packet header and directs the packet to one of four VC channels. Then, the packets are converted into 32B flit format and transferred to the target router. The asynchronous FIFO buffers in the EA module support cross-clock domain data transmission between the Router side (200 MHz) and CMAC side (333 MHz).

All BP, FP, WG convolution computations during SNN training were performed on FPGAs. Each FPGA board deployed four computing cores. Each FP16 adder or FP16 multiplier consisted of 1 DSP. Each computing core consumed 1183 DSPs, 216 K LUTs, 209 K FFs, 244 BRAMs, and 104 URAMs. The single core Verilog code was encapsulated into custom IP by the FPGA development tool Vivado (Xilinx). MicroBlaze of FPGA was used as the Controller unit of each sub-core. FPGA boards were interconnected through Routers and Ethernet interface. We developed a Transport Layer Protocol for the Ethernet interface, and RTL codes for clock and reset management circuits to achieve correct synchronization and reset of each computing core. NoC adopted a streaming protocol and multi-port transmission scheduling based on flit segmentation. The computing core operated at 150 MHz and Router operated at 200 MHz achieving 50×6 Gbps bandwidth with its six ports. All RTL codes and IPs were synthesized first, then, after place and route, bitstream files were generated in Vivado and downloaded to Configuration RAM (CRAM) on FPGA for testing and verification using JTAG tools. Finaly, the bitstream files were burned into the Flash memory in each FPGA board.

We developed a driver for initializing all sub-cores in FPGA, parsing the device-side files of each computing sub-core, and writing the corresponding instruction information into registers of the sub-core. The driver was running on the Controller unit (MicroBlaze) of each sub-core. Model weights and dataset were first loaded into the HBM from the host, then transferred to SRAM for computations by dynamically calling the DMA instructions during training.

For the smart traffic scenario, open-source datasets were adopted. The Classification and Recognition dataset includes 6,358 manually labeled samples spanning ten distinct traffic sign categories. The Vehicle and Pedestrian Detection dataset contains 5,748 images representing 8 different traffic objects such as pedestrians, small lorries, and trucks. The DVS-Gesture dataset is recorded using a DVS, consisting of spike trains with two channels corresponding to ON- and OFF-event spikes. The dataset contains 11 hand gestures from 29 subjects under 3 illumination conditions, with 1176 training samples and 288 testing samples.

## Data availability

The MNIST, CIFAR-10/100, ImageNet2012, Traffic Sign Classification and Recognition, Vehicle and Pedestrian Detection, and DVS-Gesture datasets that we used for benchmarks are publicly available [https://yann.lecun.com/exdb/mnist/index.html, https://www.cs.toronto.edu/-kriz/cifar.html, https://www.image-net.org/, https://www.kaggle.com/datasets/wjybuqi/traffic-sign-classification-and-recognition, https://www.kaggle.com/datasets/enesbayturk/vehicle-and-pedestrian-detection-dataset, http://research.ibm.com/dvsgesture/]. Other data that support the plots within this paper can be found in the GitHub repository [https://github.com/dayanhn/NeuroMC] and has been archived in Zenodo with [https://doi.org/10.5281/zenodo.18311893][52] under Apache License 2.0.

## Code availability

The code is available in the GitHub repository [https://github.com/dayanhn/NeuroMC] and has been archived in Zenodo with [https://doi.org/10.5281/zenodo.18311893][52] under Apache License 2.0.

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

## Acknowledgements

This work was supported by Shenzhen Science and Technology Program (KQTD20240729102051063) to Y.T.; the National Natural Science Foundation of China (62425101) to Y.T.; the National Natural Science Foundation of China (62332002) to Y.T.; the National Natural Science Foundation of China (62236009) to G.L. and Z.M.; the key project of the Pengcheng Laboratory to Y.T., H.Z., G.L., X.X.C., Q.M. and Z.M.

## Author contributions

Y.T., H.Z., G.L., X.X.C. provided conceptualization and supervised the project. Y.L., W.L., X.Z. contributed to software simulation of architecture; M.L., X.X., P.Q., X.L., J.Z., D.W., X.L.C., and X.X.C. contributed to architectural design, coding, register-transistor level simulations, and FPGA hardware implementation. Z.Z. contributed to software development. Z.Z., Q.M., and Y.M. contributed to FPGA applications. Z.Z., M.L., X.Z., X.X., P.Q., Y.L., W.L., and H.Z. conducted data analysis; H.Z., Q.M., Y.T., G.L., W.W., and Z.M. write the manuscript.

## Competing interests

The authors declare no competing interests.
