## [Transparent Peer Review file · Nature Communications]

A Highly Energy-Efficient Multi-Core Neuromorphic Architecture for Training Deep Spiking Neural Networks

Corresponding Author: Professor Huihui Zhou

Version 0:

Reviewer comments:

Reviewer #1

(Remarks to the Author)

In this paper, the authors propose a multi-core neuromorphic architecture to address the growing need for edge training in dynamically changing environments. The architecture supports Feedforward-Propagation, Back-Propagation, and Weight-Gradient operations, enabling parallel computing at both the engine and core levels. The authors claim that this study develops the first multi-core neuromorphic architecture supporting direct SNN training, facilitating neuromorphic computing in edge-learnable applications.

Comments for the Authors:

1. What is the rationale behind using Spiking-ResNet18, ResNet50, and VGG16 for simulation? Please provide a clear explanation for choosing these specific models.
2. Compared to the A100, the DRAM occupancy of your architecture was reduced. Please elaborate on these findings and their significance.
3. The RTL power analysis indicates that your sparse design reduces the energy consumption of the computing core during training. Please explain this result in more detail.
4. Clearly state the objective of this research study. Additionally, highlight what has already been done in this area and what new contributions your work brings, along with their scientific benefits.
5. Consider adding an algorithm or pseudocode to illustrate the design and functionality of the multi-core neuromorphic architecture for SNNs.
6. Please discuss the findings presented in Table 1 in greater detail. What insights can be drawn from the data?
7. Efficient communication between cores is critical. Can you elaborate on how your proposed architecture addresses inter-core communication challenges?
8. SNNs use spike-based communication, which makes backpropagation more complex and computationally expensive. How does your study address this issue? Additionally, how do you efficiently parallelize training operations (e.g., FP, BP, WG) across multiple cores without compromising accuracy?
9. Training SNNs in dynamically changing environments (e.g., edge devices) requires real-time adaptation, which is difficult to achieve. Are you going to tackle this challenge?
10. Are you addressing the overfitting issue in your study?
11. I recommend that the authors clearly outline the challenges of multi-core neuromorphic architectures and then specify which issues their study aims to address. I think the authors need to describe their novelty very clearly and discuss in great detail how their approach differs from the state of the art.

Reviewer #3

(Remarks to the Author)

In this paper the authors present a highly energy-efficient multi-core neuromorphic architecture for training deep spiking neural networks. While I do not have any major concerns regarding the proposed architecture, in my opinion, the paper is not of a high enough caliber for Nature Communications. The presentation of the manuscript requires substantial improvement and instead of taping out a silicon chip, the authors validate their design using an FPGA implementation. The authors mention the use of the Eclipse IDE to develop a basic operator library – it is unclear to me why existing open-source

intermediate representations were not adopted/extended and what novelty the automated network has.

I have the following specific comments:

1. The empirical training behavior is not well described, and it is unclear to me how scalable the architecture is.
2. For functional verification, what was the code coverage?
3. How flexible is the developed frontend/compiler/architecture? What are the exact steps required to deploy a new network and setup its training and what are the associated limitations?
4. Was any form of PVT (process, voltage, temperature) variation investigated?
5. How is communication handled between FPGA platforms? If a chip were to be taped out, how would this alter the performance?
6. The training power is well reported, but what about the time?
7. Why was FP16 used over BF16?
8. I understand a direct comparison is difficult to make, but to get some idea of the value proposition of the architecture, it would be appreciated if the authors could make some comparison to standard network training on edge devices, e.g., mGPUs or embedded systems. The investigated networks are relatively small, so it may be feasible to fine-tune such networks in real-time using more standard hardware platforms.

Version 1:

Reviewer comments:

Reviewer #1

(Remarks to the Author)

While the authors have addressed most of the comments and the current version appears significantly improved, I am still unclear about the key contributions of this work. I would like to request that the authors elaborate on the proposed multi-core neuromorphic architecture, as its significance is not sufficiently clear.

Several revisions have been included as supplementary materials, which raises concerns that essential information may be still missing from the main article. For a high-quality journal venue, it is important that the authors clearly address the comments related to the significant contributions within the main manuscript.

(Remarks on code availability)

Reviewer #3

(Remarks to the Author)

First, I would like to thank the authors for considering and responding to each of my original comments. I was the third reviewer during the first round of review. I have the following further comments/questions:

1. The authors have better motivated their proprietary IR. Although, I still feel like some of the arguments they have made could be better conveyed in the main text and/or supplementary notes, and I don't agree with all their arguments. While one can appreciate the comparison made in Fig. 5, exactly what level of abstraction/decomposition of operands is optimal remains a largely open question. One could argue by decomposing an operation into its primitives, more flexibility could be gained with respect to mapping and various compiler-level optimizations. Perhaps what would be more helpful, would be a comparison of the different IRs used in similar settings, clearly highlighting exactly which aspects of the other IRs are sub-optimal and motivate the development of a proprietary IR. While a size comparison is made between PyTorch and the proposed IR, it is not entirely clear how this relates to the other IRs, e.g., ones based on JAX.
2. Interfacing the proprietary IR with NIR is a great idea and I strongly suggest it is done in this paper, not in a future study. This would make the authors claims with respect to modularity and ecosystem integration much stronger, and if the network descriptions in the NIR IR were made available with the paper, would allow direct comparisons between this work and other hardware-backends.
3. I do not quite understand how the performance is determined for the Chip in Table 10. Could the authors please clarify their modeling assumptions? Was the RTL code synthesized? If so, what CMOS process was assumed? Is the achievable operating voltage/average power the same as reported on the FPGA?
4. Could the authors please clarify exactly what code will be released to the repository if the paper is accepted?

(Remarks on code availability)

The repository is a placeholder. The authors claim code and data will be released upon publication of the associated paper.

Version 2:

Reviewer comments:

Reviewer #1

(Remarks to the Author)

(Remarks on code availability)

The authors have addressed the majority of the comments raised during the review process.

Reviewer #3

(Remarks to the Author)

The authors have addressed my remaining concerns and I now endorse publication.

(Remarks on code availability)

Appears to be well documented and structured. I did not explicitly run it.

Reviewer #1:

While the authors have addressed most of the comments and the current version appears significantly improved, I am still unclear about the key contributions of this work. I would like to request that the authors elaborate on the proposed multi-core neuromorphic architecture, as its significance is not sufficiently clear.

Reply: Sincere thanks for the constructive comment. We have made thorough revisions to articulate the key contribution regarding multi-core neuromorphic architecture, multi-level parallelism, and energy efficiency optimization, and its significance in **Introduction**. We now state:

Line 123: “To the best of our knowledge, we propose the first multi-core, sparsity-aware neuromorphic architecture enabling distributed deployment of the entire deep SNN model across cores to perform BP-based training. We develop the highly parallelable computing core, efficient intra- and inter-core communication mechanisms, multi-level parallelism strategy, and comprehensive sparse computation optimization specifically tailored for SNN training. In the neuromorphic architecture aspect, we propose the three-engine computing core inter-connected with a 2D mesh network. The Feedforward-Propagation (FP), Back-Propagation (BP), and Weight-Gradient (WG) engines based on PE-array with FP16 Adder or MAC execute diverse computational forms in SNN training. Communication mechanisms featuring the six-way router, three primary inter-core/engine (FP-FP, BP-BP, FP-BP) pathways, and unified package format for bit-based spiking and floating-point data, enable efficient data transmission along both model layers and time steps across computing engines. The 2D mesh topology and Globally Asynchronous Locally Synchronous (GALS) approach deployed in our multi-core architecture support its high scalability. To accelerate SNN training, we develop a multi-level parallel computing strategy with inter-core pipeline parallelism and intra-core engine parallelism. At the inter-core level, our multi-core architecture performs parallel feedforward and backward operations on different model layers simultaneously. Within the same core, the three engines perform FP, BP, and WG computation in parallel. For energy efficiency optimization, we introduce sparsity-aware circuits in the three engines, leveraging all types of sparse signals in SNN training to skip redundant computations and memory access, reducing energy consumption by 45%–60%. Through multiple dataflows in different engines to enhance data reuse and single-bit storage of spiking data, SNN training in our architecture achieves 55 ~ 85% reductions of the high-energy-consuming DRAM access compared to the A100 GPU. Overall, our architecture achieves 190%~330% performance of Jetson AGX Orin, with a high energy efficiency of 1.05TFLOPS/W@ FP16 @ 28nm. Along with our software toolchain, our architecture is implemented on FPGAs and achieves successful supervised learning and federated learning. Together, our work significantly enhances the training efficiency of neuromorphic models and broadens the applications of neuromorphic architectures in the resource-constrained scenario, especially in edge-learnable environments.”

We also revised **Discussion** to clarify these contributions. We now state:

Line 651: “To the best of our knowledge, we developed the first multi-core neuromorphic architecture enabling BP-based training of deep SNN models in a distributed manner across computing cores. The proposed computing cores with three PE-array based engines execute diverse computational forms in SNN training. Communication mechanisms featuring the six-way router, three primary SNN training-oriented inter-core/engine pathways, and a unified package format ensure efficient transmission of multi-type data between intra- and inter-core engines.”

Line 691: “Leveraging on our multi-level parallel computing strategy with inter-core pipeline parallelism and intra-core FP, BP, WG engine parallelism, the architecture...”

We also elaborated on the proposed multi-core neuromorphic architecture, the multi-level parallelism strategy, sparse computation optimization and DRAM occupancy reduction, and FPGA implementation of the multi-core architecture in revised **Results and Methods**. Please see the details in our reply to the next question.

Several revisions have been included as supplementary materials, which raises concerns that essential information may be still missing from the main article. For a high-quality journal venue, it is important that the authors clearly address the comments related to the significant contributions within the main manuscript.

Reply: Sincere thanks for the important suggestion. We moved a significant portion of the contents from the **Supplementary Information** to the main manuscript to comprehensively cover information about our contributions:

1) To elaborate on the proposed multi-core neuromorphic architecture, we revised Figure 2 in the main manuscript. In addition to plots for single computing core, Router, Soma and Grad module in Figure 2 in the previous manuscript, we added plots for the proposed multi-core neuromorphic architecture with 8x4 2D mesh topology (Suppl. Figure 2), the important communication module-- Dispatch Unit (DU) (Suppl. Figure 3a), the Globally Asynchronous Locally Synchronous (GALS) design (Suppl. Figure 1c), and unified package format for spiking and floating-point data (Suppl. Figure 3b) into the revised Figure 2 as Figure 2b, 2f, 2g, and 2h, respectively. The revised Figure 2 in the manuscript is shown below.

Figure 2. The Multi-core Neuromorphic Architecture for SNN Training. (a) Architecture of a single computing core with an FP sub-core and a BP sub-core. (b) The architecture with a 4x8 2D-Mesh topology. (c) Soma module in FP engine. (d) Grad module in BP engine. (e) Router (middle) with its associated BW and RC modules (left), SA and ST modules (right). (f) The Dispatch Unit (DU) module. (g) The Globally Asynchronous Locally Synchronous (GALS) design. Clock frequency for orange lines: 667MHz; for blue lines: 500MHz. (h) The structure of messages, packets, and head flits. (i) Procedure of SNN deployment and parallel forward and backward computation in training. (j) Distributed training procedure in the multi-core neuromorphic architecture.

We revised the related description in **Results 2.2 Multi-Core Near-memory Neuromorphic Architecture for SNN Training**. We now state:

Line 283: “For intra- and inter-core communication, our multi-core architecture employs a 4×8 2D-Mesh topology (Figure 2b) using the XY routing strategy, with each node comprising a router and a computing core (Figure 2b). Each router features six channels (Figure 2e): four for east, west, north, and south inter-node communication, and two dedicated channels for the FP sub-core and BP sub-core within the computing core. We designed a unified packet format for both bit-based spiking and floating-point data (Figure 2h) and modules such as Dispatch Unit (DU) (Figure 2f) to support multiple types of data transmission. To support temporal dependency across cores, data of a single time step is organized as a message (Figure 2h) to avoid mixing data from different time steps. The message is split into packets. The DU module serves as a data distribution hub for data transmission with SRAMs in sub-cores (see details in Methods 4.2B).”

We move **Suppl. Contents 1.1 Inter-Core Communication in Our Multi-Core Architecture** to **Methods 4.2B**. We now state:

Line 872: “B. Intra- and Inter-Core Communication in the Multi-Core Architecture

During SNN training, an SNN model is distributed across all computing cores in our multi-core architecture, with one layer or a segment of a layer allocated to an engine within a computing core. The communication mechanisms ensure accurate transmission of multiple types of data, such as bit-based spiking data, floating-point membrane potential and its gradients, along both model layers and time steps in both sequent and reverse orders during SNN training. Our intra- and inter-core communication design focuses on the following aspects: **Unified Packet Design**. To support temporal dependency across cores, data of a single time step was organized as a message (Figure 2h) to avoid mixing data from different time steps. The message was split into 2KB packets. Both floating-point and bit-based spiking data in SNN training are transferred with a unified packet format. Packets are transmitted in 32B flits, with each flit carrying 16 FP16 data or 256 single-bit spiking data. The detailed information in the packet header is shown in Suppl. Figure 2d.

2D-Mesh Topology and Router Design. Our multi-core architecture employs a 4×8 2D-Mesh topology (Figure 2b), with each node comprising a router (Figure 2e) and a computing core (Figure 2a). Each router features six channels (Figure 2e): four for east, west, north, and south inter-node communication, and two dedicated channels for the FP sub-core and BP sub-core within the computing core to increase the bandwidth of data transmission during FP and BP computations. Each router direction supports two virtual channels (VCs), with incoming data stored in corresponding FIFOs (First-In First-Out memories). The Route Computation (RC) unit reads the head flit from the FIFO, parses the head flit, and determines the next transmission direction using the XY routing strategy by comparing the target X and Y coordinate information (*target_x*, *target_y*) in the head flit with the router's coordinates (*x*, *y*). If the router is the target router during transmission, it uses information in the *target_befe* field of the head flit to select the FE or BE output port. When multiple packets from different directions request the same output port, the Switch Allocation (SA) module employs a round-robin arbitration policy to prioritize transmission. The Switch Traversal (ST) module transfers the packet flit-by-flit based on back-pressure from the downstream unit. The 2D-Mesh topology has been commonly adopted in previous multi-core neuromorphic architectures.

Globally Asynchronous Locally Synchronous (GALS) Communication Mechanism (Figure 2g). The architecture adopts a globally asynchronous design with separate clock frequencies for computing cores and routers: all routers share a common clock (Router-clk, 667MHz), while each core has an independent clock (core-clk, 500MHz). Asynchronous FIFOs are implemented at the router-core interface to support cross-clock domain data transfer. The GALS design avoids clock skew issues in global clock tree designs and improves the scalability of our architecture. The higher router clock frequency increases data transmission bandwidth to 170Gbps/channel.

Data Transmission Modules and Pathways. We also designed Dispatch Unit (DU, Figure 2f; see Suppl. Figure 2a-b for detailed DU in FP and BP sub-cores) and Network Interface (NI) module (Suppl. Figure 2c)

to support intra- and inter-core transmission for multiple types of data. The DU serves as a data distribution hub between SRAMs in sub-cores and NI or DMA. The NI Transmitter fetches data payloads from SRAMs via DU and encapsulates them into packets, and performs flow control of data transmission via the router. The NI Receiver performs decapsulation of packets, and manages their address mapping in SRAMs/DRAMs. Based on these modules, our multi-core architecture forms three primary data transmission pathways (Suppl. Figure 1a) – inter-core FP-FP transmission, inter-core BP-BP transmission, and intra-core FP-BP transmission – to support SNN training in our multi-core architecture.”

2) To elaborate on the multi-level parallelism in our multi-core neuromorphic architecture during SNN training, we refined the Suppl. Algorithm 1 and added the refined algorithm to the main manuscript as Algorithm 1. The Suppl. Algorithm 1 is kept in the revised Supplementary Information to provide detailed information. The refined algorithm is shown below.

We revised the related description in **Result 2.3 Data Flow and Parallel FP-BP-WG Computations**. We now state:

Line 372: “Algorithm 1 shows our strategy of parallelized operations in our multi-core architecture during SNN training with inter-core pipeline parallelism and intra-core FP, BP, WG engine parallelism. At the core level, our multi-core architecture performs parallel feedforward and backward operations on different model layers, processing different samples simultaneously. Within the same core, FP and BP engines compute spike outputs and membrane potential gradients of the same model layer on different samples with opposite time-step orders simultaneously, and the WG engine computes weight gradients of the next layer in a reverse order in parallel with the computations in the FP and BP engines. Thus, our architecture supports highly parallel computing at both the core and engine levels to accelerate SNN training (Suppl. Algorithm 1 for more details).”

3) To elaborate on findings about DRAM occupancy reduction compared to the A100 GPU of our multi-core architecture in SNN training, we converted data with the batch size of 16 in the Suppl. Table 4 into plots and added them to revised Figure 3 as Figure 3r, 3s, 3t to show detailed DRAM occupancy reduction. The revised Figure 3 is shown below. Suppl. Table 4 is also kept in the Supplemental Information to cover different batch sizes.

Algorithm 1 Parallelized SNN Training Operations in the Multi-core Neuromorphic Architecture

```

1:  $N, L_i$ ; % N: batch size; L: SNN layer number
2:  $l \in \{0, \dots, L-1\}$ ; % Layer index
3:  $T$ ; % Total number of time steps
4:  $Core\_Num = 32$ ; % Total number of Compute cores in the architecture
5:  $Engine\_Num = 3$ ; % Total number of Compute engines in a core
6:  $n = \text{zero}(1, Core\_Num)$ ; % Indices of samples processed in FP engines
7:  $n_{bp} = \text{zero}(1, Core\_Num)$ ; % Indices of samples processed in BP engines
8:  $n_{wg} = \text{zero}(1, Core\_Num)$ ; % Indices of samples processed in WG engines
9: parallel for  $k = 0 : Core\_Num - 1$  do
10:   parallel for  $e_i = 0 : Engine\_Num - 1$  do
11:     if  $e_i == 0$ : % Computation in the FP engine
12:       Read weights  $w_k^l$  from SRAM;
13:       for  $t = 0 : T - 1$  do
14:         Read spike inputs  $s_{n[k],t}^{l-1}$  from SRAM;
15:         Compute FP convolution  $\text{Conv.FP}_{n[k],t}^l$ ;
16:         Compute potential  $u_{n[k],t}^l$ ;
17:         Compute spike outputs  $s_{n[k],t}^l$ ;
18:         Send the spike output to engines for following FP computations;
19:         Send the potential and spike to engines for following BP/WG computations;
20:       endfor
21:       if  $l == L - 1$  &  $t == T - 1$  calculate loss;
22:       Update sample index  $n[k]$ ; Goto line 13 to process the next sample.
23:     if  $e_i == 1$ : % computation in the BP engine
24:       Read  $w_k^{l+1}$  from SRAM;
25:       for  $t = T - 1 : 0$  do
26:         Read potential gradients  $\nabla u_{n_{bp}[k],t}^{l+1}$  from SRAM;
27:         Read inputs  $s_{n_{bp}[k],t}^l, u_{n_{bp}[k],t}^l$  from SRAM;
28:         Compute potential gradient  $\nabla u_{n_{bp}[k],t}^l$ ;
29:         Send the potential gradient to engines for following BP/WG computation;
30:       endfor
31:       Update sample index  $n_{bp}[k]$ ; Goto line 25 to process the next sample.
32:     if  $e_i == 2$ : % Computation in the WG engine
33:       for  $t = T - 1 : 0$  do
34:         Read potential gradients  $\nabla u_{n_{wg}[k],t}^{l+1}$  from SRAM;
35:         Read spike inputs  $s_{n_{wg}[k],t}^l$  from SRAM;
36:         Compute gradients  $\nabla w^{l+1}[i, j]$ ;
37:       endfor
38:       Compute weight gradients  $\nabla w^{l+1}$ ;
39:       Update sample index  $n_{wg}[k]$ ; Goto line 33 to process the next sample;
40:       Update weight when finishing a batch.
41:     endfor
42:   endfor

```

Figure 3. Dataflow and Parallel Computations in the Multi-core Neuromorphic Architecture. (a-c) show dataflows of FP, BP, and WG engines, respectively. (d-f) show the utilization of FP, BP, WG engines and computing core during Spiking ResNet-18, -50, and spikingVGG-16 training, respectively. Utilization curves during two batches of the training period are plotted. (g-i) show the comparison of DRAM occupancy between our multi-core architecture and A100 GPU during Spiking ResNet-18, -50, and spiking VGG-16 training, respectively. (j-l) show the comparison of DRAM access during the three training phases, respectively. (m-o) show the comparison of training performance (fps) during the three training phases, respectively. (p) and (q) show the effects of increasing core numbers of our 2D-mesh architecture on throughput (fps) and slope (fps/core) during SNN training, respectively. (r-t) show the comparison of DRAM occupancy consisted of intermediate data, optimizer state, weight, and weight gradient. Batch size: 16.

We revised the corresponding description in **Results 2.3 Data Flow and Parallel FP-BP-WG Computations**. We now state:

Line 455: “The above reduction was mainly due to the decrease in the occupancy of intermediate data (Figure 3r-t) accounting for 72.14%~80.08% of the total DRAM occupancy of the A100 GPU training with the batch size of 16, and training in our architecture reduced this part of occupancy by 62.43%~77.89%. Across batch sizes 16, 32, 64, and 128, the intermediate data accounted for 72.04%~97.03% of the A100 DRAM occupancy, and training in our architecture reduced this part of occupancy by 62.43%~77.89% (Suppl. Table 4).”

4) To elaborate on the effects of our sparse computation optimization, we added the percentage of the energy consumption reduction by the sparse computation optimization in Figure 4d-f. The revised Figure 4 is shown below.

Figure 4. Sparse Computation Optimization. (a-c) Sparse computation optimization circuit designs for the FP engine, BP engine, and WG engine, respectively. The red and orange lines serve as gating signals of computation units and SRAM accesses, and the blue lines serve as control signals for Multiplexers. (d-f). Comparisons of the energy consumption before and after sparse optimization for spiking ResNet-18, -50, and spiking VGG-9 training, respectively. **The percentage values near bins indicate the energy consumption reduction by the sparse computation optimization.**

5) To elaborate on the FPGA implementation of our multi-core architecture, we add inter-FPGA communication (Suppl. Figure 8a), software toolchain (Suppl. Figure 5a), steps of SNN deployment and training in our architecture (Suppl. Figure 5b) to the revised Figure 5 as Figure 5b, 5c, and 5d, respectively. The revised Figure 5 is shown below.

Figure 5. FPGA Implementation of the Multi-core Neuromorphic Architecture. (a) FPGA platform. (b) Inter-FPGA communication. (c) Software toolchain for our multi-core architecture. (d) Steps of SNN construction, deployment, and training execution. (e) Implementation of the 20-core neuromorphic architecture for SNN training based on BP on FPGAs for image classification. (f)-(g) Accuracy and loss curves of SNN training on FPGAs in the image classification tasks. (h) Smart city scenario. (i) Illustration of a continual learning procedure on an edge processor. (j)-(l) Accuracy curves of the continual learning and non-continual learning models on the Traffic Sign Classification and Recognition dataset, Pedestrian Augmented Traffic Light dataset, and DVS-Gesture dataset, respectively. (m) Illustration of federated learning procedure on an edge processor. (n) Implementation of the 5-worker federated learning based on FPGAs, with each worker implemented on one FPGA, and worker1 also used as the server. (o)-(p) Accuracy and loss curves of the five workers and the federated model on the Synthetic ASL Alphabet dataset. (q)-(r) Accuracy and loss curves of the five workers and the federated model on the DVS128 Gesture dataset.

We revised the related description in **Results 2.6 FPGA Implementation and Applications**. We now state: **Line 597: "We developed an Ethernet Adapter module (Figure 5b; see details in Methods) to support inter-FPGA data transmission between any cores deployed on different FPGAs through the XY routing strategy."**

We move **Suppl. Contents 1.4 Inter-FPGA Communication** to **Methods 4. 4 FPGA Implementation**. We now state:

Line 1234: "Inter-FPGA communication was via 100Gb CMAC (Xilinx CMAC). We developed an Ethernet Adapter (EA, Figure 5b) module to address bus width mismatch, channel merging and splitting, clock frequency mismatch between the routers and CMAC for the inter-FPGA connection. The transferred 32B flits

supported by the 256-bit AXI-S bus from two routers on one-side FPGA are converted in a two-to-one manner in the BWC of EA (Figure 5b), yielding the 64B flits supported by the 256-bit AXI-S bus connected with 100G CMAC. The merge circuit in EA_Transmitter (Figure 5b) employs a Round-Robin scheduling policy across routers and a priority-based policy across VCs to select one of four VC channels from two routers on one side for current transmission. Data is transmitted via 100G CMAC to the EA on the other side. The split circuit in EA_Receiver decodes the VC channel and targets Router information of Packets based on the packet header and directs the packet to one of four VC channels. Then, the packets are converted into 32B flit format and transferred to the target router. The asynchronous FIFO buffers in the EA module support cross-clock domain data transmission between the Router side (200MHz) and CMAC side (333MHz).”

We move **Suppl. Contents 1.2 SNN Deployment and Training Setup in Our Multi-Core Architecture** to **Methods 4.3**. We now state:

Line 1189: “The exact steps of SNN deployment and training setup in our multi-core architecture (Figure 5d) include the following steps: 1) Model Construction. Generate the SNN model represented in C Language through composing basic operators in our custom operator library in Frontend. 2) Model Compilation, including IR Generation and Hardware-side File Generation. 3) Training Initialization, including storage (SRAM, DRAM) allocation, loading model parameters and samples, and generation of instruction flow according to the Hardware-side file. 4) Training Execution. All computing cores execute their assigned computation tasks in parallel.”

Reviewer #3:

First, I would like to thank the authors for considering and responding to each of my original comments. I was the third reviewer during the first round of review. I have the following further comments/questions:

1. The authors have better motivated their proprietary IR. Although, I still feel like some of the arguments they have made could be better conveyed in the main text and/or supplementary notes, and I don't agree with all their arguments. While one can appreciate the comparison made in Fig. 5, exactly what level of abstraction/decomposition of operands is optimal remains a largely open question. One could argue by decomposing an operation into its primitives, more flexibility could be gained with respect to mapping and various compiler-level optimizations. Perhaps what would be more helpful, would be a comparison of the different IRs used in similar settings, clearly highlighting exactly which aspects of the other IRs are suboptimal and motivate the development of a proprietary IR. While a size comparison is made between PyTorch and the proposed IR, it is not entirely clear how this relates to the other IRs, e.g., ones based on JAX.

Reply: Thank you for pointing out this important question. We fully agree that the optimal level of operand abstraction remains an open question, and decomposing operations into primitives could provide more flexibility for mapping and compiler optimizations. We added size information of JAX IR and showed a comparison between our custom IR, PyTorch IR, and the JAX IR together in Suppl. Table 8. We revised the related description in the main text and Supplementary Information.

In **Methods 4.3 SNN Compilation and Deployment on the Multi-core Neuromorphic Architecture**, we now state:

Line 1203: “We adopted the custom intermediate representation (IR) to facilitate the SNN representation and its mapping to our multi-core neuromorphic architecture. Open-source neuromorphic software frameworks such as Norse, SpikingJelly, snnTorch, Sinabs, and Spyx are built upon PyTorch or JAX IRs. These IRs are primarily targeted at artificial neural networks (ANNs), lacking the explicit representation of the temporal dimension and spiking behavior of neurons in SNN (Pedersen et al., Sep. 2024). The SNN representation based on our custom IR can be mapped to our multi-core architectures more straightforwardly, compared with the representation based on those open-source IRs (for details see Suppl. Content 1).”

In **Supplementary Information**, we revised **Suppl. Content 1**, we now state:

“Supplementary Content 1. Comparison of Intermediate Representations in SNN Representation and Mapping

Open-source neuromorphic software frameworks such as Norse (Pehle et al., 2021), SpikingJelly (Fang et al., 2023), snnTorch (Eshraghian et al., 2023), Sinabs (Sheik et al., 2023), and Spyx (Heckel et al., 2024) are built upon PyTorch or JAX and share the intermediate representations (IRs) with these mainstream deep learning frameworks. PyTorch and JAX IRs are primarily targeted at artificial neural networks (ANNs) (Pedersen et al., 2024). These open-source IRs using basic operations, such as multiplications, additions, and reshaping, enable flexibility in mapping and various compiler-level optimizations. Meanwhile, these IRs lack the explicit representation of the temporal dimension and spiking behavior of spiking neurons in SNN (Pedersen et al., 2024), rendering the SNN representation based on these IRs complex. For

Suppl. Figure 7. Intermediate Representation (IR) of SNN. (a) PyTorch IR for representing a two-time-step LIF model; (b) Our IR for representing a two-time-step LIF model.

example, in the PyTorch IR representation of a simple two-time-step Leaky Integrate-and-Fire (LIF) neuron followed by convolution (Suppl. Figure 8a), crucial components like time-step indexing and spiking behaviors of the neuron are difficult to be extracted and mapped to the neuromorphic architectures with dedicated hardware to support computation through time steps and spiking mechanisms. Our custom IR introduces a dedicated Soma operation (Suppl. Figure 8b) that concisely encapsulates all key aspects of the LIF neuron, resulting in its straightforward mapping to the Soma module (Figure 2c) mainly through setting these parameters shown in the Soma operation (Suppl. Figure 8b) in corresponding registers in our neuromorphic architecture. We also compare the size between PyTorch, JAX, and our custom IRs in representing deep SNNs (Supplementary Table 8a), showing that representations based on our IR are clearly smaller than those based on PyTorch IR and JAX IR. We further compare PyTorch and JAX IRs in representing ANNs and SNNs, showing that PyTorch and JAX IRs are more efficient for the ANN representation (Supplementary Table 8b).

On the other hand, neuromorphic software tools such as PyNN (Davison et al., 2019), NEST (Gewaltig et al., 2007), Nengo (Bekolay et al., 2014), and NEURON (Carnevale et al., 2006) are primarily designed for simulating biological neural networks. Tools like Lava (Williams et al., 2023), Corelet (Amir et al., 2013), and Rockpool (Muir et al., 2019) support the SNN representation for specific neuromorphic hardware platforms such as Loihi (Davies et al., 2018, 2021), TrueNorth (Merolla et al., 2014), and Xylo (Bos et al., 2022).

Recently, Pedersen et al. (2024) proposed a general-purpose Neuomorphic IR (NIR). NIR shares a similar level of operand abstraction as our custom IR, which also encapsulates components of the neuron model into one single operation. The optimization of operand abstraction of IRs might improve SNN representation and mapping. For example, IRs with more primitive operations and explicit representation of the temporal dimension and spiking behavior might enable various compiler-level optimizations and straightforward mapping to neuromorphic architectures.”

Suppl. Table 8. Comparisons of the proposed IR, PyTorch IR, and the JAX IR

(a) Comparison of PyTorch, JAX, and our custom IRs for SNNs

Model	Pytorch	JAX	Ours
Spiking ResNet-18	28.5MB	48K	4KB
Spiking ResNet-50	141MB	65K	11KB
Spiking VGG-16	209MB	37K	3KB

(b) Comparison of PyTorch/JAX IRs between ANNs and SNNs

Model	ANN (PyTorch)	SNN (PyTorch)	ANN (JAX)	SNN (JAX)
ResNet-18	11KB	28.5MB	16K	48K
ResNet-50	26KB	141MB	48K	65K
VGG-16	8KB	209MB	14K	37K

2. *Interfacing the proprietary IR with NIR is a great idea and I strongly suggest it is done in this paper, not in a future study. This would make the authors claims with respect to modularity and ecosystem integration much stronger, and if the network descriptions in the NIR IR were made available with the paper, would allow direct comparisons between this work and other hardware-backends.*

Reply: Thank you for the constructive suggestion. We developed two Python-based IR conversion modules, *convert_ir_to_nir.py* and *convert_nir_to_ir.py*, to implement bidirectional conversion between our proprietary IR and NIR. Specifically, for NIR-to-IR conversion, the IR converted from the NIR framework (NIR version 1.0.6, <https://github.com/neuromorphs/NIR/tree/v1.0.6>) using *convert_nir_to_ir.py* is identical to the IR directly exported from our software toolchain for the same model (Suppl. Figure 10a); for IR-to-NIR conversion, the NIR converted from our IR using *convert_ir_to_nir.py* is also identical to the NIR directly exported from the NIR framework (Suppl. Figure 10b). Suppl. Figure 10 is shown below and added to the revised Supplementary Information. The two conversion modules *convert_ir_to_nir.py* and *convert_nir_to_ir.py* are available in our code repository through the link <https://1drv.ms/f/c/ba81c74852403b0e/IgDH2hgiUZJSoYkfpAvh-AARi2MLyDVABelVedP0j7KfM?e=ZVHcf8>.

We added this information in **Methods 4.3 SNN Compilation and Deployment on the Multi-core Neuromorphic Architecture**. We now state:

Line 1218: “To improve the cross-platform compatibility of our software toolchain, we developed two Python-based IR conversion modules, *convert_ir_to_nir.py* and *convert_nir_to_ir.py*, to implement bidirectional conversion between our custom IR and NIR. The results of conversion are determined by comparisons between the converted and the directly exported IRs using the Notepad++’s Compare plugin (Suppl. Figure 10).”

We added this information in **Results 2.6**. We now state:

Line 586: “The multi-core near-memory neuromorphic architecture was deployed on an FPGA platform (Figure 5a) to perform SNN training through our software toolchain (Figure 5c-d). Bidirectional conversion between our custom IR and general-purpose open-source Neuromorphic IR (NIR, Suppl. Figure 10) is developed to improve the cross-platform compatibility of the toolchain.”

We also added this information in **Discussion** and removed the description of further study on this. We now state:

Line 732: “The bidirectional conversion between our custom IR in our software toolchain and NIR, a recent general-purpose IR for neuromorphic computing (Pedersen et al., 2024), renders our software toolchain compatible with multiple hardware platforms.”

Suppl. Figure 10. Bidirectional conversion between our proprietary IR format and the NIR format. (a) The IRs of Spiking ResNet-18, -50, and Spiking VGG-16 models converted from the NIR framework (version 1.0.6) are identical to those IRs directly exported from our software toolchain. (b) The NIRs of those models converted from our IRs are identical to the NIRs directly exported from the NIR framework. “Identical” indicates that the data in the two files is an exact match, as determined by Notepad++’s Compare plugin.

3. I do not quite understand how the performance is determined for the Chip in Table 10. Could the authors please clarify their modeling assumptions? Was the RTL code synthesized? If so, what CMOS process was assumed? Is the achievable operating voltage/average power the same as reported on the FPGA?

Reply: We are sorry for the confusion. The RTL code for the architecture was synthesized using the Synopsys Design Compiler (syn-vL-2016.03-SP1, Synopsys Inc.) tool. Performance evaluation of the architecture was based on the TSMC N28HPC process under the

Supplementary Table 10.

Performance Comparison of the Architecture on FPGAs and Chip Taped Out

	FPGA	Chip
Cores	20	32
NoC Topology	2x10 (2D mesh)	4x8 (2D mesh)
Engine Clock(MHz)	150	500
Router Clock (MHz)	200	667
Inter-core Bandwidth(Gbps)	50 (within FPGA), 100 (inter- FPGAs, 2 core – 2 core)	170
Routing Strategy	X-Y Routing	X-Y Routing

tt_0.9V_25°C condition. During the performance evaluation, the engine and router clocks reach 500 MHz (clock cycle: 2 ns; Suppl. Figure 9a, c, e) and 667 MHz (clock cycle: 1.5 ns; Suppl. Figure 9g), respectively, with zero violating paths (Suppl. Figure 9b, d, f, h). The bitwidth of a channel of routers in our multi-core architecture is 256 bits, corresponding to the bandwidth 256 bits × 667 MHz = 170.8 Gbps. The FPGA with XCVU37P chip in our study was based on the 16nm process, operating under multiple voltages ranging from 0.85V to 2.5V, junction temperature at 48.8°C, ambient and PCB board temperatures at 25°C (Suppl. Figure 6b), which are different from our operating conditions of the Chip Taped-Out.

Suppl. Figure 9. The clock frequency setting and time closure report in the performance evaluation. (a) The clock frequency setting of the FP Engine. (b) Time closure report of the FP Engine. (c) The clock frequency setting of the BP Engine. (d) Time closure report of the BP Engine. (e) The clock frequency setting of the WG Engine. (f) Time closure report of the WG Engine. (g) The clock frequency setting of the Router. (h) Time closure report of the Router.

Supplementary Figure 6. FPGA block design and operating conditions. (a) Partial FPGA Block Design of Multi-Core Architecture. (b) FPGA operating condition report.

Suppl. Figure 9 and Suppl. Figure 6b were added to the revised **Supplementary Information**. We added this information in **Results 2.5 Area and Power Consumption Analysis**. We now state:

Line 538: “All reported data is based on TSMC N28HPC process under the tt_0.9V_25°C condition. During the performance evaluation, the engine and router clocks reach 500 MHz and 667 MHz, respectively (Suppl. Figure 8).”

4. Could the authors please clarify exactly what code will be released to the repository if the paper is accepted?

Reply: The codes will be released on GitHub if the paper is accepted. The codes include Verilog Netlist files of Multi-Core Neuromorphic Architecture, FPGA implementation netlist file of the architecture, software toolchain for SNN Compilation and Deployment on the neuromorphic architecture, software simulation tool, bidirectional conversion of NIR and our custom IR, and SNN models including Spiking ResNet-18, -50, and Spiking VGG-16. A detailed README.txt file contains instructions for running the software. Currently, these codes are also available through the Microsoft OneDrive link:

<https://1drv.ms/f/c/ba81c74852403b0e/IgDH2hgiUZJSoYkfpAvh-AARi2MLyDVABeIVedP0j7KfM?e=ZVHcf8>.

Reviewer #3 (Remarks on code availability): The repository is a placeholder. The authors claim code and data will be released upon publication of the associated paper.

Reply: Yes, the codes will be released on GitHub if the paper is accepted.

Replies to Reviewers

Reviewer #1:

In this paper, the authors propose a multi-core neuromorphic architecture to address the growing need for edge training in dynamically changing environments. The architecture supports Feedforward-Propagation, Back-Propagation, and Weight-Gradient operations, enabling parallel computing at both the engine and core levels. The authors claim that this study develops the first multi-core neuromorphic architecture supporting direct SNN training, facilitating neuromorphic computing in edge-learnable applications.

Comments for the Authors:

1. What is the rationale behind using Spiking-ResNet18, ResNet50, and VGG16 for simulation? Please provide a clear explanation for choosing these specific models.

Reply: Sincere thanks for the question. We selected Spiking-ResNet18, -ResNet50, and -VGG16 for simulation based on their representativeness in SNN models. Table 1 summarizes certain representative works published in top conferences (NeurIPS, CVPR, ICML, ICCV, ICLR, ECCV, AAAI) and journals (such as TPAMI) in the SNN field over the past five years. It can be seen that almost all convolutional SNNs adopt ResNet or VGG structures, among which spiking-ResNet18, -ResNet50, and -VGG16 are three common models. Regarding deep SNN simulation in recent neuromorphic architectures, Darwin3 (Ma et al., 2024) adopted spiking-VGG16 and -ResNet50. Given their representativeness in both SNN modeling and neuromorphic architecture simulation, we chose Spiking-ResNet18, -ResNet50, and -VGG16 in this study.

Table 1. Representative Convolutional Spiking Neural Networks

	Method	Year	Publication	Model Structure	FP16/BF16
1	PLIF	2021	CVPR [11]	Simple CNNs	n/a
2	SEW	2021	NeurIPS [21]	ResNet-18, -34, -50, -101, -152	n/a
3	BNTT	2021	Front. Neurosci. [31]	VGG-9, -11	n/a
4	STBP-tdBN	2021	AAAI [41]	ResNet-19, -34, -50	n/a
5	DSpike	2021	NeurIPS [51]	ResNet-18, -34; VGG-16	n/a
6	Diet-SNN	2021	TNNLS [61]	VGG-16; ResNet-20	n/a
7	Tandem	2021	TPAMI [71]	VGG-11, -16	n/a
8	Spiking ResNet	2021	TNNLS [81]	ResNet-18, -34, -50	n/a
9	TTRBR	2022	Neural Networks [91]	ResNet-18, -20, -32, -56, -110	n/a
10	LTL	2022	NeurIPS [101]	VGG-11, -16; ResNet-20	n/a
11	RecDis-SNN	2022	CVPR [111]	ResNet-19, -34; VGG-16	n/a
12	NDA	2022	ECCV [121]	VGG-11, -19; ResNet-19, -34	FP16
13	Temporal Pruning	2022	ECCV [131]	VGG16	n/a
14	TET	2022	ICLR [141]	ResNet-19, -34; VGGSNN	n/a
15	n/a	2022	AAAI [151]	VGG-16; ResNet-18, -20	n/a
16	QCFS	2022	ICLR [161]	VGG16; ResNet-18, -20, -34	n/a
17	DSR	2022	CVPR [171]	ResNet-18, -34; VGG-11	FP16
18	OTTT	2022	NeurIPS [181]	VGG-7; ResNet-34	FP16
19	LATS	2023	ICLR [191]	ResNet-18, -50	FP16
20	SML	2023	ICML [201]	ResNet-18, -19, -34; VGGSNN	FP16
21	PSN	2023	NeurIPS [211]	ResNet-18, -34; VGGSNN	FP16
22	SSF	2023	ICCV [221]	ResNet-18, -34; VGG-11	n/a

23	Dual-Phase	2023	TNNLS [231]	ResNet-18, -20, -34; VGG-16	n/a
24	Fast-SNN	2023	TPAMI [241]	VGG-11, 16; ResNet-18	n/a
25	Attention SNN	2023	TPAMI [251]	ResNet-18, -34, -104	FP16
26	MPBN	2023	ICCV [261]	ResNet-19, -20, -34; VGG-16	n/a
27	KDSNN	2023	CVPR [271]	ResNet-18; VGG-11	n/a
28	SLTT	2023	ICCV [281]	ResNet-18, -34, -50, -101; VGG-11	FP16
29	LocalZO	2023	NeurIPS [291]	ResNet-19, -34; VGG-SNN	n/a
30	IM-LIF	2024	IEEE TETCI [301]	ResNet-19; VGG-SNN	n/a
31	Parameter Calibration	2024	IJCV [311]	VGG-16; ResNet-20, -34	n/a
32	LM-H	2024	ICLR [321]	ResNet-18, -34, -50; VGG-16	FP16
33	BKDSNN	2024	ECCV [321]	ResNet-19, -34, -50	FP16
34	T-RevSNN	2024	ICML [331]	ResNet-18	FP16
35	n/a	2024	CVPR [341]	ResNet-34, -50, -19, -20	n/a
36	FSTA-SNN	2025	AAAI [351]	ResNet-18, -34	FP16

We added Table 1 to the revised **Supplementary Information** as Suppl. Table 12 and also added its References. We supplemented the rationale for selecting these SNNs in simulations in **Results 2.3 Data Flow and Parallel FP-BP-WG Computations** in the revised manuscript. We now state:

Lines 396: “The three SNNs were selected based on their representativeness (Suppl. Table 12) and adoption in previous neuromorphic architecture studies (Ma et al., 2024).”

[Ma et al., 2024] Ma, D. et al. Darwin3: a large-scale neuromorphic chip with a novel isa and on-chip learning. Natl. Sci. Rev. 11, nwae102 (2024).

2: Compared to the A100, the DRAM occupancy of your architecture was reduced. Please elaborate on these findings and their significance.

Reply: Sincere thanks for the question. Table 2 provides a detailed comparison of DRAM occupancy during SNN training between our multi-core architecture and the A100 GPU under different conditions. DRAM occupancy primarily consists of four components: intermediate data (including spikes, membrane potentials, optimizer states, weights, and weight gradients).

Table 2. Comparisons of DRAM Occupancy during SNN Training

Model	Batch size	Architecture	Total Occupancy (GB)	Intermediate data (GB)	Optimizer state (GB)	Weight (GB)	Weight gradient(GB)	
S-ResNet18	16	A100	2.01	1.45	0.42	0.07	0.07	
		Ours	1.17 (-41.57%)	0.54 (-62.43%)	0.42	0.14	0.07	
	32	A100	3.46	2.90	0.42	0.07	0.07	
		Ours	1.72 (-50.31%)	1.09 (-62.40%)	0.42	0.14	0.07	
	64	A100	6.35	5.79	0.42	0.07	0.07	
		Ours	2.81 (-55.85%)	2.18 (-62.39%)	0.42	0.14	0.07	
	128	A100	12.15	11.59	0.42	0.07	0.07	
		Ours	4.99 (-58.96%)	4.36 (-62.39%)	0.42	0.14	0.07	
	S-ResNet50	16	A100	8.79	6.61	1.63	0.27	0.27
			Ours	4.87 (-44.62%)	2.42 (-63.39%)	1.63	0.54	0.27
		32	A100	15.4	13.22	1.63	0.27	0.27
			Ours	7.28 (-52.69%)	4.78 (-63.82%)	1.63	0.54	0.27
64		A100	28.62	26.44	1.63	0.27	0.27	
		Ours	12.12 (-57.65%)	9.68 (-63.41%)	1.63	0.54	0.27	
128		A100	55.06	52.89	1.63	0.27	0.27	
		Ours	21.79 (-60.42%)	19.34 (-63.43%)	1.63	0.54	0.27	
S-VGG16		16	A100	10.34	8.28	1.55	0.26	0.26
			Ours	4.15 (-59.81%)	1.83 (-77.89%)	1.55	0.52	0.26
		32	A100	18.62	16.56	1.55	0.26	0.26
			Ours	5.99 (-67.82%)	3.68 (-77.80%)	1.55	0.52	0.26
	64	A100	35.17	33.12	1.55	0.26	0.26	
		Ours	9.66 (-72.53%)	7.34 (-77.84%)	1.55	0.52	0.26	
	128	A100	68.28	66.25	1.55	0.26	0.26	
		Ours	17.01 (-75.09%)	14.69 (-77.83%)	1.55	0.52	0.26	

As shown in Table 2, the DRAM occupancy of our architecture during SNN training is reduced by 41.57%~75.10% compared to the A100, and this reduction is mainly due to the decrease in the DRAM occupancy of intermediate data, which accounts for 72.04%~97.03% of the total DRAM occupancy in A100 GPU. Notably, our architecture reduces this part of occupancy by 62.43%~77.89% compared to A100. In contrast, the DRAM occupancy of weight gradients and optimizer states

remains consistent with that of A100. We adopt dual storage of weights to facilitate parallel forward and backward computation during SNN training. As DRAM occupancy of weights only accounts for 1~3% of the A100's DRAM occupancy, increase of the occupancy caused by the dual storage has little impact on the overall DRAM occupancy during SNN training. The DRAM occupancy reduction significantly alleviated the DRAM storage requirements, facilitating model training in storage constrained conditions

We added Table 2 to the revised **Supplementary Information** as Suppl. Table 4 and elaborated the findings in **Result 2.3 Data Flow and Parallel FP-BP-WG Computations**. We now state:

Line 437: “The above reduction was mainly due to the decrease in the occupancy of intermediate data accounting for 72.04%~97.03% of the total DRAM occupancy of A100 GPU. Our architecture reduced this part of occupancy by 62.43%~77.89% (see Suppl. Table 4 for details).”

Furthermore, we also discussed the advantages of this reduced DRAM occupancy in **Discussion**. We now state:

Line 681: “which ... significantly alleviated the DRAM storage requirements, facilitating the training in storage constrained conditions, ...”

3. The RTL power analysis indicates that your sparse design reduces the energy consumption of the computing core during training. Please explain this result in more detail.

Reply: We sincerely thank you for the suggestion. Table 3 shows detailed sparsity information of SNN models, and Table 4 shows detailed energy consumption information during SNN training with and without sparse computation optimization.

Table 3. Sparsity of SNN Models

Models	Spike sparsity	Spike16 sparsity	fire' sparsity	$\forall u$ sparsity
S-ResNet18	90%	31%	68%	7%
S-ResNet50	80%	12%	54%	0%
S-VGG9	88%	25%	56%	62%

*The spike16 sparsity means portion of all 16 channels of spikes are zero in FP or WG computation.

We added Tables 3 and 4 to the revised **Supplementary Information** as Suppl. Tables 5 and 6, respectively. We added the sparsity and energy consumption information (in blue) to explain this result to **Results 2.4 Sparse Computation Optimization**. We now state:

Table 4. Effects of Sparse Computation Optimization

Models	Conditions	FP (mW)	BP (mW)	WG (mW)	Total (mW)
S-ResNet18	no optimization	64.39	187.66	39.20	291.24
	with optimization	37.01	60.44	15.63	113.15
	Power decrease (%)	42.53%	67.74%	60.16%	61.09%
S-ResNet50	no optimization	64.38	187.66	39.20	291.24
	with optimization	46.95	86.63	21.13	154.71
	Power decrease (%)	27.01%	53.92%	46.24%	46.81%
S-VGG9	no optimization	64.38	187.66	39.20	291.24
	with optimization	40.45	82.89	17.05	140.39
	Power decrease (%)	37.11%	55.75%	56.52%	51.69%

Line 494: “The RTL power analysis showed that our sparse design could significantly reduce the energy consumption of the computing core during spiking-ResNet18, -ResNet50, and -VGG9 training by 61.09%, 46.81%, and 51.69%, respectively (Figures 4d-f), including 42.53%, 27.01%, 37.11% of reductions in the FP engine leveraged on the spike and spike16 sparsity of these models (Suppl. Table 5), 67.74%, 53.92%, 55.75% of reductions in the BP engine leveraged on the fire', and $\forall u$ sparsity (Suppl. Table 5), and 60.16%, 46.24%, 56.52% of reductions in the WG engine leveraged on the spike, spike16, and $\forall u$ sparsity (Suppl. Table 5). Suppl. Table 6 provides detailed energy consumption information during SNN training with and without sparse computation optimization.”

4: Clearly state the objective of this research study. Additionally, highlight what has already been done in this area and what new contributions your work brings, along with their scientific benefits.

Reply: Sincere thanks for the comment. We modified **Introduction** to add a more detailed research background, clearly state the objective of our study, and highlight our key contributions. We added the information on scientific benefits in **Discussion**. We now state:

- Research background

Line 74: “Neuromorphic processors supporting SNN computation have been developed quickly, such as TrueNorth (Merolla et al., 2014), Loihi (Davies et al., 2018, 2021), Tianjic (Pei et al., 2019), BrainScale (Pehle et al., 2022; Cramer et al., 2022), Neurogrid (Benjamin et al., 2014), DYNAPs (Moradi et al., 2017), Darwin (Ma et al., 2017, 2024). These neuromorphic architectures commonly adopt a 2D Mesh multi-core structure, whose computing powers are generally below 1000 GOPS with computational precision from 1 bit to 9 bits (Alam et al. 2024). Early architectures are designed mainly for brain simulation using sparsely activated neural networks with certain connection randomness (e.g., SpiNNaker, 2013; TrueNorth, 2014). Recent architectures (Loihi 2018; Loihi2 2021; Darwin3 2024; etc.) also support more structured networks such as spiking convolutional networks with higher computation intensity, while Tianjic (2019) supports ANN/SNN hybrid neural network models. For on-chip learning capabilities, TrueNorth (2014) only supports forward inference, DYNAPs (2017) supports Hebbian learning, Loihi (2018) enables two-factor local learning (e.g., STDP), and Loihi2 (2022) and Darwin3 (2024) introduce more complex three-factor local learning (e.g., RSTDP, SDSP). However, these local learning methods only support shallow network structure optimization (Meng et al., 2022). BP-based direct training of deep SNNs has not been achieved on existing multi-core neuromorphic architectures.”

- Objective of the study

Line 124: “To overcome the limited learning ability of existing neuromorphic architectures, this study aims to develop a highly efficient multi-core neuromorphic architecture supporting BP-based direct training of deep SNNs.”

- Contributions

Line 130: “Our contributions include:

(1) To the best of our knowledge, we proposed the first multi-core, neuromorphic architecture enabling deep SNN training based on BP. It introduces an instruction set, highly parallelable computing cores with Feedforward-Propagation (FP), Back-Propagation (BP), and Weight-Gradient (WG) engines, and a highly scalable 2D-mesh Network-on-Chip (NoC) specifically tailored for SNN training, which enables distributed deployment of the entire deep SNN model across computing cores to perform training. Along with our software toolchain, our architecture has been implemented on FPGAs and achieved successful supervised learning and federated learning.

(2) We developed an efficient strategy for parallelized SNN training operations in the multi-core architecture, achieving highly parallel computing at both the core and engine levels. It enables parallel execution of FP, BP, and WG computations on different samples, model layers, and time steps simultaneously across computing cores.

(3) We implemented single-bit spike storage, addition-based spiking computation, multiple dataflows within computing engines, the inter- and intra-core parallelism, and sparse computation optimization in our multi-core architecture, improving its computational efficiency and reducing DRAM occupancy and access significantly during SNN training.”

- Scientific benefits

Line 719: “Together, our study extends multi-core neuromorphic computing architectures in global learning capability, efficient parallel computing, DRAM usage, and sparse computation optimization, facilitating their applications, such as those in edge-learnable environments.”

5: Consider adding an algorithm or pseudocode to illustrate the design and functionality of the multi-core neuromorphic architecture for SNNs.

Algorithm 1 Design of Parallelized SNN Training Operations in the Multi-core Neuromorphic Architecture

```

1:  $N, L;$  % N: batch size; L: SNN layer number  $l \in \{0, \dots, L-1\};$  % Layer index
2:  $T;$  % Total number of time steps  $t \in \{0, \dots, T-1\};$  % Index of a time step
3:  $Core\_Num = 32;$  % Total number of computing cores in the architecture  $k \in \{0, \dots, Core\_Num-1\};$  % Index of a computing core
4:  $Engine\_Num = 3;$  % Total number of computing engines in a core  $e_i \in \{0, \dots, Engine\_Num-1\};$  % Index of a computing engine
5:  $n = \text{zero}(1, Core\_Num);$  % Indices of samples processed in FP engines  $n\_bp = \text{zero}(1, Core\_Num);$  % Indices of samples processed in BP engines
6:  $n\_wg = \text{zero}(1, Core\_Num);$  % Indices of samples processed in WG engines  $\alpha;$  % Leakage factor

7: parallel for  $k = 0 : Core\_Num - 1$  do % parallel computation across computing cores
8:   parallel for  $e_i = 0 : Engine\_Num - 1$  do % parallel computation across computing engines
9:     if  $e_i == 0$  % computation in the FP engine
10:      Read  $w_k^l$  from SRAM; %  $w_k^l$ : weights of layer  $l$  in core  $k$ 
11:      for  $t = 0 : T - 1$  do % forward computation along time steps
12:        While  $\sim\text{exist}(s_{n[k],t}^{l-1})$  wait; % wait until spike input from layer  $l-1$  arrives
13:        Read  $s_{n[k],t}^{l-1}$  from SRAM %  $s_{n[k],t}^{l-1}$ : spikes input from layer  $l-1$  at time step  $t$  of sample  $n[k]$ 
14:        Conv_FP $_{n[k],t}^l = \sum s_{n[k],t}^{l-1}[i]w_k^l[j,i];$  % convolution computation between spike input from layer  $l-1$  and weights of layer  $l$ 
15:        if  $t == 0$   $u_{n[k],t-1}^l[i] = 0;$  endif % initialize membrane potential when  $t = 0$ 
16:         $u_{n[k],t}^l[i] = \alpha u_{n[k],t-1}^l[i](1 - s_{n[k],t-1}^l[i]) + \text{Conv\_FP}_{n[k],t}^l[i];$  % update the membrane potential at time step  $t$ 
17:         $s_{n[k],t}^l[i] = \text{fire}(u_{n[k],t}^l[i]);$  % generate spike output of layer  $l$  at time step  $t$ 
18:        Send  $s_{n[k],t}^l$  to current FP engine; % data dispatching for upcoming computation of Conv_FP of layer  $l$  in next time step
19:        Send  $s_{n[k],t}^l$  to another FP engine; % data dispatching for upcoming computation of Conv_FP of layer  $l+1$  in another core
20:        Send  $s_{n[k],t}^l, u_{n[k],t}^l$  to BP and WG engines in the same core or DRAM; % data dispatching for upcoming BP and WG computation
21:      endfor % end of FP computation across time steps
22:      if  $l == L - 1$  and  $t == T - 1$  calculate error  $E[n[k]];$  endif % computing calculate error when FP computation in the last layer is finished
23:      if  $n[k] < N - 1$   $n[k] = n[k] + 1;$  goto Line 11; endif % update sample index and go to Line 11 to process the next sample
24:    endif % end of computation in the FP engine

25:   if  $e_i == 1$  % computation in the BP engine
26:     Read  $w_k^{l+1}$  from SRAM % read weights of layer  $l+1$  to BP Array in core  $k$ 
27:     for  $t = T - 1 : 0$  do % backward computation along time steps
28:       While  $\sim\text{exist}(\nabla u_{n\_bp[k],t}^{l+1})$  wait; % wait until membrane potential gradients of layer  $l+1$  arrive
29:       Read  $\nabla u_{n\_bp[k],t}^{l+1}$  from SRAM; %  $\nabla u_{n\_bp[k],t}^{l+1}$ : membrane potential gradients of layer  $l+1$  at time step  $t$  of sample  $n\_bp[k]$ 
30:       Read  $s_{n\_bp[k],t}^l, u_{n\_bp[k],t}^l$  from SRAM;
31:       Conv_BP $_{n\_bp[k],t}^l = \sum \nabla u_{n\_bp[k],t}^{l+1}[j]w_k^{l+1}[i,j];$  % convolution computation between membrane potential gradients and weights from layer  $l+1$ 
32:       calculate  $\text{fire}'(u_{n\_bp[k],t}^l[i]);$  % calculate the surrogate gradient of the spike function
33:       if  $\text{fire}'(u_{n\_bp[k],t}^l[i]) \neq 0$  %  $\nabla s_{n\_bp[k],t}^l[i]$ : spike gradients of layer  $l$  at time step  $t$  of the sample  $n\_bp[k]$ 
34:          $\nabla s_{n\_bp[k],t}^l[i] = -\alpha \nabla u_{n\_bp[k],t+1}^l[i]u_{n\_bp[k],t}^l[i] + \text{Conv\_BP}_{n\_bp[k],t}^l[i];$  % compute the spike gradients of layer  $l$  at time step  $t$ 
35:          $\nabla u_{n\_bp[k],t}^l[i] = \alpha \nabla u_{n\_bp[k],t+1}^l[i](1 - s_{n\_bp[k],t}^l[i]) + \nabla s_{n\_bp[k],t}^l[i]\text{fire}'(u_{n\_bp[k],t}^l[i]);$ 
36:       else  $\nabla u_{n\_bp[k],t}^l[i] = \alpha \nabla u_{n\_bp[k],t+1}^l[i](1 - s_{n\_bp[k],t}^l[i]);$  % compute membrane potential gradients of layer  $l$  at time step  $t$ 
37:       endif
38:       Send  $\nabla u_{n\_bp[k],t}^l$  to current BP engine; % data dispatching for upcoming computation of the gradients at time step  $t-1$ 
39:       Send  $\nabla u_{n\_bp[k],t}^l$  to WG engine in another core; % data dispatch for upcoming computation of weight gradients of layer  $l$ 
40:       Send  $\nabla u_{n\_bp[k],t}^l$  to BP engine in another core; % data dispatching for upcoming computation of Conv_BP of layer  $l-1$ 
41:     endfor % end of BP computation across time steps
42:     if  $n\_bp[k] < N$   $n\_bp[k] = n\_bp[k] + 1;$  goto Line 27; endif % update sample index and go to Line 27 to process the next sample
43:   endif % end of computation in the BP engine

44:   if  $e_i == 2$  % Computation in the WG engine
45:     for  $t = T - 1 : 0$  do % backward computation along time steps
46:       While  $\sim\text{exist}(\nabla u_{n\_wg[k],t}^{l+1})$  wait; % wait until membrane potential gradients of layer  $l+1$  arrive
47:       Read  $\nabla u_{n\_wg[k],t}^{l+1}$  from SRAM;
48:       Read  $s_{n\_wg[k],t}^l$  from SRAM;
49:        $\nabla w^{l+1}[i,j] = \sum_t \nabla u_{n\_wg[k],t}^{l+1}[j]s_{n\_wg[k],t}^l[i];$  % integrate weight gradients of layer  $l+1$  across time steps
50:     endfor % end of WG computation across time steps
51:      $\nabla w^{l+1} = \nabla w^{l+1} / T;$  % generate weight gradients of layer  $l+1$ 
52:     if  $n\_wg[k] < N$   $n\_wg[k] = n\_wg[k] + 1;$  goto Line 45; % update sample index and go to Line 45 to process the next sample
53:     else  $w^{l+1} = w^{l+1} + \text{d}w^{l+1};$  % Update weights of layer  $l+1$  after processing a batch of samples
54:     endif
55:   endif % end of computation in the WG engine
56: endfor % end of parallel for  $e_i = 0 : Engine\_Num - 1$ 
57: endfor % end of parallel for  $k = 0 : Core\_Num - 1$ 

```

Reply: Sincere thanks for the very important suggestion. We added an algorithm to describe the parallelized operations for one batch of training in the multi-core neuromorphic architecture (Algorithm 1). During SNN training, an SNN model is deployed in a distributed manner across all computing cores of our

multi-core architecture, with one layer or a segment of a layer allocated to an engine within a computing core. Within the same core, without loss of generality, the FP engine computes the spike outputs and membrane potentials of layer l , and the BP engine calculates the membrane potential gradients of layer l , while the WG engine computes the weight gradients of layer $l+1$. Prior to training, all sample indices are set to 0 for all engines (Lines 5-6), indicating that all engine computations start from the first sample. Our architecture supports both inter-core parallelism (Line 7) and intra-core engine parallelism (Line 8) computations:

The FP engine computes spike outputs and membrane potentials of layer l neurons (Lines 14-17) along time steps (Line 11), after ensuring correct spike inputs from layer $l-1$ (Line 12). The computed results are dispatched (Lines 18-20) for subsequent FP computations of layer l at the next time step (Line 18), for FP computations of layer $l+1$ at the current time step (Line 19), and for future BP/WG computations (Line 20). After completing computations along all time steps, the FP engine processes the next sample (Line 23). The architecture supports FP computations of different samples (Line 23), layers (Line 13), and time steps (Line 11) simultaneously across computing cores (Line 7).

The BP engine calculates membrane potential gradients of layer l (Lines 31-36) in reverse time order (Line 27), after ensuring correct potential gradient inputs from layer $l+1$ (Line 28). The computed results are dispatched (Lines 38-40) for upcoming BP calculations of layer l at the preceding time step (Line 38), for WG computations of layer l (Line 39), and for BP computations of layer $l-1$ at the current time step (Line 40). Upon completing computations along all time steps, the BP engine proceeds with the next sample (Line 42). Within the same core, FP and BP engines compute spike outputs, membrane potentials (Lines 16-17) and membrane potential gradients (Lines 35-36) of the same model layer (l) across different samples and time steps simultaneously (Line 8). The architecture also supports BP computations of different samples (Line 42), layers (Line 30), and time steps (Line 27) simultaneously across computing cores (Line 7).

The WG engine computes weight gradients of layer $l+1$ in reverse time order (Line 45), after ensuring correct potential gradient inputs from layer $l+1$ (Line 46). Upon completing computations along all time steps, the WG engine processes the next sample (Line 52). Within the same core, the WG engine computes weight gradients of layer $l+1$ (Line 49) in parallel (Line 8) with the computations in the FP (Lines 16-17) and BP (Lines 35-36) engines. The architecture also supports WG computations of different samples (Line 52), layers (Line 47), and time steps (Line 45) simultaneously across computing cores (Line 7).

Together, our multi-core architecture supports parallel execution of FP, BP, and WG computations across different samples, model layers, and time steps simultaneously in different cores. Within the same core, FP and BP engines compute spike outputs, membrane potentials and membrane potential gradients of the same SNN layer (l) across different samples and time steps simultaneously, and WG engine computes weight gradients of layer $l+1$ in parallel with the computations in the BP and FP engines. Thus, the architecture supports highly parallel computing at both the core and engine levels during SNN training.

Due to limited space, we added the Algorithm 1 and its description to the revised **Supplementary Information** as Suppl. Algorithm 1. We described the parallel computations in our architecture in **Results 2.3 Data Flow and Parallel FP-BP-WG**. We now state:

Line 353: “During SNN training, an SNN model is deployed in a distributed manner across all computing cores of our multi-core architecture, with one layer or a segment of a layer allocated to an engine within a computing core. Suppl. Algorithm 1 shows parallelized operations in our multi-core architecture during SNN training. At the core level, our multi-core architecture supports parallel execution of FP, BP, and WG computations across different samples, model layers, and time steps simultaneously in different cores. Within the same core, FP and BP engines compute spike outputs, membrane potentials and membrane potential gradients of the same model layer across different samples and time steps simultaneously, and WG engine computes weight gradients of the next layer in parallel with the computations in the FP and BP engines (see details in the description for Suppl. Algorithm 1). Thus, our architecture supports highly parallel computing

at both the core and engine levels for SNN training.”

6: Please discuss the findings presented in Table 1 in greater detail. What insights can be drawn from the data?

Reply: Sincere thanks for this question. In the revised table, we classified previous architectures into neuromorphic architecture and general AI architectures two categories to make comparisons clearer. We also added details of on-chip learning, architecture topology, Loihi, Jetson AGX Orin 64G information (in blue) to the revised table (Table 5 in this document). We replaced the original Table 1 with the revised table, and discussed the findings in greater detail (in blue) in **Results 2.5 Area and Power Consumption Analysis**. We now state:

Line 537: “Table 1 shows the performance comparison between our 32-core near-memory neuromorphic architecture and representative architectures such as neuromorphic chips and GPUs. The computation capacity of our 32-core architecture (16 TFLOPS@FP16) was clearly higher than that of prior neuromorphic architectures (typically ≤ 1 TOPS with computational precisions 1-9 bit). Notably, only our multi-core architecture supported BP-based SNN training, while prior multi-core neuromorphic architectures only supported inference and local learning. A recent study proposed SATA (Yin et al., 2023), which was only a single-core SNN training architecture and lacked hardware implementation. Compared with general AI architectures such as GPUs, a commonality was that both can support SNN training. Our architecture achieved an energy efficiency of 1.05 TFLOPS/W @ FP16 @ 28nm, which was comparable to the A100, DOJO D1, and Graphcore with 7nm technology, demonstrating the high energy efficiency of our architecture.”

Table 5. Comparison between Our Work and Representative Architectures

Items	Neuromorphic Architecture							General AI Architecture				This Work
	Loihi	Loihi2	TianjiC	True North	Brain Scale2	Darwin3	SATA	Jetson AGX Orin 64GB	Graph core	DOJO	GPU A100	
Process	14 nm	7 nm	28 nm	28 nm	65 nm	22 nm	65 nm	8 nm	7 nm	7 nm	7 nm	28 nm
Network	SNN	SNN	ANN/SNN	SNN	SNN	SNN	SNN	ANN/SNN	ANN/SNN	ANN	ANN/SNN	SNN
Frequency (MHz)	Async	Async	300	Async	-	333	500	1300	1850	2000	1410	500
On-chip Memory (MB)	32	24	14.44	260	0.08	-	4	-	368	440	88.38	64.78
Topology	8×16	8×16	12×13	64×64	2×2	24×24	-	-	-	18×20	-	4×8
Area (mm ²)	60	31	14.44	430	65	358.53	-	-	832	645	826	218.63
Power (W)	0.11	-	0.95	0.065	0.2	1.8	-	15-60	300	400	400	14.49
Peak Performance	0.03 TOPS	-	1.214 TOPS	0.058 TSOPS	-	-	-	85 TFLOPS	250 TFLOPS	362 TFLOPS	312 TFLOPS	16 TFLOPS
Precision	1-9 bit	1-9 bit	8 bit	1 bit	6 bit	1/2/4/8/16 bit	8 bit	FP16	FP/BF16	FP16	FP/BF16	FP16
Energy Efficiency	0.27 TOPS/W	-	1.278 TOPS/W	0.89 TSOPS/W	-	-	-	-	0.83 TFLOPS/W	0.91 TFLOPS/W	0.78 TFLOPS/W	1.05 TFLOPS/W
On-chip Learning	Two-factor Local Learning	Three-factor Local Learning	No	No	Two-factor Local Learning	Three-factor Local Learning	BP	BP	BP	BP	BP	BP

7: Efficient communication between cores is critical. Can you elaborate on how your proposed architecture addresses inter-core communication challenges?

Reply: Sincere thanks for the important question. During SNN training, an SNN model is distributed across all computing cores in our multi-core architecture, with one layer or a segment of a layer allocated to an engine within a computing core. Challenges of inter-core communication including: 1) Data for transmission includes both bit-based spike data and floating-point membrane potential and its gradient data, far more than data

transferred in inference, resulting in high demand in transmission bandwidth; 2) Complex data transmission along both model layers and time steps in both sequent and reverse orders during SNN training required accurate data transmission in both spatial and temporal dimensions.

To address these challenges, our inter-core communication design focuses on the following aspects:

(1) Unified Packet Design.

To support temporal dependency across cores, data of a single time step was organized as a message (Figure 1a) to avoid mixing data from different time steps. The message was split into 2KB packets. Both floating-point and bit-based spiking data in SNN training were transferred with a unified packet format. Packets were transmitted in 32B flits, with each flit carrying 16 FP16 floats or

Figure 1. Transmission data Packet design. (a) The structure of messages, packets, and head flits designed for NoC in the multi-core architecture. (b) The function description of each field in the head flit.

256 single-bit spike data. The detailed information in packet header is shown in Figure 1b

(2) 2D-Mesh Topology and Router Design.

Our multi-core architecture employs a 4×8 2D-Mesh topology (Figure 2a), with each node comprising a router and a computing core. Each router features six channels (Figure 2c): four for east, west, north, and south inter-node communication, and two dedicated channels for the FP sub-core and BP sub-core within the computing core to increase bandwidth of data transmission during FP and BP computations. Each router direction supports two virtual channels (VCs), with incoming data stored in corresponding FIFOs (Figure 2b). The Route Computation (RC) unit reads the head flit from the FIFO, parses the head flit, and

determines the next transmission direction using the XY routing strategy by comparing the target X and Y coordinate information ($target_x, target_y$) in the head flit with the router's coordinates (x, y ; Figure 2b). If the router is the target router during transmission, it uses information in the target_befe field of head flit to select the FE or BE output

Figure 2. The multi-core neuromorphic architecture. (a) The 32-core architecture with 4×8 2D-Mesh topology. (b-d) Router Architecture Overview. (b) Internal structure of BW and RC. (c) Overall Router structure. (d) Internal structure of SA and ST. (e) Globally Asynchronous Locally Synchronous (GALS) Network-on-Chip (NoC) Design. All routers share an identical 667 MHz global clock (thin red lines), with individual computing cores synchronized via local 500 MHz clock (thin blue lines). Thick red lines and blue lines indicated data transmission at 667 MHz and 500 MHz, respectively.

port (Figure 2b). When multiple packets from different directions request the same output port, the Switch Allocation (SA) module employs a round-robin arbitration policy to prioritize transmission (Figure 2d). The Switch Traversal (ST) module transfers the packet flit-by-flit based on back-pressure from the downstream unit (Figure 2d). The 2D-Mesh topology has been commonly adopted in previous multi-core neuromorphic architectures.

(3) Globally Asynchronous Locally Synchronous (GALS) Communication Mechanism (Figure 2e).

The architecture adopts a globally asynchronous design with separate clock frequencies for computing cores and routers: all routers share a common clock (Router-clk, 667MHz), while each core has an independent clock (core-clk, 500MHz). Asynchronous FIFOs are implemented at the router-core interface to support cross-clock domain data transfer. The GALS design avoids clock skew issues in global clock tree designs and improves scalability of our architecture. The higher router clock frequency increases data transmission bandwidth to 170Gbps/channel.

(4) Data Transmission Modules.

We also designed Dispatch Unit (DU) and Network Interface (NI) module to support inter-core transmission for multiple types of data. The DU (Figure 3a) serves as a data distribution hub between SRAMs in sub-cores and NI or DMA. The NI Transmitter (Figure 3b) fetches data payloads from SRAMs via DU and encapsulates them into packets, and performs flow control of data transmission via router. The NI Receiver (Figure 3b) performs decapsulation of packets, and manages their address mapping in SRAMs/DRAMs.

Figure 3. Data Transmission Modules. (a) The Dispatch Unit (DU) modules in the FP sub-core (left) and the BP sub-core (right), which serve as a data distribution hub for data transmission with SRAMs in these sub-cores via NI or DMA. (b) Network Interface (NI) module.

Figure 4. Inter-core data transmission pathways in the multi-core architecture. Green lines represent the data transmission pathway in forward computation, while red lines represent the pathway in backpropagation.

(0) Data Transmission Pathways in SNN Training.

To support SNN training, the multi-core architecture includes three primary on-chip data transmission channels: 1) FP-FP transmission transferring the output spike s of a layer to the FP sub-core processing the next layer; 2) BP-BP transmission transferring the ∇u of a layer to the BP sub-core processing the previous layer; 3) FP-BP transmission transferring the spike

s and membrane potential u obtained in forward computation to BP sub-cores through the path within the same computing core: source FP sub-core SRAM \rightarrow DU \rightarrow NI \rightarrow Router-p NI \rightarrow DU \rightarrow target BP sub-core SRAM (Figure 4). The inter-core communications support the FP-FP (or BP-BP) transmission through the paths: source FP (or BP) sub-core SRAM \rightarrow DU \rightarrow NI \rightarrow Router \rightarrow target core Router-p NI \rightarrow DU \rightarrow target FP (or BP) sub-core SRAM in our architecture (Figure 4).

Through (1)-(5), we are able to address inter-core communication challenges, allowing high bandwidth of transmission of both bit-based spike data and floating-point membrane potential and gradient data in SNN training. Coordination of modules in these data transmission pathways ensures accurate inter-core communication spatially and temporally across model layers and time steps in both sequent and reverse orders in SNN training.

We added Figures 2a, 2e, and 4 to Suppl. Figure 1 in **Supplementary Information**, and added Figures 1 and 3 to Suppl. Figure 3. We added Figures 2b-d to Figure 2 in the main content. We added the above detailed description of inter-core communication to the **Supplementary Information** as Supplementary Contents 1.1 and revised the description of inter-core communication (in blue) in our multi-core architecture in **Results 2.2 Multi-Core Near-memory Neuromorphic Architecture for SNN Training**. We now state:

Line 266: “Our multi-core architecture employs a 4×8 2D-Mesh topology (Suppl. Figure 1a) using the XY routing strategy, with each node comprising a router and a computing core (Suppl. Figure 1a). Each router features six channels: four for east, west, north, and south inter-node communication, and two dedicated channels for the FP sub-core and BP sub-core within the computing core (Figure 2f). To support temporal dependency across cores, data of a single time step is organized as a message (suppl. Figure 3b-c) to avoid mixing data from different time steps. The message is split into packets. We designed a unified packet format and modules such as Dispatch Unit (DU) (Suppl. Figure 3a), Network Interface (NI) (Suppl. Figure 3d), and DMA to support multiple types of data transmission (See details in Supplementary Contents 1.1). The DU module serves as a data distribution hub for data transmission with SRAMs in sub-cores via NI or DMA. NI module performs encapsulation and decapsulation of packets and conducts flow control of data transmission via the router. The Controller unit in each sub-core monitors its working status and coordinates the orderly operation of modules in the sub-core. We implemented clock domain isolation to decouple the working frequency of Routers and computing cores, with the computing core operating at 500 MHz and the Routers operating at a higher frequency of 667 MHz in the RTL simulation, achieving a higher data transmission bandwidth of 170Gbps. This Globally Asynchronous Locally Synchronous (GALS) communication mechanism (Suppl. Figure 1c) also avoids clock skew issues in global clock tree designs and improves scalability of our multi-core architecture. The 32-core neuromorphic architecture achieves 16 TFLOPS@ FP16 @ 500 MHz and contains 64.78MB SRAM (Suppl. Table 3).

To support SNN training, the multi-core architecture includes three primary on-chip data transmission channels: 1) FP-FP transmission transferring the spike output s of a layer to the FP sub-core processing the next layer; 2) BP-BP transmission transferring the output v_{uu} of a layer to the BP sub-core processing the previous layer; 3) FP-BP transmission transferring the spike s and membrane potential u obtained in forward computation to BP sub-cores through the path within the same computing core: source FP sub-core SRAM -p DU -p NI -p Router \rightarrow NI -p DU -p target BP sub-core SRAM. The inter-core communications support the FP-FP (or BP-BP) transmission through the paths: source FP (or BP) sub-core SRAM -p DU -p NI -p Router -p target core Router \rightarrow NI -p DU -p target FP (or BP) sub-core SRAM in our architecture (Suppl. Figure 1b). Coordination of modules in these data transmission pathways ensures accurate intra- and inter-core communication spatially and temporally across model layers and time steps in both sequent and reverse orders in SNN training.”

8: *SNNs use spike-based communication, which makes backpropagation more complex and computationally expensive. How does your study address this issue? Additionally, how do you efficiently parallelize training operations (e.g., FP, BP, WG) across multiple cores without compromising accuracy?*

Reply: Sincere thanks for the important questions. To address the challenges in SNN backpropagation:

(1). In the computation aspect, we specifically developed the BP and WG engines to compute membrane potential gradients and weight gradients during backpropagation, respectively. The custom Grad module (Figure 2c) in the BP engine calculates the spike gradients V_s , the surrogate gradient $fire'$, and V_u in SNN backpropagation. To improve energy efficiency, we also developed sparse computation optimization by fully leveraging the sparsity of spikes, $fire'$ and V/u in BP and WG computations, achieving 53~67% of reduction in energy consumption in the BP engine, and 46~60% of the reduction in the WG engine.

(2). In the communication aspect, we achieve high bandwidth of transmission of both bit-based spike data and floating-point membrane potential and gradient data in SNN training and accurate communication spatially and temporally across model layers and time steps through unified packet design, 2D-mesh topology of NoC and router design, Globally Asynchronous Locally Synchronous (GALS) communication mechanism, and custom designed data transmission modules and pathways (see details in **Reply to question 7**). The BP-BP and FP-BP data transmission pathways support the data transmission along both model layers and time steps in reverse orders in SNN backpropagation.

We summarized the information of the computation aspect to support SNN training in **Discussion**. We now state:

Line 624: "... This architecture combines highly parallelable computing cores with FP, BP, and WG engines and a 2D-mesh NoC, enabling distributed deployment and BP-based training of entire SNN models across computing cores."

Line 671: "Compared to prior studies (e.g., Loihi, TrueNorth) that focused exclusively on spike sparsity, our architecture fully leveraged the sparsity of spikes, $fire'$, and V/u in SNN training, ..."

For Detailed description of the BP and WG engines see **Results 2.2 Multi-Core Near-memory Neuromorphic Architecture for SNN Training (Line 237-265)** and sparse computation optimization during the backpropagation in **Results 2.4 Sparse Computation Optimization (Line 468-509)**.

We revised the description of NoC design in our multi-core architecture in **Results 2.2 Multi-Core Near-memory Neuromorphic Architecture for SNN Training. (Line 266-327; see details in our Reply to question 7)**.

For efficiently parallelizing training operations across multiple cores without compromising accuracy:

We described the detailed parallelized FP, BP, WG operations in our multi-core architecture for SNN training with Algorithm 1 in our **Reply to question 5**, achieving highly parallel computing at both the core and engine levels for SNN training. In this parallel computing design, all floating-point data are maintained in FP16 precision, and there are no training algorithm approximations introduced for the sake of parallelism-thereby excluding compromising accuracy during training.

We added an algorithm to describe the design of SNN training operations (see Suppl. Algorithm 1) and described the parallel computations in our architecture in **Results 2.3 Data Flow and Parallel FP-BP-WG (Line 353; see details in our Reply to question 5)**.

9: *Training SNNs in dynamically changing environments (e.g., edge devices) requires real-time adaptation, which is difficult to achieve. Are you going to tackle this challenge?*

Reply: Sincere thanks for the valuable questions. Table 6 shows the convergence speed under fine-tuning and full training of SNNs in our architecture. In the fine-tuning setting, we started with a pre-trained model, which subsequently processes and learns on new samples through BP training. Fine-tuning on traffic sign classification task (see Figure 5.h in the revised manuscript) in our architecture is quite fast and requires only 9.29s to complete. Fine-tuning on CIFAR-10 requires minutes, while full training from scratch on CIFAR-10 requires 2.5 hours. Thus, our architecture is more suitable for fine-tuning applications, which have been the primary applications in edge learnable environments given the computational constraints on edge devices. To tackle the challenge of real-time adaptation, it relies both on the hardware architecture and the algorithmic optimization, such as development of high-performance small-scale models.

Table 6. Time to Achieve Convergence in Our Architecture for Different Experimental Settings

Training Mode	Fine-Tuning	Fine-Tuning	Full Training
Dataset	Traffic Sign Classification and Recognition	CIFAR-10	CIFAR-10
Network	3-layer Spiking CNN	Spiking ResNet-18	Spiking ResNet-18
Time until Convergence	9.29s	9.19min	2.30h

We added Table 6 to the revised **Supplementary Information** as Suppl. Table 11. We added the information to **Results 2.3. Data Flow and Parallel FP-BP-WG Computations**. We now state:

Line 417: “Suppl. Table 11 shows the convergence speed of SNN training in our architecture. Fine-tuning Spiking-ResNet18 on newly emerged CIFAR-10 samples required minutes, while training from scratch on CIFAR-10 required 2.5 hours. Fine-tuning a shallow SNN on the traffic sign classification task required less than 10 seconds.”

We also discussed the real-time adaptation in **Discussion**. We now state:

Line 665: “The convergence speed of SNN training in our architecture indicates its real-time learning potential in fine-tuning conditions. To tackle the challenge of real-time adaptation, it relies on both the hardware architecture and the algorithmic optimization.”

10: *Are you addressing the overfitting issue in your study?*

Reply: Sincere thanks for this question. Our WG engine supports the weight decay technique, a classical algorithmic design to mitigate the overfitting effect. This technique constrains parameter magnitudes to improve generalization and is commonly used in applications (D'Angelo et al., 2024). We evaluated the performance of weight decay with Spiking ResNet-18 on CIFAR-100, and found that it improves the accuracy on unseen data by 1.30% (from 62.59% to 63.89%). Our implementation of weight decay partially mitigates overfitting but cannot completely resolve it. Further improvements could involve offline pre-training (Chen et al., 2023), data augmentation (Shorten et al., 2019), normalization (Huang et al., 2023), and so on.

We added the information of adaptation to weight decay in **Results 2.2 Multi-Core Near-memory Neuromorphic Architecture for SNN Training**. We now state:

Line 262: “The WG engine supports the weight decay mechanism (D'Angelo et al., 2024) to alleviate overfitting during training.”

We added the effect of weight decay in **Results 2.6 FPGA Implementation and Applications**. We now state:

Line 594: “For CIFAR-100, our implemented weight decay in the WG engine improves the accuracy on unseen

data by 1.30% (from 62.59% to 63.89%).”

Reference

- [D'Angelo et al., 2024] D'Angelo F, Andriushchenko M, Varre AV, et al. Why do we need weight decay in modern deep learning?. *Advances in Neural Information Processing Systems*, 37, 23191-23223 (2024).
- [Shorten et al., 2019] Shorten C, Khoshgoftaar TM. A survey on image data augmentation for deep learning. *Journal of big data*, 6(1), 1-48 (2019).
- [Huang et al., 2023] Huang L, Qin J, Zhou Y, Zhu F, Liu L, Shao L. Normalization techniques in training dnns: Methodology, analysis and application. *IEEE transactions on pattern analysis and machine intelligence*, 45(8), 10173-96 (2023).
- [Chen et al., 2023] Chen FL, Zhang DZ, Han ML, Chen XY, Shi J, Xu S, Xu B. Vlp: A survey on vision-language pre-training. *Machine Intelligence Research*, 20(1), 38-56 (2023).

11: I recommend that the authors clearly outline the challenges of multi-core neuromorphic architectures and then specify which issues their study aims to address. I think the authors need to describe their novelty very clearly and discuss in great detail how their approach differs from the state of the art.

Reply: We sincerely thank you for this suggestion. We revised the **Introduction** to clearly clarify the challenges of the training process in multi-core neuromorphic architectures and specify the issues our study aims to address. We now state:

Line 105: “High-efficiency computation for SNN training still faces fundamental challenges in parallel computation, inter- and intra-core communication, and storage usage. First, diverse computational forms, including both floating-point and binary (0/1) spike-based computations, and complex data dependencies across computing cores during SNN training, encompassing both spatial and temporal dimensions, make it difficult to achieve high-efficiency parallel computing. Second, bandwidth demand for data transmission is far more than that required in inference, and complex data transmission along both model layers and time steps in both sequent and reverse orders, increases the complexity for inter- and intra-core communication design. Finally, deep SNN training generates a large volume of intermediate data, requiring extensive access to off-chip memory (e.g., DRAM), leading to high power consumption. Reducing DRAM access is particularly significant for SNN training.”

We modified the description and discussion (in blue) on the novelty of our study in detail in the revised

Discussion. We now state:

Line 622: “To the best of our knowledge, we developed the first multi-core neuromorphic architecture supporting deep SNN training based on BP. This architecture combines highly parallelable computing cores with FP, BP, and WG engines and a 2D-mesh NoC, enabling distributed deployment and BP-based training of entire SNN models across computing cores. Previous neuromorphic computing approaches rely exclusively on GPUs (Rueckauer et al., 2022; Meng et al., 2022; Yin et al., 2023) or implement a hybrid training loop, combining inference on the neuromorphic platforms with BP on GPUs (Cramer et al., 2022; Friedmann et al., 2016). Recently, Renner et al. (2024) combined a gating mechanism with Hebbian learning to enable BP for a single hidden-layer fully connected network on Loihi, while its applicability to popular SNNs, such as convolutional SNNs, remains a question. Additionally, the proposed training paradigm consumes over 200 times the energy of inference, raising concerns about its training efficiency. Our architecture also shows high scalability with its 2D-mesh topology and the GALS design, exhibiting a near-linear increase of throughput with core number. Various scales of the architecture have been implemented successfully in FPGAs for supervised learning and federated learning, indicating its high flexibility in scale. The computation capacity of our 32-core architecture is clearly higher than that of prior neuromorphic architectures. It achieves a high energy efficiency of 1.05 TFLOPS/W @ 28nm, which is comparable to that of A100, DOJO, and Graphcore with 7nm technology.

Our architecture supports highly efficient parallel computing at both the engine and core levels in deep SNN training, a capability absent in previous neuromorphic architectures. Leveraging this hierarchical

parallelization strategy, the architecture achieves 15%~40% performance (fps) of an NVIDIA A100 GPU with only ~5% of A100 computing capability during SNN training, and 190%~330% performance of an NVIDIA Jetson AGX Orin. The convergence speed of SNN training in our architecture indicates its real-time learning potential in fine-tuning conditions. To tackle the challenge of real-time adaptation, it relies on both the hardware architecture and the algorithmic optimization.

Compared to prior studies (e.g., Loihi, TrueNorth) that focused exclusively on spike sparsity, our architecture fully leveraged the sparsity of spikes, $fire'$, and v_{uu} in SNN training, achieving 45%~60% reduction of energy consumption. Through single-bit spike storage, multiple dataflows within computing engines, and the inter- and intra-core parallelism, SNN training in our architecture achieved 40~75% reduction of DRAM occupancy and 55%~85% of DRAM access compared to GPU baselines, which not only significantly alleviated the DRAM storage requirements, facilitating the training in storage constrained conditions, but also improved the energy efficiency as energy consumption of DRAM access is an order higher than that of on-chip SRAM (Jouppi et al. 2021). DRAM usage optimization has not been the focus in previous neuromorphic architectures designed for inference and local learning.”

Reviewer #3: *In this paper the authors present a highly energy-efficient multi-core neuromorphic architecture for training deep spiking neural networks. While I do not have any major concerns regarding the proposed architecture, in my opinion, the paper is not of a high enough caliber for Nature Communications.*

Reply: We sincerely thank you for your comments on our work. Following your constructive comments, we added the substantial contents about scalability and flexibility, the code coverage, PVT variation, empirical training behavior, and training time information of our architecture, and described in details about inter-FPGA communication, performance comparison between FPGA and taped-out chip implementation of our architecture, and flexibility of our software toolchain. We also performed comparison to standard network training on edge devices (Jetson AGX Orin 64GB), achieving promising results. We acknowledged the limitation of FPGA implementation and pointed out that our architecture requires further assessment on taped-out chips in revised **Discussion**. We strongly believe that our work has been significantly improved with the help of your kind suggestions. Our work extends existing multi-core neuromorphic computing architectures in global learning capability, efficient parallel computing, DRAM usage and sparse computation optimization.

The presentation of the manuscript requires substantial improvement and instead of taping out a silicon chip, the authors validate their design using an FPGA implementation. The authors mention the use of the Eclipse IDE to develop a basic operator library – it is unclear to me why existing open-source intermediate representations were not adopted/extended and what novelty the automated network has.

Reply: We sincerely thank you for the comments on this important issue. We had investigated open-source software supporting SNN computations before we chose to develop our software toolchain with the custom operator library and intermediate representation (IR) for the following reasons:

(1). Open-source neuromorphic software frameworks such as Norse (Pehle et al., 2021), SpikingJelly (Fang et al., 2023), snnTorch (Eshraghian et al., 2023), Sinabs (Sheik et al., 2023), and Spyx (Heckel et al., 2024) are built upon PyTorch or JAX and share the intermediate representations (IRs) with these mainstream deep learning frameworks. However, PyTorch and JAX are primarily targeted on artificial neural networks (ANNs) rather than SNNs (Pedersen et al., Sep. 2024). These open-source IRs lack the explicit representation of the temporal dimension, rendering them incapable of directly and explicitly representing temporal evolution and spiking behavior of spiking neurons in SNNs. Thus, the SNN representation based on these

IRs becomes complex and inefficient, making it difficult to map them to hardware architectures. Figure 5a shows that PyTorch IR represents a simple Leaky Integrate-and-Fire (LIF) neuron with two time steps using numerous basic operations such as multiplications, additions, and reshaping, in which crucial components like time-step indexing, membrane potential updating, and spike generation of the LIF neuron are difficult to be extracted. In contrast, our custom IR introduces a dedicated Soma operation that concisely encapsulates all key aspects of the LIF neuron (Figure 5b), resulting in a straightforward and efficient representation.

Figure 5. (a) PyTorch IR for representing a two-time-step LIF model. (b) Our IR for representing a two-time-step LIF model.

Table 7 shows that our IRs of three typical SNN models are approximately 1/7,000 ~ 1/70,000 the size of the PyTorch IRs. The comparison in Table 8 indicates that PyTorch IR is efficient for representing ANNs.

(2). Neuromorphic software tools such as PyNN (Davison et al., 2019), NEST (Gewaltig et al., 2007), Nengo (Bekolay et al., 2014), and NEURON (Carnevale et al., 2006) are primarily designed for simulating biological neural networks and lack key operators of deep SNNs. For instance, PyNN does not provide operators such as convolution, pooling, nor does it support key SNN training mechanisms such as backpropagation and gradient computation. NEST and NEURON do not support convolution, pooling, BP-based learning mechanisms either. Tools like Lava (Williams et al., 2023), Corelet (Amir et al., 2013), and Rockpool (Muir et al., 2019) are designed for specific neuromorphic hardware platforms such as Loihi (Davies et al., 2018, 2021), TrueNorth (Merolla et al., 2014), and Xylo (Bos et al., 2022), lacking cross-platform generalizability.

Table 7. Comparison between PyTorch IR and Our IR for SNNs

Model	PyTorch IR	Our IR
S-ResNet18	28.5MB	4KB
S-ResNet50	141MB	11KB
S-VGG16	209MB	3KB

Table 8. Comparison between PyTorch IR and Our IR for SNNs

Model	ANN	SNN
ResNet18	11KB	28.5MB
ResNet50	26KB	141MB
VGG16	8KB	209MB

Together, we had not found efficient open-source IRs when we started to develop our software toolchain. We totally agree with the reviewer on the importance of leveraging open IRs to enhance the generalizability of our system. Leveraging on the recent progress in open-source IRs such as the general-purpose Neuromorphic Intermediate Representation (NIR, Pedersen et al., Sep. 2024), the cross-platform generalizability of our software chains could be improved through bidirectional conversion between our custom IR and general-purpose neuromorphic IRs in further study.

We added the explanation in **Methods 4.3 SNN Compilation and Deployment on the Multi-core Neuromorphic Architecture**, we now state:

Line 1103: “We chose to develop our software chain with the custom operator library and intermediate representation (IR) rather than open-source IR for two main reasons (see details in Suppl. Contents 1.3): Open-source neuromorphic software frameworks such as Norse, SpikingJelly, snnTorch, Sinabs, and Spyx built upon PyTorch or JAX shared IRs with these mainstream deep learning frameworks. The SNN representations based on these IRs were complex and inefficient (Suppl. Figure 9; Suppl. Table 8), making it difficult to map them to hardware architectures; Neuromorphic software tools such as PyNN, NEST, Nengo, and NEURON were primarily designed for simulating biological neural networks and lack key operators of deep SNNs.”

We added the above description of existing open-source intermediate representations to Supplementary Information as Suppl. Contents 1.3. Figure 5 and Tables 7-8 were also added as Suppl. Figure 9 and Suppl. Table 8, respectively.

We acknowledged this limitation and pointed out the promising further study in **Discussion**, we now state:

Line 709: “The cross-platform compatibility of our software toolchain is limited so far. Bidirectional conversion between our custom IR and general-purpose open-source neuromorphic IRs such as NIR could be promising in further study, rendering ours compatible with mainstream deep learning frameworks such as PyTorch, TensorFlow, and SpikingJelly, as well as with multiple hardware platforms.”

For the comments “*What novelty the automated network has.*”

We sincerely thank you for the comment. Our software toolchain enables one-time deployment of the entire

SNN model across all computing cores in a distributed manner flexibly in our multi-core architecture prior to training, with one layer or a segment of a layer allocated to an engine within a computing core. It automatically generates task-aware instruction flows for intra- and inter-core data transmission, data loading into the computing array, and computation execution for each core. During training, all computing cores autonomously detect their own input data, execute computations, and dispatch results in the preloaded manner. For each training batch, only one command is issued from the host to trigger the training in the multi-core architecture, while GPU-based training typically requires layer-by-layer control from the host. Thus, this automated mechanism significantly reduces control overhead.

We added this information to **Discussion**. We now state:

Line 694: “While GPU-based training typically requires layer-by-layer control from the host, the toolchain enables hierarchical parallelism for SNN training with one-time deployment of the entire model in our multi-core architecture, reducing control overhead significantly.”

Reference

- [Amir et al., 2013] Amir, A. et al. Cognitive computing programming paradigm: A corelet language for composing networks of neurosynaptic cores. In The 2013 International Joint Conference on Neural Networks (IJCNN), 1–10 (2013).
- [Bekolay et al., 2014] Bekolay, T. et al. Nengo: a python tool for building large-scale functional brain models. *Front. Neuroinform.* 7, 48 (2014).
- [Bos et al., 2022] Bos, H. & Muir, D. R. Sub-mW Neuromorphic SNN Audio Processing Applications with Rockpool and Xylo. In *Embedded Artificial Intelligence: Devices, Embedded Systems, and Industrial Applications*, River Publishers, pp. 69–78 (2022).
- [Carnevale et al., 2006] Carnevale, N. T. & Hines, M. L. *The NEURON Book* (Cambridge University Press, 2006).
- [Davies et al., 2018] Davies, M. et al. Loihi: A neuromorphic manycore processor with on-chip learning. *IEEE Micro* 38, 82–99 (2018).
- [Davies et al., 2021] Davies, M. et al. Advancing neuromorphic computing with Loihi: a survey of results and outlook. *Proc. IEEE* 109, 911–934 (2021).
- [Davison et al., 2009] Davison, A. P. et al. PyNN: a common interface for neuronal network simulators. *Front. Neuroinform.* 2, 388 (2009).
- [Eshraghian et al., 2023] Eshraghian, J. K. et al. Training spiking neural networks using lessons from deep learning. *Proc. IEEE* 111, 1016–54 (2023).
- [Fang et al., 2023] W. Fang, Y. Q. Chen, J. H. Ding, Z. F. Yu, T. Masquelier, D. Chen, L. W. Huang, H. H. Zhou, G. Q. Li, Y. H. Tian, SpikingJelly: An open-source machine learning infrastructure platform for spike-based intelligence. *Sci. Adv.* 9, eadi1480 (2023).
- [Gewaltig et al., 2007] Gewaltig, M.-O. & Diesmann, M. Nest (neural simulation tool). *Scholarpedia* 2, 1430 (2007).
- [Heckel et al., 2024] Heckel, K. M. & Nowotny, T. Spyx: A library for just-in-time compiled optimization of spiking neural networks. Preprint at <https://doi.org/10.48550/arXiv.2402.18994> (2024).
- [Merolla et al., 2014] Merolla, P. et al. A million spiking-neuron integrated circuit with a scalable communication network and interface. *Science* 345, 668–673 (2014).
- [Muir et al., 2019] Muir, D. R., Bauer, F. & Weidel, P. Rockpool documentation <https://doi.org/10.5281/zenodo.3773845>. (2019).
- [Pedersen et al., Sep. 2024] Pedersen, J. E. et al. Neuromorphic intermediate representation: A unified instruction set for interoperable brain-inspired computing. *Nat. Commun.* 15, 8122 (2024).
- [Pehle et al., 2021] Pehle, C. & Pedersen, J. E. Norse - A deep learning library for spiking neural networks. <https://doi.org/10.5281/zenodo.4422025> (2021).
- [Sheik et al., 2023] Sheik, S., Lenz, G., Bauer, F. & Kuepelioglu, N. SINABS: A simple Pytorch based SNN library specialised for Speck <https://github.com/svnsense/sinabs> (2023).
- [Stimberg et al., 2020] Stimberg, M., Goodman, D. F. & Nowotny, T. Brian2genn: accelerating spiking neural network simulations with graphics hardware. *Sci. Rep.* 10, 410 (2020).
- [Williams et al., 2023] Williams, M. G. K., Plank, P. & Shrestha, S. B. Lava - a software framework for neuromorphic computing. <https://github.com/lava-nc/lava> (2023).

1: The empirical training behavior is not well described, and it is unclear to me how scalable the architecture is.

Reply to the sub-question “The empirical training behavior is not well described”

We sincerely thank you for this comment. We provided additional information about empirical training behaviors (in blue) in **Results 2.6 FPGA Implementation and Applications** and in legend for Figure 5j-o. We now state:

Line 584: “Training from scratch of spiking-ResNet18 deployed on the 20-core architecture converged normally in image classification tasks on MNIST, CIFAR-10, and CIFAR-100 datasets (Figures 5c-d), with accuracies 99.23%, 85.5%, and 63.89%, respectively. The accuracy loss compared to GPU was 0.05%, 0.49%, and 2.41%. It required increased training epochs to obtain stable classification accuracy with the increased complexity of the datasets, from nearly 1 epoch for MNIST, 30 epochs for CIFAR-10, to 70 epochs for

CIFAR-100 (Figure 5c-d). For CIFAR-100, our implemented weight decay in the WG engine improves the accuracy on unseen data by 1.30% (from 62.59% to 63.89%). We then implemented continual learning (Figure 5f) and federated learning (Figure 5j) applications in the FPGAs for a smart traffic scenario (Figure 5e). In continual learning, pre-trained SNN models were fine-tuned based on BP with samples of new classes. The continual learning improved inference accuracy on mixed new and old data by 17.97% (from 75.01% to 92.98%) for the Traffic Sign Classification and Recognition dataset (Figure 5g, orange line vs blue line), 33.21% (from 58.09% to 91.30%) for the Vehicle and Pedestrian Detection dataset (Figure 5h), and 42.41% (from 43.06% to 85.47%) for the DVS-Gesture dataset (Figure 5i). In the 5-worker federated learning, the federated model aggregated gradient information generated separately by each worker based on its local dataset. Figures 5l-o show that the federated model (red line) significantly outperformed worker models (gray lines), achieving accuracies of 94.75%, 94.09%, 94.85%, and 84.03% on the Traffic Sign Classification and Recognition dataset, the Vehicle and Pedestrian Detection dataset, the Pedestrian Augmented Traffic Light dataset, and the DVS-Gesture dataset, respectively.”

Line 577: “(j) Illustration of federated learning at the edge. Orange line: accuracy with continual learning; Blue line: accuracy of inference without any learning. (k) Implementation of the 5-worker federated learning on FPGAs. Each worker was implemented on one FPGA. The federated model in server aggregated gradient information generated separately by each worker based on each’s local dataset. Both worker 1 and server were implemented on FPGA1. (l)-(o) Accuracy curves of the five workers (5 gray lines in each plot) and the federated model (FL, red line in each plot) on the Traffic Sign Classification and Recognition dataset, the Vehicle and Pedestrian Detection dataset, the Pedestrian Augmented Traffic Light Dataset, and the DVS-Gesture dataset.”

Renly to the sub-question “how scalable the

Scalability is important for the multi-core architecture. Our multi-core architectures adopted a 2D mesh topology with high scalability. We estimated the throughputs (fps) our multi-core neuromorphic architectures with 16 (4×4), 32 (4×8), and 64 (8×8) cores during SNN training. Figure 6a shows that throughput (fps) during S-ResNet18, S-ResNet50, and S-VGG16 trainings increases almost linearly with the number of cores, with the slope (fps/core) metric holding steady with the number of cores during these trainings (Figure 6b). The results support the high scalability of our architecture.

In addition, the 2D mesh NoC network in our 32-core architecture can be extended in four directions through the six custom SerDes to support high inter-chip scalability. Figure 7 shows our inter-chip communication design.

Figure 7. Inter-chip communication design. The 32-cores architecture supports extension in the east-west direction via two SerDes links with 8 ports, and in the north-south direction via one SerDes link with 4 ports.

Figure 6. The effect of increasing core number of our multi-core architecture on performance of SNN training. (a) Throughput (fps); (b) Slope (fps/core) of throughput change with core number.

Our architecture supports extension in the east-west direction via two SerDes links with 8 ports, and in the north-south direction via one SerDes link with 4 ports. Thus, adjacent chips can directly interface across chip boundaries, thereby supporting chip tiling and high chip-level scalability.

We added Figures 6a-b to Figure 3 as Figures 3p-q. We also added description of scalability of our architecture to **Results 2.3 Data Flow and Parallel FP-BP-WG**. We now state:

Line 455: “Our multi-core architecture showed high scalability. Its throughputs (fps) increased almost linearly with the number of cores (Figure 3p), with the slope (fps/core) metric holding steady with the number of cores during these trainings (Figure 3q). It also supported inter-chip extension in 4 directions for chip-level scalability (Suppl. Figure 2).”

And we discussed the scalability of our architecture in **Discussion**. We now state:

Line 643: “Our architecture also shows high scalability with its 2D-mesh topology and the GALS design, exhibiting a near-linear increase of throughput with core number.”

We added Figure 7 and the description of inter-chip communication design into the revised **Supplementary Information** as Suppl. Figure 2.

2: For functional verification, what was the code coverage?

Reply: Thank you for this good question. We performed functional verification of our code using Verdi (version 2020.03, Synopsys Inc.). For computing core, code coverage scores of FP engine, BP engine, and WG engine are 96.95%, 97.75%, and 99.15%, respectively. For NoC, code coverage scores of Router, Controller, Network Interface, Dispatch Unit, and DMA are 95.35%, 97.82%, 95.70%, 97.39%, and 95.11%, respectively. Figure 8 shows the details of the code coverage in the RTL functional verification.

Figure 8. The code coverage in the RTL functional verification.

We added Figure 8 to Supplementary Data as Suppl. Figure 7 and added the description of code coverage to **Results 2.5 Area and Power Consumption Analysis**. We now state:

Line 534: “The code coverage scores in the RTL functional verification were higher than 95% (Suppl. Figure 7).”

We added this information in **Discussion**. We now state:

Line 700: “Our multi-core architecture has been assessed in FPGA implementation, RTL functional verification including code coverage and time closure analysis, and PVT variation analysis, demonstrating the stability and reliability of our multi-core architecture design under different conditions.”

3: How flexible is the developed frontend/compiler/architecture? What are the exact steps required to deploy a new network and setup its training and what are the associated limitations?

Sincere thanks for the important questions.

Reply to the sub-question “How flexible is the developed frontend/compiler/architecture?”:

The Frontend supports construction of various deep convolutional SNN models through flexible composition of basic operators from our custom SNN operator library, covering a large number of recent SNN models (as listed in Suppl. Table 12). Additionally, it provides a C-language interface to support rapid expansion of new operators at the kernel level to follow the latest SNN progress.

The compiler automatically constructs various hardware-friendly computing graphs compatible with our architecture depending on various SNN tasks. It also automatically generates task-aware instruction flows for intra- and inter-core data transmission and computation execution associated with each core.

Our 2D-mesh neuromorphic architecture offers high flexibility in scale. The architecture is designed as 8×4 2D-mesh, while various mesh scales (e.g., 2×10, 2×2) of the architecture have been implemented on FPGAs successfully. In combination with the software toolchain, our architecture supported different training approaches, such as supervised learning and federated learning.

We added this information to **Methods 4.3 SNN Compilation and Deployment on the Multi-core Neuromorphic Architecture** and **4.4 FPGA Implementation**. We now state:

Line 1061: “The Frontend supported construction of various deep convolutional SNN models through flexible composition of basic operators from our custom SNN operator library, covering a large number of recent SNN models (as listed in Suppl. Table 12). Additionally, it provided a C-language interface to support the rapid development of new operators at the kernel level to follow the latest SNN progress. The compiler automatically constructed various hardware-friendly computing graphs compatible with our architecture, depending on various SNN tasks.”

Line 1084: “Task-aware instruction flows for intra- and inter-core data transmission and computation execution associated with each core were generated automatically based on the device-side files.”

Line 1131: “...Various mesh scales (e.g., 2×10, 2×2) of the architecture had been implemented on FPGAs successfully.”

We highlighted the flexibility of software toolchain and our architecture in **Discussion**. We now state:

Line 690: “We also developed a software toolchain for our architecture, allowing construction of various deep SNN models and automatic construction of hardware-friendly computing graphs depending on various SNN tasks.”

Line 646: “...Various scales of the architecture have been implemented successfully in FPGAs for supervised learning and federated learning, indicating its high flexibility in scale.”

Reply to the sub-question “What are the exact steps required to deploy a new network and setup its training?”: The exact steps of SNN deployment and training setup in our multi-core architecture are shown in Figure 9, including the following steps:

Step 1: Model Construction.

Generate the SNN model represented in C Language through composing basic operators in our custom operator library in Frontend.

Step 2: Model Compilation.

IR Generation: Through operator fusion compatible with our architecture, the IR representing the hardware-friendly computing graphs are generated from the C Language represented SNN model.

Device-side File Generation: The IR of the SNN model is mapped to the multi-core architecture, with each cores deploying one portion of the IR. Device-side Files for each core are generated, which describe computational tasks and information for data transmission associated with each core.

Step 3: Training Initialization.

According to the device-side file of each core, Training Initialization for each core includes storage (SRAM, DRAM) allocation, loading model parameters and samples, and generation of instruction flow.

Step 4: Training Execution.

The host sends a command to the first computing core processing the first layer of the SNN model to trigger the start of SNN training. All computing cores then execute their assigned computation tasks in parallel.

Figure 9. SNN deployment and training setup in our multi-core architecture.

We added the above description of steps to Supplementary Information in Suppl. Content 1.2, and Figure 9 as Suppl. Figure 5b. We also add the information to **Methods 4.3 SNN Compilation and Deployment on the Multi-core Neuromorphic Architecture**. We now state:

Line 1099: “The exact steps of SNN deployment and training setup in our multi-core architecture (Suppl. Figure 5b) are described in Suppl. Contents 1.2.”

Reply to the sub-question “what are the associated limitations?”:

The software toolchain is well compatible with our multi-core training architecture. The main limitation of it is in cross-platform compatibility. We acknowledged this limitation and point out further study to address this issue in **Discussion**. We now state:

Line 709: “The cross-platform compatibility of our software toolchain is limited so far. Bidirectional conversion between our custom IR and general-purpose open-source neuromorphic IRs such as NIR could be promising in further study, rendering ours compatible with mainstream deep learning frameworks such as PyTorch, TensorFlow, and SpikingJelly, as well as with multiple hardware platforms.”

4: Was any form of PVT (process, voltage, temperature) variation investigated?

Reply: Thanks for the kind suggestion. Table 9 shows power consumption analysis using Design Compiler across four corners based on TSMC N28HPC technology. The power consumption at the tt_0.9V_25°C, ss_0.81V_125°C, ss_0.81V_-40°C, and ff_1.05V_-40°C corners are 14.493, 25.254, 9.503 and 22.620 W, respectively. Figure 10

Table 9. PVT Variation of Multi-Core Neuromorphic Architecture

Corners	Power of 32-Core Architecture(W)
tt_0.9V_25°C	14.493
ss_0.81V_125°C	25.254
ss_0.81V_-40°C	9.503
ff_1.05V_-40°C	22.620

shows consistent timing closure of our architecture design in the four corners. These analyses under different

voltage and temperature support the stability and reliability of our multi-core architecture design under various conditions.

Figure 10. Timing closure of modules in our architecture

Table 9 and Figure 10 are added to the revised **Supplementary Information** as Suppl. Table 9 and Suppl. Figure 10, respectively.

We added this information in **Results 2.5 Area and Power Consumption Analysis**. We now state: **Line 516**: “All reported power consumption data is under the tt_0.9V_25°C condition. PVT (process, voltage, temperature) variation (Suppl. Table 9) and timing closure analysis (Suppl. Figure 10) support the stability and reliability of our multi-core architecture design under different conditions.”

We added this information in **Discussion**. We now state:

Line 700: “Our multi-core architecture has been assessed in FPGA implementation, RTL functional verification including code coverage and time closure analysis, and PVT variation analysis, ...”

5: How is communication handled between FPGA platforms? If a chip were to be taped out, how would this alter the performance?

Sincere thanks for the important questions. The two questions are answered separately as follows:

Reply to the sub-question “How is communication handled between FPGA platforms?”

Our FPGA system includes five FPGA cards. 4 (2×2) computing cores are deployed on each FPGA card, and the five FPGA cards are interconnected to form a 2×10 2D Mesh architecture with 20 cores. Inter-FPGA communication is via 100G CMAC (Xilinx CMAC), with two routers on one FPGA connected with two routers on the other FPGA (Figure 11). We designed a unified packet format (Figure 12), with the packets transmitted between cores within an FPGA through 32B flits.

Figure 11. Inter-FPGA communication. FPGAs are interconnected via 100G Ethernet links. The Ethernet Adapter (EA) module implements bit-width conversion (BWC) and clock domain crossing (CDC) processing between Router and CMAC interfaces. The EA transmitter merges dual-router channels through priority-based packet scheduling, while the EA receiver dynamically selects virtual channels and routing paths based on packet header semantics.

Figure 12. Design of Network Data Packets. (a)The structure of messages, packets, and head flits designed for NoC in the multi-core architecture. (b)The function description of each field in the head flit.

We developed an Ethernet Adapter (EA, Figure 11) module to address bus width mismatch, channel merging and splitting, clock frequency mismatch between the routers and CMAC for the inter-FPGA connection:

The 32B flits transferred supported by the 256bit AXI-S bus from two routers on one-side FPGA are converted in a two- to-one manner in the BWC of EA (Figure 11), yielding the 64B flits supported by 256bit AXI-S bus connected with 100G CMAC. The merge circuit in EA_Transmitter (Figure 11) employs a Round-Robin scheduling policy across routers and a priority-based policy across VCs to select one of four VC channels from two routers on one side for current transmission. Data is transmitted via 100G CMAC to the EA on the other side. The split circuit in EA_Receiver decodes the VC channel and target Router information of Packets based on the packet header and directs the Packet to one of four VC channels, then, the packets are converted into 32B flit format and transferred to the target router. The asynchronous FIFO buffers in the EA module support cross-clock domain data transmission between the Router side (200MHz) and CMAC side (333MHz).

The inter-FPGA communication supports data transmission between any cores deployed on different FPGAs through XY routing strategy. Figure 13 shows the FPGA's block design with Vivado (2021.1), where 2 EAs and 2 CMACs in an FPGA are encapsulated together as EA-CMAC0 and EA-CMAC1.

Figure 13. Partial FPGA Block Design of Multi-Core Architecture.

We added a description of inter-FPGA communications in **Methods 4.4 FPGA Implementation**. We now state:

Line 1125: “Inter-FPGA communication was via 100G CMAC (Xilinx CMAC). We developed an Ethernet Adapter module to address bus width mismatch, channel merging and splitting, clock frequency mismatch between the routers and CMAC for the inter-FPGA connection (see details in Suppl. Content 1.4).”

We also added a description of inter-FPGA communications in **Results 2.6 FPGA Implementation and Applications**. We now state:

Line 568: “The inter-FPGA communication (Suppl. Figure 8) supported data transmission between any cores deployed on different FPGAs through XY routing strategy.”

The detailed description of inter-FPGA communications is added to the revised **Supplementary Information** as Suppl. Content 1.4. Figures 11 and 13 are added into Suppl. Figure 8, and Figure 12 into Suppl. Figure 3.

Reply to the sub-question “If a chip were to be taped out, how would this alter the performance?”

Table 10 compares the performance on FPGA and the taped-out chip. The taped-out chip should achieve significantly higher clock frequencies for both computing cores and routers, and higher inter-core bandwidth.

Table 10. Comparison of Architecture Performance on FPGA and the Taped-Out Chi

	FPGA	Chip
Cores	20	32
NoC Topology	2x10 (2D mesh)	4x8 (2D mesh)
Engine Clock(MHz)	150	500
Router Clock (MHz)	200	667
Inter-core Bandwidth (Gbps)	50 (within FPGA), 100 (inter- FPGAs, 2 core – 2 core)	170
Routing Strategy	X-Y Routing	X-Y Routing

Table 10 is added to the revised **Supplementary Information** as Suppl. Table 10. We acknowledged the limitation of FPGA implementation and point out further study to address this issue in **Discussion**. We now state:

Line 700: “Our multi-core architecture has been assessed in FPGA implementation, RTL functional verification including code coverage and time closure analysis, and PVT variation analysis, demonstrating the stability and reliability of our multi-core architecture design under different conditions. It still requires further assessment on taped-out chips, and the taped-out chip should achieve significantly higher performance in clock frequencies and data transmission bandwidth (Suppl. Table 10).”

6: *The training power is well reported, but what about the time?*

Reply: Sincere thanks for your kind reminder. The training of Spiking-ResNet18, -ResNet50 and -VGG16 in our architecture achieved averaged performance (fps) of 233.8, 115.8, and 44.8, respectively. We added this information to in **Results 2.2.3 Data Flow and Parallel FP-BP-WG Computations**. We now state:

Line 399: “In the software simulations, our 32-core architecture achieved the average performance (frames per second, fps) of 233.8, 115.8, and 44.8 during training of Spiking-ResNet18, -ResNet50, and -VGG16, respectively.”

Table 6 (see also **Reply to question 9 of Reviewer 1**) shows the convergence speed under fine-tuning and full training of SNNs in our architecture.

Table 6. Time to Achieve Convergence in Our Architecture

Training Mode	Fine-Tuning	Fine-Tuning	Full Training
Dataset	Traffic Sign Classification and Recognition	CIFAR-10	CIFAR-10
Network	3-layer Spiking CNN	Spiking ResNet-18	Spiking ResNet-18
Time until Convergence	9.29s	9.19min	2.30h

We added Table 6 to the revised **Supplementary Information** as Suppl. Table 11. We added the information to **Results 2.3 Results 2.2.3 Data Flow and Parallel FP-BP-WG Computations**. We now state: **Line 417:** “Suppl. Table 11 shows the convergence speed of SNN training in our architecture. Fine-tuning Spiking-ResNet18 on newly emerged CIFAR-10 samples required minutes, while training from scratch on

CIFAR-10 required 2.5 hours. Fine-tuning a shallow SNN on the traffic sign classification task required less than 10 seconds.”

7: Why was FP16 used over BF16?

Reply: We fully acknowledged the common practice of BF16 in conventional ANN training. Our choice of FP16 was aligned with the common usage of FP16 in SNN trainings in previous works. Table 1 (see appears in the **Reply to question 1 of Reviewer 1**) summarizes SNN papers over the past five years in top-tier conferences (NeurIPS, CVPR, ICML, ICCV, ICLR, ECCV, AAAI) and journals (e.g., TPAMI), in which all studies that reported explicitly computational accuracy metrics used FP16 in SNN trainings. Therefore, we adopted FP16 in this study.

Table 1. Representative Convolutional Spiking Neural Networks

	Method	Year	Publication	Model Structure	FP16/BF16
1	PLIF	2021	CVPR [11]	Simple CNNs	n/a
2	SEW	2021	NeurIPS [21]	ResNet-18, -34, -50, -101, -152	n/a
3	BNTT	2021	Front. Neurosci. [31]	VGG-9, -11	n/a
4	STBP-tdBN	2021	AAAI [41]	ResNet-19, -34, -50	n/a
5	DSpike	2021	NeurIPS [51]	ResNet-18, -34; VGG-16	n/a
6	Diet-SNN	2021	TNNLS [61]	VGG-16; ResNet-20	n/a
7	Tandoom	2021	TPAMI [71]	VGG-11, -16	n/a
8	Spiking ResNet	2021	TNNLS [81]	ResNet-18, -34, -50	n/a
9	TTRBR	2022	Neural Networks [91]	ResNet-18, -20, -32, -56, -110	n/a
10	LTL	2022	NeurIPS [101]	VGG-11, -16; ResNet-20	n/a
11	RecDis-SNN	2022	CVPR [111]	ResNet-19, -34; VGG-16	n/a
12	NDA	2022	ECCV [121]	VGG-11, -19; ResNet-19, -34	FP16
13	Temporal Pruning	2022	ECCV [131]	VGG16	n/a
14	TET	2022	ICLR [141]	ResNet-19, -34; VGGSNN	n/a
15	n/a	2022	AAAI [151]	VGG-16; ResNet-18, -20	n/a
16	QCFS	2022	ICLR [161]	VGG16; ResNet-18, -20, -34	n/a
17	DSR	2022	CVPR [171]	ResNet-18, -34; VGG-11	FP16
18	OTTT	2022	NeurIPS [181]	VGG-7; ResNet-34	FP16
19	LATS	2023	ICLR [191]	ResNet-18, -50	FP16
20	SML	2023	ICML [201]	ResNet-18, -19, -34; VGGSNN	FP16
21	PSN	2023	NeurIPS [211]	ResNet-18, -34; VGGSNN	FP16
22	SSF	2023	ICCV [221]	ResNet-18, -34; VGG-11	n/a
23	Dual-Phase	2023	TNNLS [231]	ResNet-18, -20, -34; VGG-16	n/a
24	Fast-SNN	2023	TPAMI [241]	VGG-11, 16; ResNet-18	n/a
25	Attention SNN	2023	TPAMI [251]	ResNet-18, -34, -104	FP16
26	MPBN	2023	ICCV [261]	ResNet-19, -20, -34; VGG-16	n/a
27	KDSNN	2023	CVPR [271]	ResNet-18; VGG-11	n/a
28	SLTT	2023	ICCV [281]	ResNet-18, -34, -50, -101; VGG-11	FP16
29	LocalZO	2023	NeurIPS [291]	ResNet-19, -34; VGGSNN	n/a
30	IM-LIF	2024	IEEE TETCI [301]	ResNet-19; VGGSNN	n/a

31	Parameter Calibration	2024	IJCV [311]	VGG-16; ResNet-20, -34	n/a
32	LM-H	2024	ICLR [321]	ResNet-18, -34, -50; VGG-16	FP16
33	BKDSNN	2024	ECCV [321]	ResNet-19, -34, -50	FP16
34	T-RevSNN	2024	ICML [331]	ResNet-18	FP16
35	n/a	2024	CVPR [341]	ResNet-34, -50, -19, -20	n/a
36	FSTA-SNN	2025	AAAI [351]	ResNet-18, -34	FP16

We add our rationale to use FP16 to **Methods 4.2.A. Multi-core Neuromorphic Architecture Design**. We now state:

Line 781: “...with a computational precision of FP16 due to its common usage in SNN training (Suppl. Table 12).”

We added Table 1 to the revised **Supplementary Information** as Suppl. Table 12 with its references.

8: *I understand a direct comparison is difficult to make, but to get some idea of the value proposition of the architecture, it would be appreciated if the authors could make some comparison to standard network training on edge devices, e.g., mGPUs or embedded systems. The investigated networks are relatively small, so it may be feasible to fine-tune such networks in real-time using more standard hardware platforms.*

Reply: Sincere thanks for your kind suggestion. We performed SNN training on a Jetson edge system (Jetson AGX Orin 64GB @ 8nm with 85 FP16 Sparse TFLOPs, Figure 14a) for comparison. Training of Spiking ResNet-18, -50 and VGG-16 on the system achieved 123.3 fps, 35.0 fps, and 21.0 fps, respectively (Figure 14 b-d). On average, the performance of our architecture was 1.9, 3.3, and 2.1 times that of the NVIDIA Jetson AGX Orin, respectively (Figure 14 b-d), indicating the high performance of our architecture over popular edge AI system in SNN trainings.

Figure 14. (a) Jetson AGX Orin. (b-d) Comparison of training performance (fps) between our architecture and Jetson AGX Orin during the three training, respectively.

We added results in Figure 14 b-d to Figure 3m-o in the revised manuscript, and describe them in **Results 2.3 Data Flow and Parallel FP-BP-WG Computations**, we now state:

Line 413: “The performance of our architecture was 190%, 330%, and 210% of that of the NVIDIA Jetson AGX Orin, respectively (Figure 3m-o), indicating the high performance of our architecture over popular edge AI system in SNN training.”

We highlighted this finding in **Discussion**, we now state:

Line 663: “... and 190%~330% performance of an NVIDIA Jetson AGX Orin.”

References for Table 1

[11. Fang, W. et al. Incorporating learnable membrane time constant to enhance learning of spiking neural networks. In ICCV (2021). [21. Fang, W. et al. Deep residual learning in spiking neural networks. In NeurIPS (2021).

- [31. Kim, Y. & Panda, P. Revisiting batch normalization for training low-latency deep spiking neural networks from scratch. *Front. Neurosci.* (2020).
- [41. Zheng, H., Wu, Y., Deng, L., Hu, Y. & Li, G. Going deeper with directly-trained larger spiking neural networks. In *AAAI* (2021).
- [51. Li, Y. et al. Differentiable spike: Rethinking gradient-descent for training spiking neural networks. In *NeurIPS* (2021).
- [61. Rathi, N. & Roy, K. Diet-snn: A low-latency spiking neural network with direct input encoding and leakage and threshold optimization. *TNNLS* (2021).
- [71. Wu, J. et al. Progressive tandem learning for pattern recognition with deep spiking neural networks. *IEEE Transactions on Pattern Analysis Mach. Intell.* 44, 7824–7840 (2021).
- [81. Hu, Y., Tang, H., Wang, Y. & Pan, G. Spiking deep residual network. *TNNLS* (2021).
- [91. Meng, Q. et al. Training much deeper spiking neural networks with a small number of time-steps. *Neural Networks* 153, 254–268 (2022).
- [101. Yang, Q. et al. Training spiking neural networks with local tandem learning. In *NeurIPS* (2022).
- [111. Guo, Y. et al. Reccdis-snn: Rectifying membrane potential distribution for directly training spiking neural networks. In *CVPR* (2022).
- [121. Li, Y., Kim, Y., Park, H., Geller, T. & Panda, P. Neuromorphic data augmentation for training spiking neural networks. In *ECCV* (2022).
- [131. Chowdhury, S. S., Rathi, N. & Roy, K. Towards ultra low latency spiking neural networks for vision and sequential tasks using temporal pruning. In *ECCV* (2022).
- [141. Deng, S., Li, Y., Zhang, S. & Gu, S. Temporal efficient training of spiking neural network via gradient re-weighting. In *ICLR* (2022).
- [151. Bu, T., Ding, J., Yu, Z. & Huang, T. Optimized potential initialization for low-latency spiking neural networks. In *AAAI* (2022).
- [161. Bu, T. et al. Optimal ANN-SNN conversion for high-accuracy and ultra-low-latency spiking neural networks. In *ICLR* (2022).
- [171. Meng, Q. et al. Training high-performance low-latency spiking neural networks by differentiation on spike representation. In *CVPR* (2022).
- [181. Xiao, M., Meng, Q., Zhang, Z., He, D. & Lin, Z. Online training through time for spiking neural networks. In *NeurIPS* (2022).
- [191. Chen, Y. et al. A unified framework for soft threshold pruning. In *ICLR* (2023).
- [201. Deng, S., Lin, H., Li, Y. & Gu, S. Surrogate module learning: Reduce the gradient error accumulation in training spiking neural networks. In *ICML* (2023).
- [211. Fang, W. et al. Parallel spiking neurons with high efficiency and ability to learn long-term dependencies. In *NeurIPS* (2023).
- [221. Wang, J. et al. SSF: Accelerating training of spiking neural networks with stabilized spiking flow. In *ICCV* (2023).
- [231. Wang, Z. et al. Toward high-accuracy and low-latency spiking neural networks with two-stage optimization. *TNNLS* (2023).
- [241. Hu, Y., Zheng, Q., Jiang, X. & Pan, G. Fast-SNN: Fast spiking neural network by converting quantized ann. *IEEE Transactions on Pattern Analysis Mach. Intell.* 45, 14546–14562 (2023).
- [251. Yao, M. et al. Attention spiking neural networks. *IEEE transactions on pattern analysis machine intelligence* 45, 9393–9410 (2023).
- [261. Guo, Y. et al. Membrane potential batch normalization for spiking neural networks. In *ICCV* (2023).
- [271. Xu, Q. et al. Constructing deep spiking neural networks from artificial neural networks with knowledge distillation. In *CVPR* (2023).
- [281. Meng, Q. et al. Towards memory- and time-efficient backpropagation for training spiking neural networks. In *ICCV* (2023).
- [291. Mukhoty, B. et al. Direct training of snn using local zeroth order method. In *NeurIPS* (2023).
- [301. Lian, S., Shen, J., Wang, Z. & Tang, H. Im-lif: Improved neuronal dynamics with attention mechanism for direct training deep spiking neural network. *IEEE Transactions on Emerg. Top. Comput. Intell.* 8, 2075–2085 (2024).
- [311. Li, Y., Deng, S., Dong, X. & Gu, S. Error-aware conversion from ann to snn via post-training parameter calibration. *Int. J. Comput. Vis.* 132, 3586–3609 (2024).
- [321. Hao, Z. et al. A progressive training framework for spiking neural networks with learnable multi-hierarchical model. In *The Twelfth International Conference on Learning Representations* (2024).
- [331. Xu, Z., You, K., Guo, Q., Wang, X. & He, Z. Bkdsnn: Enhancing the performance of learning-based spiking neural networks training with blurred knowledge distillation. In *European Conference on Computer Vision*, 106–123 (Springer, 2024).
- [341. Hu, J. et al. High-performance temporal reversible spiking neural networks with $O(1)$ training memory and $O(1)$ inference cost. In *Forty-first International Conference on Machine Learning* (2024).
- [351. Shen, G., Zhao, D., Li, T., Li, J. & Zeng, Y. Are conventional snns really efficient? a perspective from network quantization. In *Proceedings of the IEEE/CVF Conference on Computer Vision and Pattern Recognition*, 27538–27547 (2024).
- [361. Yu, K., Zhang, T., Wang, H. & Xu, Q. Fsta-snn: Frequency-based spatial-temporal attention module for spiking neural networks. In *Proceedings of the AAAI Conference on Artificial Intelligence*, vol. 39, 22227–22235 (2025).